# Investigating spatial heterogeneity within fracture networks using hierarchical clustering and graph distance metrics

Rahul Prabhakaran[1,2], Giovanni Bertotti[1], Janos Urai[3], and David Smeulders[2]

[1]Department of Geoscience and Engineering, Delft University of Technology, Delft, the Netherlands
[2]Department of Mechanical Engineering, Eindhoven University of Technology, the Netherlands
[3]Structural Geology, Tectonics and Geomechanics, RWTH Aachen University, Aachen, Germany

**Correspondence:** Rahul Prabhakaran (r.prabhakaran@tudelft.nl)

**Abstract.**

Rock fractures organize as networks, exhibiting natural variation in their spatial arrangements. Therefore, identifying, quantifying, and comparing variations in spatial arrangements within network geometries are of interest when explicit fracture representations or discrete fracture network models are chosen to capture the influence of fractures on bulk rock behaviour. Treating fracture networks as spatial graphs, we introduce a novel approach to quantify spatial variation. The method combines graph similarity measures with hierarchical clustering and is applied to investigate the spatial variation within large-scale 2D fracture networks digitized from the well-known Lilstock limestone pavements, Bristol Channel, UK. We consider three large, fractured regions, comprising nearly 300,000 fractures spread over 14,200 sq.m. from the Lilstock pavements. Using a moving-window sampling approach, we first subsample the large networks into sub-graphs. Four graph similarity measures: fingerprint distance, D-measure, NetLSD, and portrait divergence, that encapsulate topological relationships and geometry of fracture networks, are then used to compute pair-wise sub-graph distances serving as input for the statistical hierarchical clustering technique. In the form of hierarchical dendrograms and derived spatial variation maps, the results indicate spatial autocorrelation with localized spatial clusters that gradually vary over distances of tens of metres with visually discernable and quantifiable boundaries. Fractures within the identified clusters exhibit differences in fracture orientations and topology. The comparison of graph similarity-derived clusters with fracture persistence measures indicates an intra-network spatial variation that is not immediately obvious from the ubiquitous fracture intensity and density maps. The proposed method provides a quantitative way to identify spatial variations in fracture networks, guiding stochastic and geostatistical approaches to fracture network modelling.

## 1 Introduction

Fracture networks in rocks develop due to loading paths that vary over geological time-scale (Laubach et al., 2019). The evolution of the network exhibits characteristics of a complex system. There is feedback between the evolving spatial structure and the rock substrate in which the networks are positioned (Laubach et al., 2018). The resulting spatial arrangement that emerges after cumulative network evolution is of considerable interest as it influences flow, transport, and geomechanical stability in multiple anthropogenic subsurface applications such as geothermal energy (Vidal et al., 2017), nuclear waste disposal (Wang

and Hudson, 2015), aquifer management (Witherspoon, 1986), and hydrocarbon exploitation (Nelson, 2001). Systematically documenting near-surface fracture patterns is essential, for example, in mining applications where fracture patterns often provide clues to ore deposit patterns (Jelsma et al., 2004), and in geotechnical engineering, where fractures influence stability in human-made structures such as tunnels (Lei et al., 2017).

An important property of natural fracture networks is that of *spatial organization*, which means that the arrangements are not random but follow a statistically discernable pattern. One can view the spatial arrangement of fractures as a set of objects within a geographical reference system. Within such a framework, fracture objects are either regularly spaced, irregularly spaced with statistically significant regions of close spacing, and irregularly spaced with statistically insignificant regions of close spacing (Laubach et al., 2018). An alternate framework is a network, where fracture objects are described in relation to one another (Valentini et al., 2007; Andresen et al., 2013; Sanderson and Nixon, 2015). Spatial variations in fracture network organization are quite common. The physical phenomena commonly used to explain spatial variation in fracture arrangements are stress shadowing, layer thickness differences, host rock lithology, layered mechanical anisotropy, high-strain events such as faulting/folding, and diagenesis. However, it is generally not easy to associate a type of spatial arrangement to any unique set of input boundary conditions as similar loading paths can lead to diverging patterns, and dissimilar loading paths can lead to converging patterns (Laubach et al., 2019).

Quantifying variations in spatial arrangements of fractures involves the sampling of fracture data. Such quantifications can be in the form of 1D (using scanline methods, borehole sampling), in 2D (fracture trace maps from outcrop imagery), or 3D (ground-penetrating radar, microseismic). 1D scanlines provide a method to quantify arrangements and variation, and several statistical measures have been proposed, such as fracture spacing (Priest and Hudson, 1976), fracture intensity (Dershowitz and Herda, 1992), coefficient of variation (Gillespie et al., 1993), normalized correlation count (Marrett et al., 2018), and cumulative spacing derivative (Bistacchi et al., 2020). These measurements, however, only indicate the variation of fracture arrangements on the scanline and fail to depict the variation in directions away from the scanline direction. Scanlines do not provide information on properties such as fracture length, spatial arrangements, and relationships with other fractures.

2D fracture trace maps are especially useful as this type of data combines both geometric and topological information in the form of a network. Recent advances in UAV-photogrammetry (Bemis et al., 2014; Bisdom et al., 2017) and automated image processing algorithms (Prabhakaran et al., 2019) have led to large datasets of 2D fracture traces that reveal much more about network attributes than is possible from 1D sampling. Given such large datasets with rich information, it is pertinent to directly quantify spatial variation from the network structure. Spatial fracture persistence (Dershowitz and Herda, 1992) can quantify 2D spatial variation but only considers some aspects of the network (such as the sum of trace lengths, number of traces, etc., within a sampling region). Thus, there is a need for more advanced techniques specific to 2D fracture trace data and which can use the combined geometric and topological structure.

From a geostatistical perspective, the concept of spatial variability describes how a measurable attribute varies across a spatial domain (Deutsch, 2002). Quantifying magnitude and directional dependence of the variability can also be done using geostatistical tools, provided there is a means to measure variability across multiple spatial samples. The variability in fracture data has typically been reduced to variability in attributes (such as fracture length by sampling area, number of intersections,

number of sets, and orientations), and attribute variability used to make decisions of *stationarity*. The identification of representative element volumes (REVs) then follows from the choice of stationarity. However, given that natural fracture networks display spatial heterogeneity, the suitability of such REVs based on stationarity assumptions needs to be re-examined. Therefore, it is interesting to compare network variation (rather than attribute variation) across the spatial domain. Any comparative method must retain topological and geometric structures encoded within the spatial samples.

## 2   Graph theory in fracture network analysis

### 2.1   Fracture networks as graphs

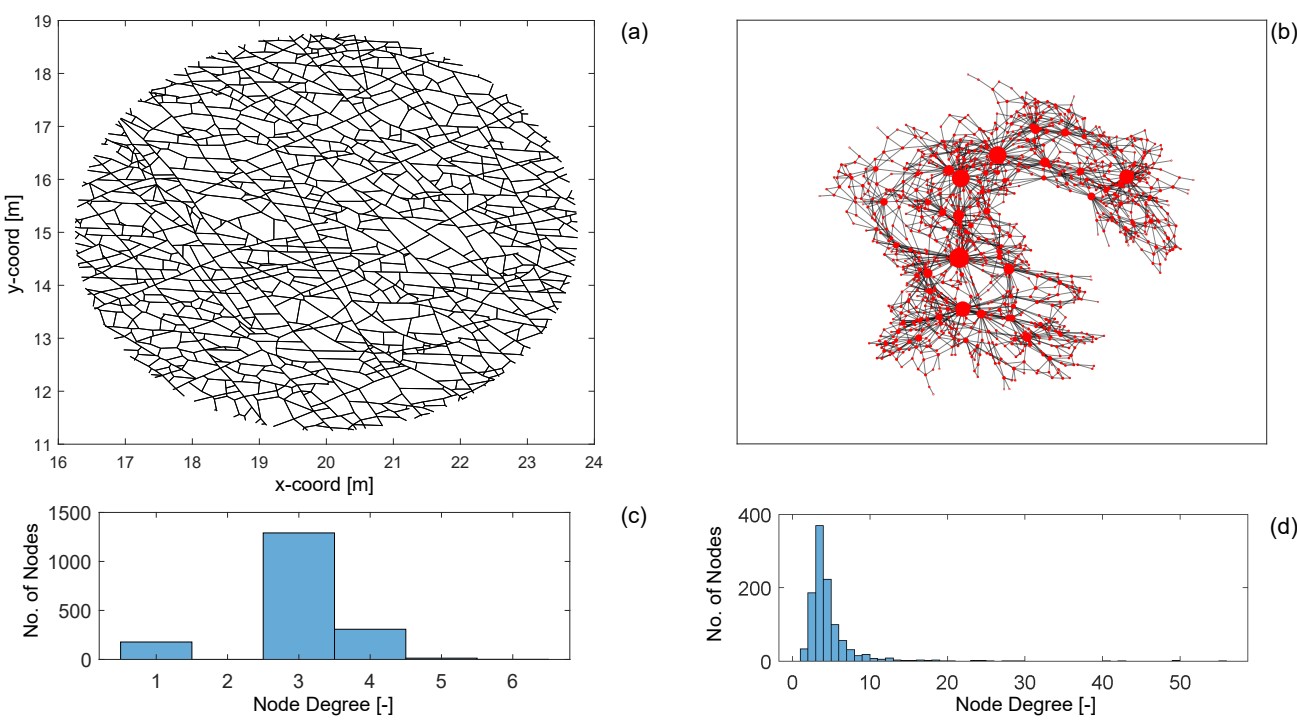

**Figure 1.** Comparing primal and dual forms of a fracture network from data published by Prabhakaran et al. (2021b) (a) a fracture network depicted in the primal form with dimensions in metres (b) corresponding dual representation of the fracture network with node sizing proportional to dual graph node degree and plotted using a force layout (c) node degree distribution of primal graph (d) node degree distribution of the dual graph

Many authors have suggested using graph theory for the characterization of fracture networks (such as Valentini et al., 2007; Andresen et al., 2013; Vevatne et al., 2014; Sanderson and Nixon, 2015; Sanderson et al., 2019). In graph theory and network science, graphs are structures that comprise a set of edges and vertices representing relationships between data. In fracture networks, the vertices are intersections between fractures, and the edges represented by fracture segments connecting the

vertices (Sanderson and Nixon, 2015). By assigning positional information to the vertices (also called nodes), fractures in the form of graphs encapsulate both topological and spatial information (Sanderson et al., 2019). An alternate graph representation is when fractures from tip-to-tip are vertices, and intersections with other fractures are edges. Barthelemy (2018) refers to these types of representations as to *primal* and *dual* forms, respectively. Others, such as Doolaeghe et al. (2020), call the two
representations as *intersection graphs* and *fracture graphs*.

We depict an example of a fracture network in its primal form (see Fig. 1.a) and in its dual form (see Fig. 1.b). The degree of a graph node is simply the number of edges that are incident at a particular node. As seen in the primal graph in Fig. 1(c), the maximum node degree is 6, with the most common degree value being 3. This type of degree distribution is typical for a spatial graph in which physical constraints limit the maximum possible node degree. We may note that node degrees in spatial graph
representations of fracture networks are most likely to be 1,3, or 4. For fracture networks interpreted from outcrop images as depicted in Fig. 1(a), eroded fractures and enlarged apertures may lead to higher degrees due to issues in resolving closely spaced nodes.

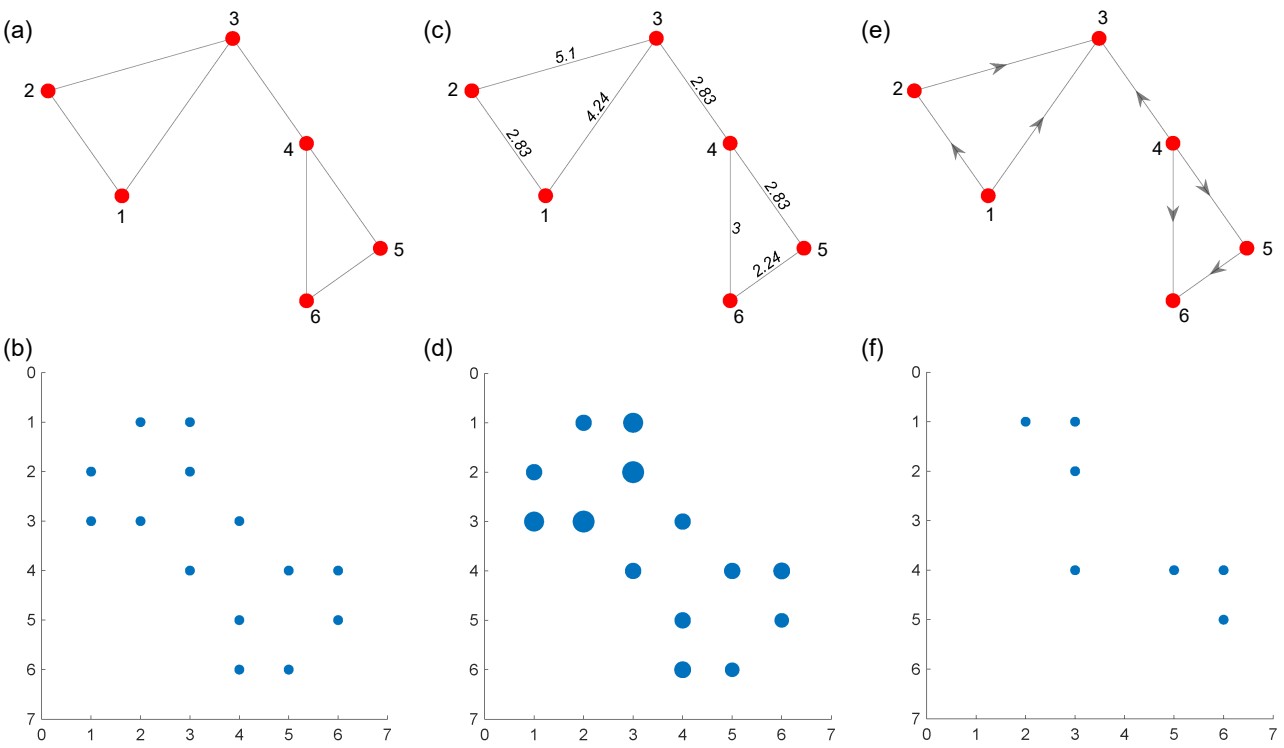

**Figure 2.** (a) An unweighted planar graph with six nodes and seven edges (b) adjacency matrix of unweighted graph (c) a weighted planar graph with edge weights proportional to euclidean distances between connecting nodes (d) weighted sparse adjacency matrix for weighted planar graph (e) a directed, unweighted graph (f) adjacency matrix of directed graph

In the case of the alternate representation, referred to as dual graphs by Barthelemy (2018) and depicted in Fig. 1(d), the maximum degree can be much higher, and the longest fractures that have the highest number of intersections also have the highest degree. Andresen et al. (2013) and Vevatne et al. (2014) suggested that fracture networks are *disassortative* in that shorter fractures preferentially attach on to the longer fractures. The property of disassortativity is quantitatively defined using assortativity coefficients (Newman, 2002) with disassortative networks having negative assortativity coefficients. Andresen et al. (2013) and Vevatne et al. (2014) report negative assortativity coefficients for fracture networks that are represented in the dual form. Prabhakaran et al. (2021b) found such a correlation between dual graph node degree and length.

In graph representations, weights can be assigned to edges that are proportional to the importance of that edge. In the case of fracture networks in the primal form, this can be the euclidean distance between the nodes (or fracture edge intersections). The weight may also be the direction cosine of the particular edge that indicates orientation. In the dual graph representation, intersections between fractures represent the edges. Therefore the edge weight may be specified in terms of intersection angle. Graphs may also be *directed* with a specific direction to edges. In the case of spatial graphs derived from fracture networks, an undirected but weighted representation is sufficient. Fig. 2(a), Fig. 2(c), and Fig. 2(e) depicts examples of unweighted, weighted, and directed planar graphs, respectively. The corresponding adjacency matrices are depicted in Fig. 2(b), Fig. 2(d), and Fig. 2(f).

## 2.2   Graph distance measures to quantify network similarity

Several graph similarity measures exist within the graph theory literature to compare graphs (see Hartle et al., 2020; Tantardini et al., 2019; Emmert-Streib et al., 2016 for recent reviews). Graph comparisons are a challenging, non-trivial problem in terms of computing complexity (Schieber et al., 2017). Still, various measures exist that can capture and highlight useful aspects of the graph structure that facilitate comparisons. Graph *isomorphism* between two graphs implies that there exists a series of necessary conditions such as an equal number of nodes, edges, degree sequences, and sufficient conditions such as equal adjacency matrices (Van Steen, 2010). An isomorphism test on two graphs $G_1$ and $G_2$, can only yield two results, either isomorphic or not. Graph *similarity* can therefore be differentiated from graph *isomorphism* in that the latter comparison can only return a binary outcome. Graph similarity on $G_1$ and $G_2$, on the other hand, returns a real-valued quantity that converges to zero when the two graphs approach isomorphism (or complete similarity).

Tantardini et al. (2019) classify distance measures based on whether the metric is capable of comparing graphs with an unequal number of nodes or not. The metrics may also be classified based on whether they can also handle weighted and directed graphs. Using a graph-similarity measure on a fracture network, we can explore spatial variations in network structure by comparing multiple sampling points.

## 2.3   Combining dissimilarity measures with clustering algorithms

Since we are interested in quantifying spatial variability, we may recast the problem as that of identifying clusters within the network. Clustering is also referred to as unsupervised classification and is a process of finding groups within a set of objects with an assigned measurement (Everitt et al., 2011). If we consider a dataset, $D = [X_1, X_2, ...X_n]$, containing 'n' data

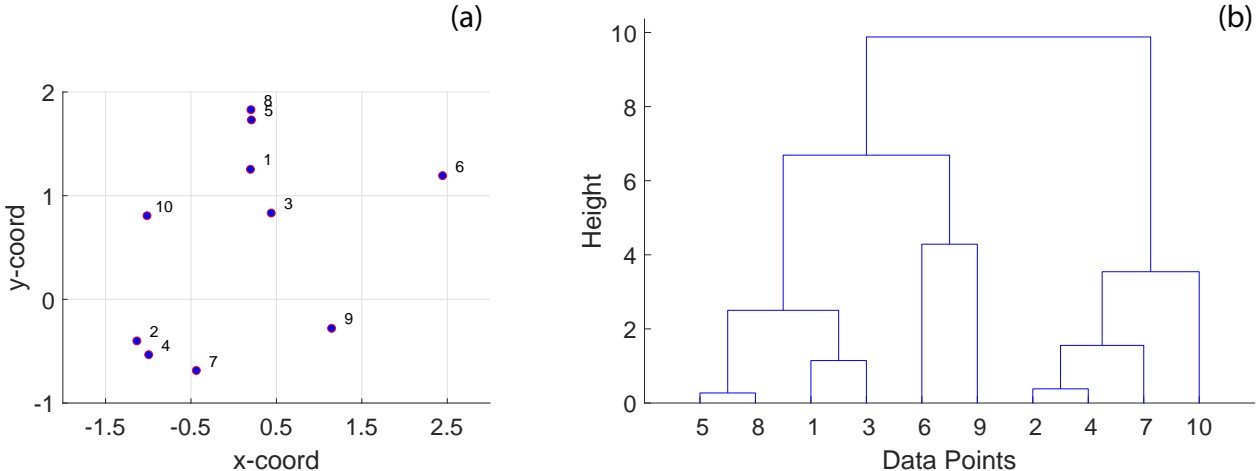

**Figure 3.** A simple example of hierarchical clustering using euclidean distance (a) 10 randomly positioned points in 2D space (b) dendrogram computed from hierarchical clustering using the euclidean distance depicting clusters of the 10 individual points at different levels organized into a hierarchy. The procedure of hierarchical clustering is shown in Algorithm 1.

samples, clustering then implies arranging the elements of $D$ into 'm' distinct subsets, $C = [C_1, C_2, ...C_m]$, where $m \leq n$. From a statistical perspective, the clustering task is different from classification because the former is *exploratory*, whereas the latter is *predictive*, although both attempt to assign labels. Therefore clustering must precede classification.

In the existing literature on fracture networks, assigning labels to specific perceived archetypal networks (or end-members) is
120 standard. These typologies include terms such as orthogonal, nested, ladder-like, conjugated, polygonal, corridors, etc. (Bruna et al., 2019a,b; Peacock et al., 2018). However, when faced with the reality of outcrop-derived 2D fracture trace data, it is not easy to assign such labels. Therefore, clustering is a significant and necessary step in exploratory fracture data analysis.

Hierarchical clustering (HC) is an unsupervised statistical clustering method (Kaufman, 1990) that can identify clusters within a set of observations given a distance matrix computed by applying a well-defined distance function, pair-wise on each
125 observation. In contrast to other clustering methods such as *k-means* or *k-medoids*, which require an a priori known number of clusters as input arguments, HC re-organizes observations into hierarchical representations from which the user can pick a level of granularity. At the lowest level, there is just one cluster containing all the observations. At the highest level, the number of clusters is equal to the observations. HC algorithms are referred to as *agglomerative* or *divisive* depending upon whether they begin from a lower level or from the highest level. The clustering then organises the discrete data into a hierarchical dendrogram
structure that positions the clusters based on the magnitude of similarity. By combining graph distance computations across spatially distinct samplings with unsupervised HC, cluster detection automatically leads to quantified spatial variation. A simple example of HC is illustrated on a set of randomly distributed points in space (see Fig. 3.a). The result is the hierarchical dendrogram structure depicted in Fig. 3(b).

## 3 Fracture Datasets

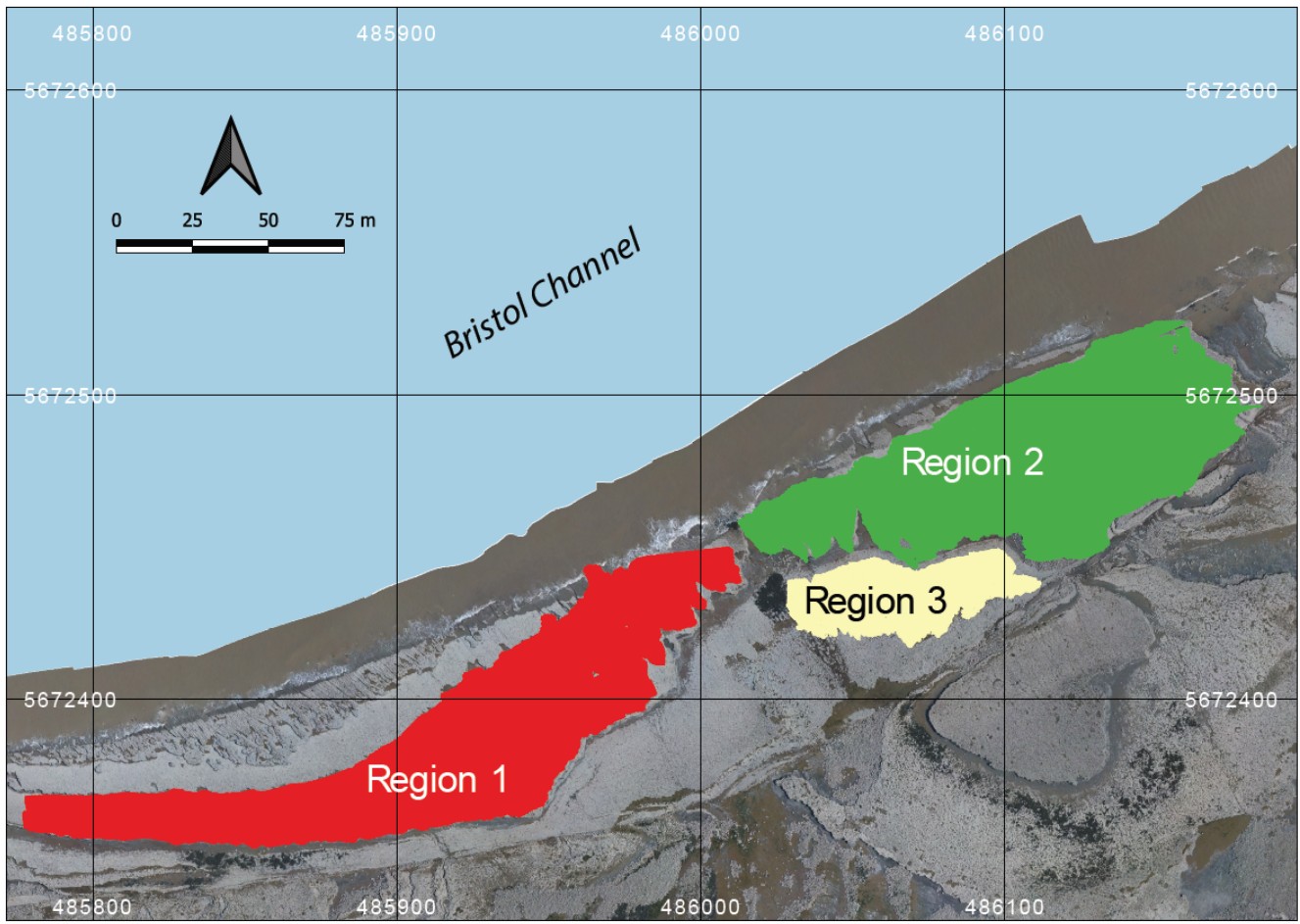

**Figure 4.** Overview of fracture networks corresponding to the three considered regions. This map is derived from an open image dataset published by Weismüller et al. (2020) and available for download with a CC-BY license

To validate the proposed approach based on graph distance metrics and hierarchical clustering, we utilize a 2D joint fracture dataset from the Lilstock pavement in the Bristol Channel, UK (Prabhakaran et al., 2021b). The dataset consists of fracture joints automatically traced using a technique described in Prabhakaran et al. (2019) from UAV photogrammetric data published by Weismüller et al. (2020). The joint networks correspond to Jurassic limestones with very dense joint networks spread across multiple layers. The joints are stratabound and perpendicular to bedding. We consider three large-scale fracture networks from
this dataset, as depicted in Fig. 4. There is considerable spatial variation in the jointing. From previous literature documenting joints within the Lilstock pavements, the spatial variation is attributed to multiple reasons.

The proposed explanations include proximity and influence of faults explained by fluid-driven radial-jointing emanating from asperities within fault (e.g., Rawnsley et al., 1998; Gillespie et al., 1993), spatial variation of thicknesses of intercalated limestone and shale layers (e.g., Belayneh, 2004), proximity to high-deformation features such as folding (e.g., Belayneh and Cosgrove, 2004), the interplay between regional and local stresses resulting in complex stress fields (e.g., Whitaker and Engelder, 2005), inheritance from the spatial distribution of pre-existing vein/stylolite networks that influenced later joint network development (e.g., Wyller, 2019; Dart et al., 1995), and synkinematic cementation in veins affecting later development of joints (Hooker and Katz, 2015). Recent work on fractures at the Kilve outcrop (Procter and Sanderson, 2018), exposing the same geological units as those considered in this work, conclude that anomalous fracture intensity exists in fracturing at various locations and suggest that variability in fracture intensity cannot be fully explained by variations in thickness, compositional, or textural variations.

**Table 1.** Summary statistics for the three regions

| Region | Approx. area (sq.m) | Fractures | Edges | Nodes |
|--------|---------------------|-----------|--------|--------|
| Region 1 | 6017 | 124006 | 364703 | 228661 |
| Region 2 | 6749 | 141344 | 365333 | 235089 |
| Region 3 | 1473 | 28892 | 78151 | 49771 |

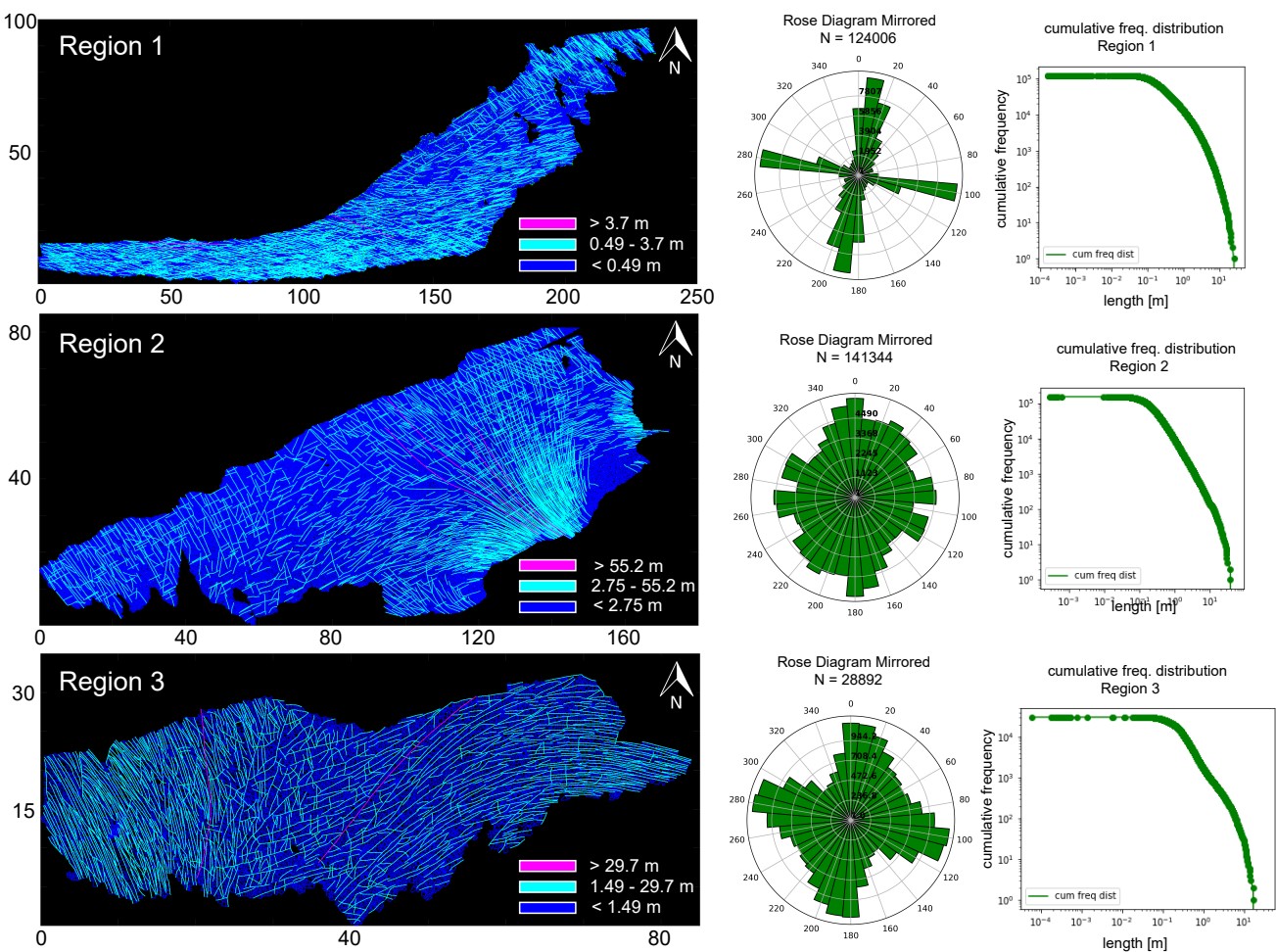

**Figure 5.** Comparison of the three regions in terms of networks, orientations and length distributions. Map dimensions are in metres. This image has been modified from Prabhakaran et al. (2021b) with permission

From this dataset, we utilize fracture networks corresponding to three contiguous regions. Figure. 4 depicts the three areas' spatial extent labelled as Regions 1 to 3. The intensity of fracturing is such that the spatial graphs corresponding to each region have a single connected component. Table 1 tabulates summary statistics for the three networks. The number of edges and nodes correspond to the primal graph representation. What is referred to as *fractures* in Table 1 are sequences of graph edges that are clubbed together based on continuity and a strike direction threshold (or number of dual graph nodes). Regions 1 and 2 correspond to a single stratigraphic layer but, owing to erosion, they are not contiguous within the outcrop. We treat them separately in our analysis of spatial variation.

The detailed resolution, topological accuracy, and spatial extent of the traced networks make the dataset appropriate for a detailed analysis of spatial variation in fracturing. The networks have significant intra- and inter-network variability in fracturing.

Figure. 5, Fig. 6, and Fig. 7 illustrate these differences. From Fig. 5, the fracture orientations of Region 1 depict discernable angular bins of fracture orientations. On the other hand, rose plots of Regions 2 and 3 show considerable scatter owing to the presence of long and curved fractures. Fracture length distributions are different, with Region 2 having the longest fractures and Region 1 the shortest. The distribution of joints within a particular length bin is also highly variable. In Fig. 6, the cumulative variation in the strike along individual fracture edges that comprise a tip-to-tip fracture is plotted as a function of the total length. The slope of the scatter plots give an indication of the fracture curvature. The slope of the scatter plot is higher in Regions 2 and 3 than in Region 1. We interpret the curvature to, therefore, be the least in Region 1.

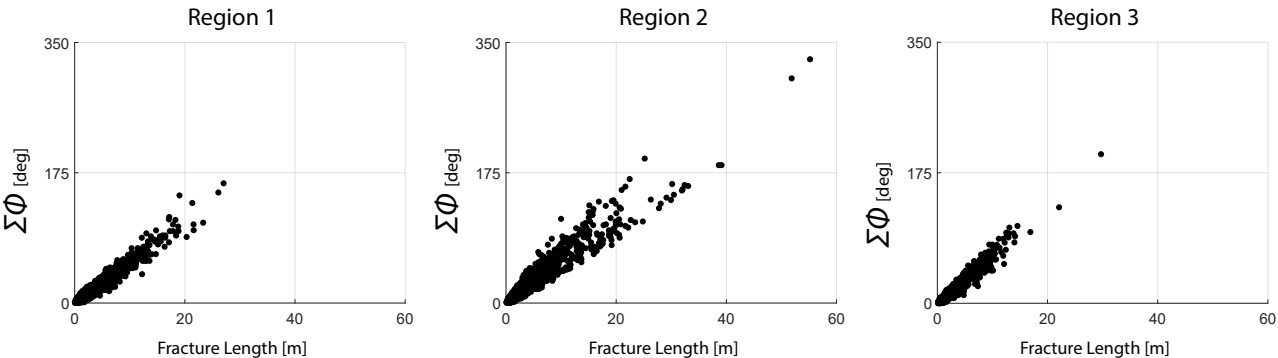

**Figure 6.** Correlation between sum of strike differences of fracture segments constituting tip-to-tip fractures versus total fracture length for the three regions

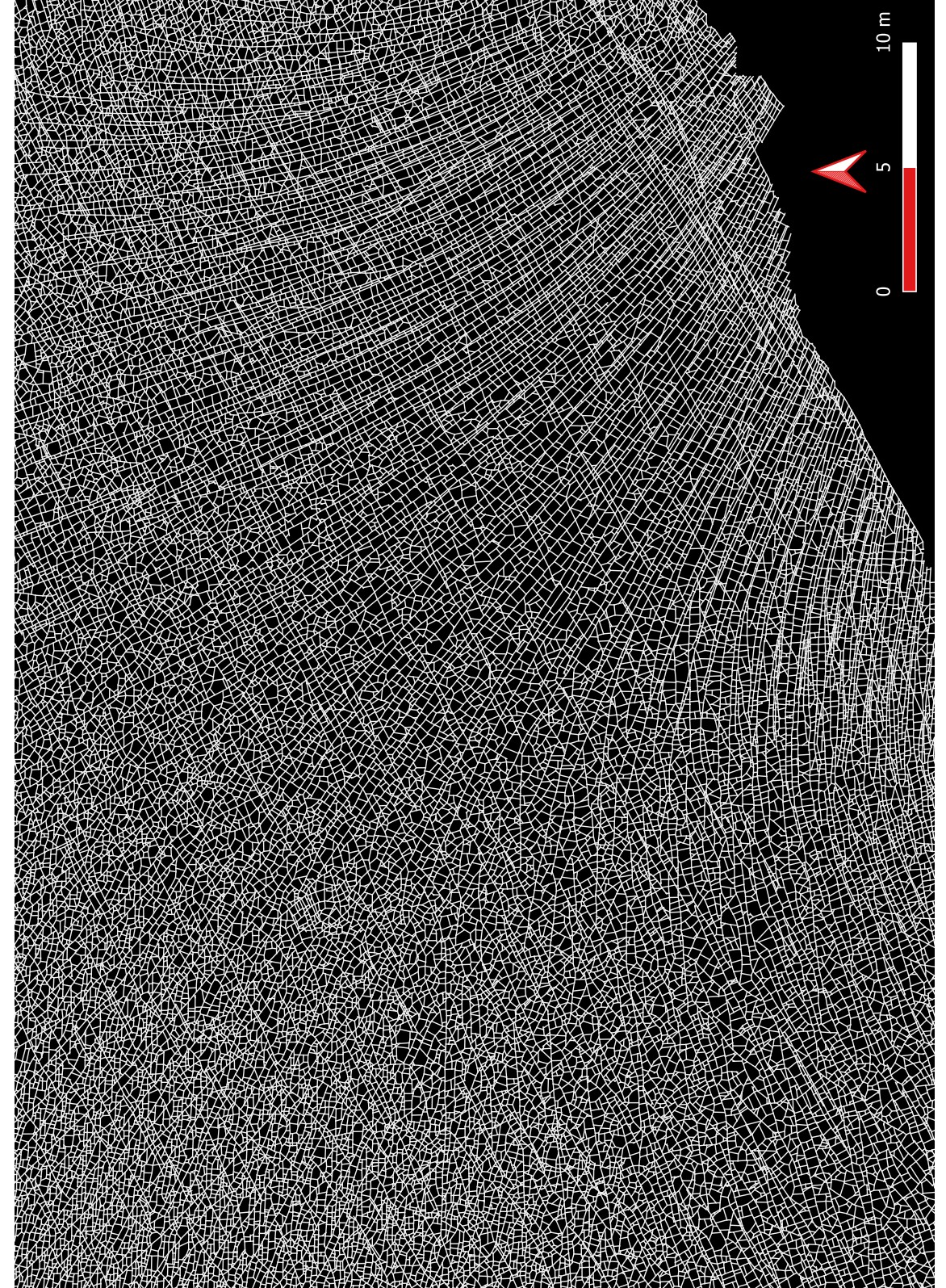

**Figure 7.** Cutout from Region 2 depicting the detailed resolution of the fracture dataset

# 4 Methods

## 4.1 Sub-sampling the network data

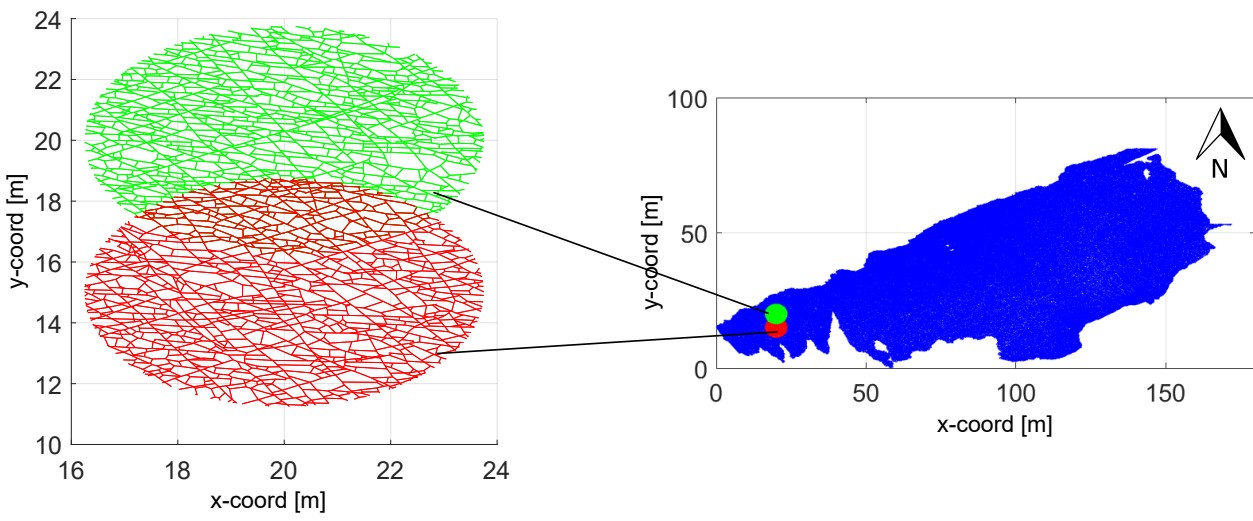

**Figure 8.** Sub-sampling of a fracture graph corresponding to full region into sub-graphs of 7.5 m diameter and spacing of 5 m

We circularly sample the fracture networks on a cartesian grid with a sub-graph extracted within a circular region centered at each grid point. The grid spacing-to-circle diameter is maintained such that neighboring sub-graphs share some portion of the area (see Fig. 8). Near the networks' boundaries, the sub-graphs are either too small or result in disconnected graph components. We neglect these samples so that they do not affect the clustering results. The process of circular sampling creates edge nodes with degree 1, which has the effect of altering node topology by introducing isolated, degree-1 nodes. To prevent this from impacting clustering results, we remove all edges from the sub-graphs emanating from degree-1 nodes that contact the periphery of the circular sample. This effect is illustrated in Fig. 9. Each sub-graph can now be compared to every other sub-graph using a graph distance metric to compute a pair-wise distance matrix. The distance matrix serves as the input to the hierarchical clustering algorithm.

**Table 2.** Number of subgraphs obtained per region

| Region | No. of sub-graphs |
|--------|-------------------|
| Region 1 | 219 |
| Region 2 | 212 |
| Region 3 | 117 |

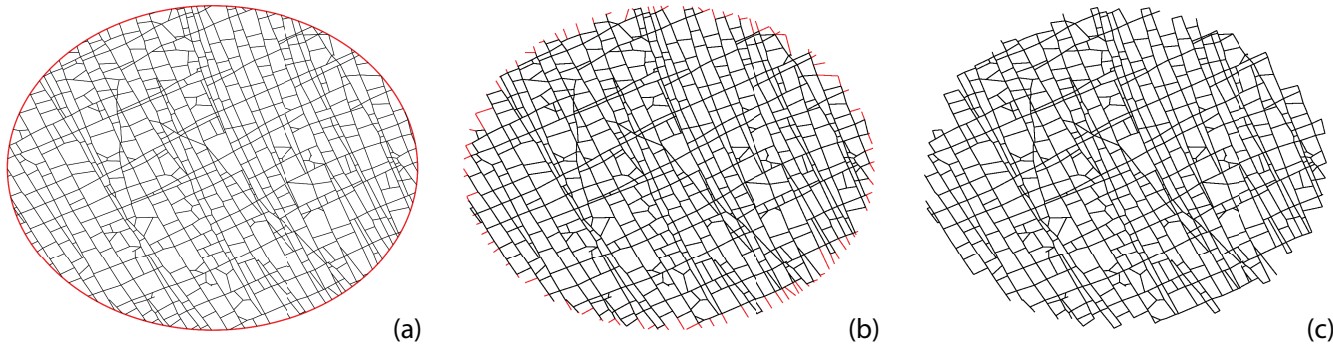

**Figure 9.** Treating isolated nodes and dangling edges that arise due to circular-sampling (a) circularly sampled subgraph with a diameter of 7.5 m (b) edges connected to isolated nodes intersected by circle (c) subgraph after removing isolated nodes and corresponding dangling edges

For $N$ sub-graphs, the number of comparisons necessary are $\frac{N(N-1)}{2}$. The computational complexity of graph comparison increases polynomially with the size of sub-graphs in terms of node sizes. Since the number of comparisons increases quadratically with the number of sub-graphs, we seek to balance grid spacing and sampling diameter. For Regions 1 and 2, we choose a spacing of 5 metres for circularly sampled subgraphs with a diameter of 7.5 m. For Region 3, which is also the smallest region, a spacing of 5 metres would lead to quite a smaller number of sub-graphs. Therefore, we use a more dense spacing of 3 metres with a diameter of 7.5 m. Table 2 tabulates the number of sub-graphs pertaining to each region.

## 4.2 Graph similarity measures

We use the following four graph similarity measures to compare the sub-graphs.

- Fingerprint Distance (Louf and Barthelemy, 2014)

- D-measure (Schieber et al., 2017)

- NetLSD (Tsitsulin et al., 2018)

- Portrait Divergence (Bagrow and Bollt, 2019)

The performance of these similarity measures have been validated previously by Hartle et al. (2020) and Tantardini et al. (2019) for a variety of benchmark graph datasets. Each similarity measure is described briefly in the following subsections. The reader is referred to the references above for further details on the similarity measures.

### 4.2.1 Fingerprint distance

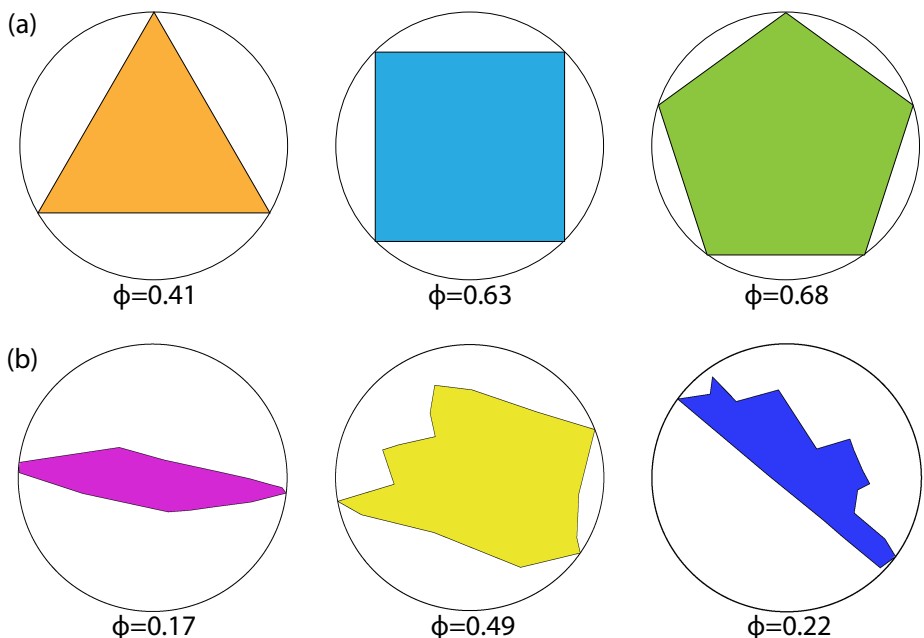

**Figure 10.** (a) shape factors for regular block shapes with equal edge lengths (b) shape factors for polygonal blocks resulting from real fracture networks in Region 1 (dimensions are relative)

The fingerprint distance introduced by Louf and Barthelemy (2014) is purely geometric and combines statistics of block faces and shape factors in computing a probability distribution of a spatial graph. Louf and Barthelemy (2014) formulated the measure in the context of quantifying differences in street patterns. A *block* denotes the 2D region enclosed by graph edges. For any given spatial graph, this corresponds to the number of bounded sub-graphs or primary cycles. We neglect isolated fractures and those having dead ends when computing these blocks. Given the network intensity in our dataset, such isolated fractures

are minimal. Every block has an associated *shape factor*, '$\phi$' which is expressed in terms of block area '$A$' and circumscribing circle area, '$A_c$',

$$\phi = \frac{A}{A_c} \tag{1}$$

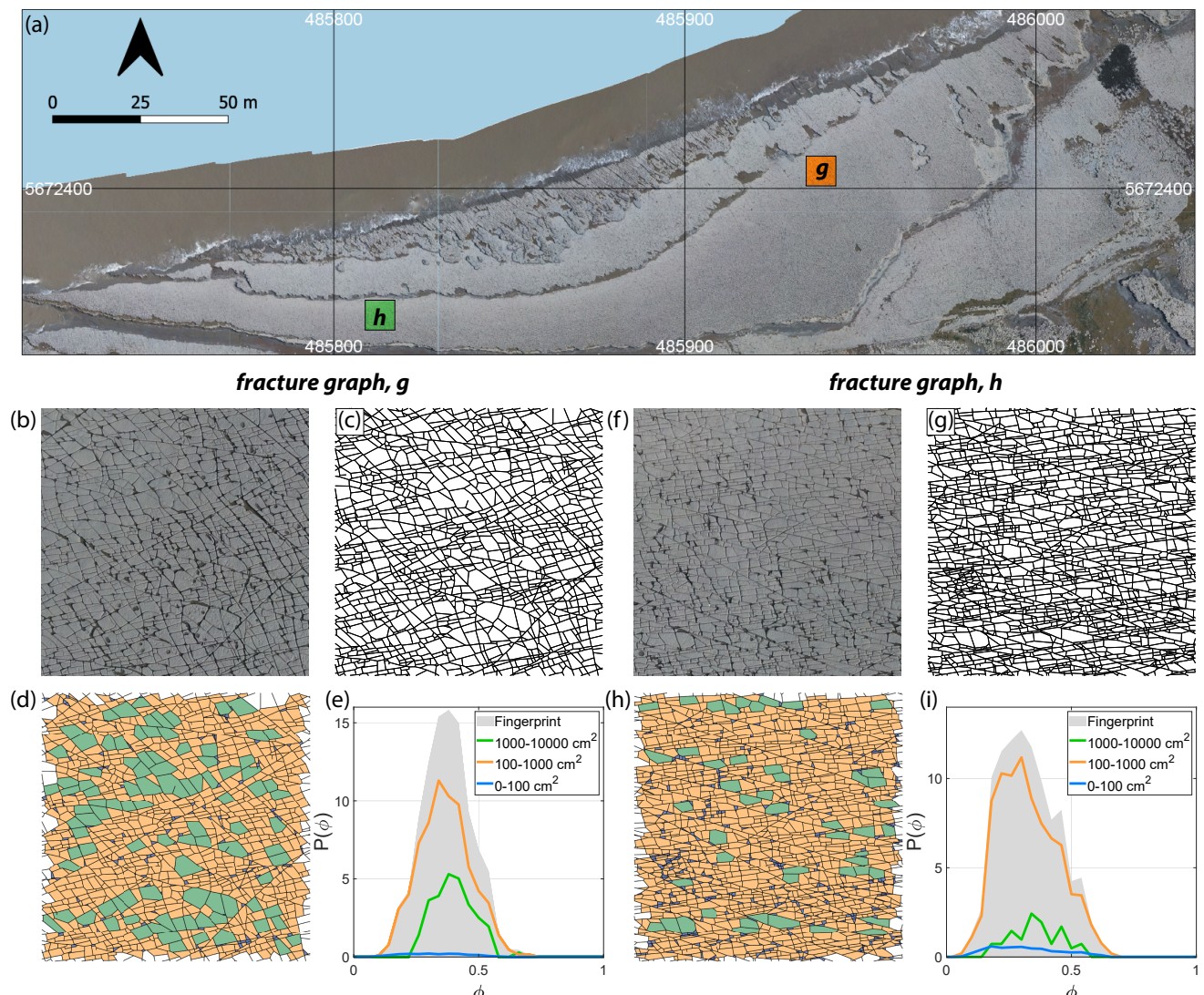

**Figure 11.** (a) Overview of Region 1 with two selected 1000 x 1000 pixel image tiles (b) enlarged view of first image tile (c) fracture network corresponding to first tile as a spatial graph with dimensions of 8.6 m x 6.75 m and having 3583 edges and 2382 nodes (d) block face areas coloured as per three area bins, 0-100 $cm^2$, 100-1000 $cm^2$, and 1000-10000 $cm^2$ (e) $P(\phi)$ or fingerprint of the sub-graph depicting the combined effects of area and shape factor, $\phi$ pertaining to the three area bins. (f) enlarged view of second image tile (g) fracture network corresponding to second image tile as a spatial graph with 5418 edges and 3539 nodes (h) block face areas binned logarithmically (i) fingerprint of second spatial graph. (a),(b), and (f) are derived from images contained in the open dataset (CC-BY license) published by Weismüller et al. (2020)

The value of $\phi$ is always smaller than 1, with larger values meaning that the block face shape is closer to that of a regular polygon. Figure. 10(a) depicts shape factors of regular polygons versus that of polygons derived from spatial networks in Fig.

10(b). No unique correspondence exists between a particular shape and a magnitude of $\phi$; however, the overall distribution of $\phi$ indicates reveals block shape distribution patterns and highlights differences between spatial graphs. The shape factor alone does not fully serve as a similarity measure as blocks can have similar shapes but different face areas. The distribution of the block-face areas is binned logarithmically to integrate information from the shape factor and block area distributions. A conditional probability distribution, $P(\phi|A)P(A)$, is then defined representing the contribution of $P(\phi)$ for each area bin and the summation of which yields the fingerprint curve, $P(\phi)$,

$$P(\phi) = \sum_A P(\phi|A)P(A). \tag{2}$$

An example of a *fingerprint*, so named by Louf and Barthelemy (2014), is depicted in Fig. 11(e) and Fig. 11(j), with the distribution curves for three area bins, for two fracture networks derived from image tiles (see Figs. 11.b,c,f,g) corresponding to Region 1 (Fig. 11.a). The curves in Fig. 11(e) and Fig. 11(j) encapsulates information based on shape factors and block areas (see Figs. 11.d,h), including the proportional contribution from all logarithmic area bins considered.

Denoting $f_\alpha(\phi)$ as the ratio of the number of faces with a shape factor '$\phi$' that lie in a bin '$\alpha$' over the total number of faces for that graph, a distance $d_\alpha$ between two graphs $G_a$ and $G_b$ is computed by integrating over $f_\alpha(\phi)$ for the two different graphs. The distance based on $f_\alpha(\phi)$ of the two graphs for a single area bin is defined as:

$$d_\alpha(G_a, G_b) = \int_0^1 |f_\alpha^a(\phi) - f_\alpha^b(\phi)|^n d\phi \tag{3}$$

As per Louf and Barthelemy (2014), the value of $n$ can either be 1 or 2. We choose $n = 1$ in our computation. The global fingerprint distance $D_{FP}$ between $G_a$ and $G_b$ can then be computed summing over all area bins $\alpha$,

$$D_{FP}(G_a, G_b) = \sum_\alpha d_\alpha(G_a, G_b)^2 \tag{4}$$

We have attached our MATLAB implementation of the fingerprint distance in the code supplement. We computed the distance matrix for all sub-graphs corresponding to the three regions using this implementation.

### 4.2.2 D-measure

The D-measure introduced by Schieber et al. (2017) is a three-component distance metric with weighting constants for each component. The three properties of graphs compared are the network node dispersion (NND), node distance distribution ($\mu$), and the alpha centrality ($\alpha$). The dissimilarity measure, $D_{DM}$, is the weighted sum:

$$D_{DM}(g,h) = w_1 \sqrt{\frac{\mathcal{J}(\mu_g, \mu_h)}{log2}} + w_2 \left| \sqrt{NND(g)} - \sqrt{NND(h)} \right| + \frac{w_3}{2} \left( \sqrt{\frac{\mathcal{J}(P_\alpha(g), P_\alpha(h))}{log2}} + \sqrt{\frac{\mathcal{J}(P_\alpha(g^c), P_\alpha(h^{c'}))}{log2}} \right),$$

where $\mathcal{J}$ indicates the Jensen-Shannon divergence. The constants $w_1$, $w_2$, and $w_3$ in Eq.5 are real and non-negative weights such that $w_1 + w_3 + w_3 = 1$.

As per Schieber et al. (2017) the first term in Eq.5 compares averaged connectivity node's patterns as per node distance distribution. Schieber et al. (2017) define NND, within the second term, as a measure of the heterogeneity of a graph with respect to connectivity distances that capture global topological differences. The NND is computed as:

$$NND(G) = \frac{\mathcal{J}(P_1,....,P_N)}{log(d+1)},$$

(6)

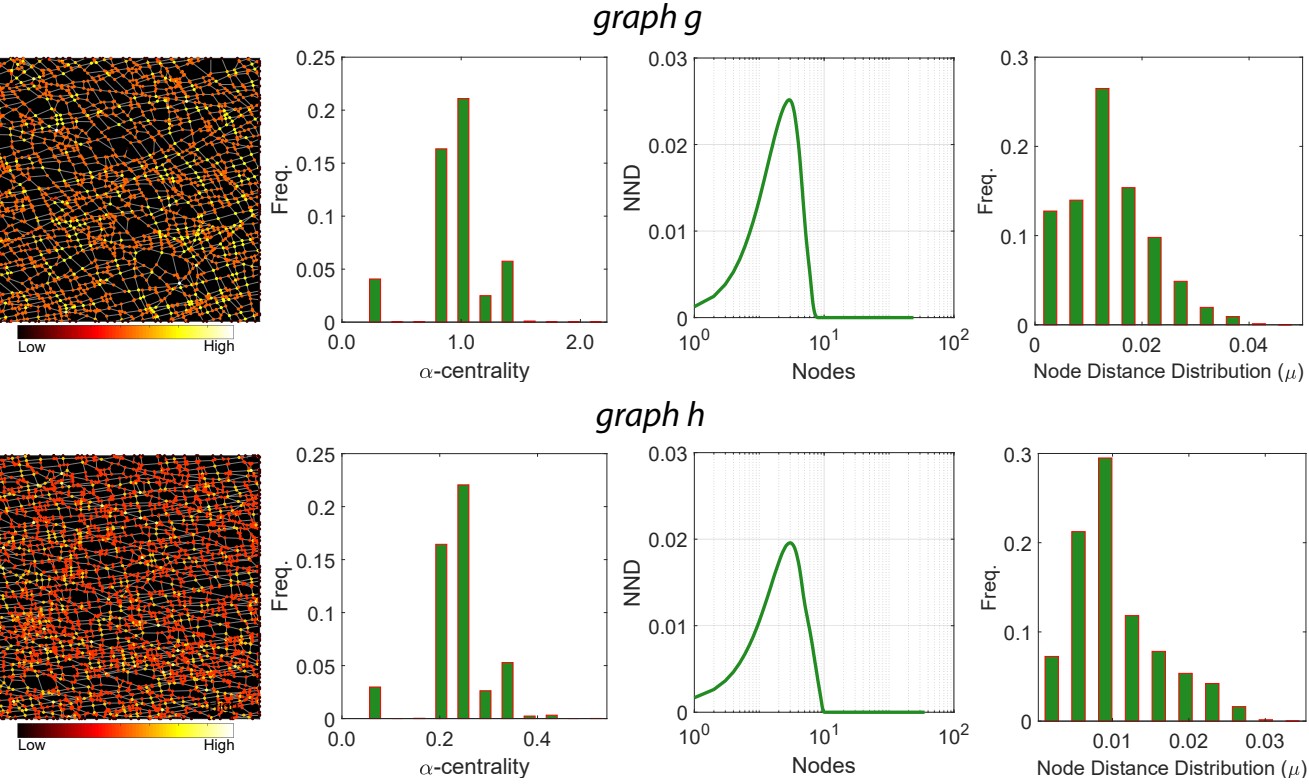

**Figure 12.** D-measure components for the two example fracture graphs comparing $\alpha$-centrality of nodes, distributions of $\alpha$-centrality, NND distributions, and node distance distributions

where the numerator in Eq.6 is the Jensen Shannon divergence of $N$ connectivity distance distributions $[P_1, P_2.....P_N]$. $P_i$ is constructed as $P_i = p_i(j)$ where $p_i(j)$ is the fraction of nodes connected to node $i$ at distance $j$. The Jensen-Shannon divergence of $[P_1, P_2.....P_N]$ is expressed as:

$$\mathcal{J}(P_1,....,P_N) = \frac{1}{N} \sum_{i,j} p_i \log \left( \frac{p_i(j)}{\mu_j} \right). \tag{7}$$

$\mu_j$ in Eq.7 is the average of $N$ distributions and can be written as,

$$\mu_j = \frac{1}{N} \sum_{i=1}^{N} p_i(j). \tag{8}$$

The third term in Eq.5 is based on probability density functions associated with alpha centrality of graph $P_\alpha(g)$ and alpha centrality of the graph complement $P_\alpha(g^c)$. The value of weights was suggested by Schieber et al. (2017) as $w_1 = w_2 = 0.45$ and $w_3 = 0.1$. We use the implementation provided by Schieber et al. (2017) with these sets of weights to build the distance matrices for all sub-graphs within the three regions of interest. We depict in Fig. 12 for the two example fracture networks, the three properties that are used in computing the D-measure.

### 4.2.3 Portrait Divergence

The Portrait Divergence similarity score derives from *network portraits* introduced by Bagrow et al. (2008) for unweighted graphs and extended to weighted graphs by Bagrow and Bollt (2019). For a graph $g$ with $N$ nodes, the network portrait is defined as a matrix $B_{lk}$ where each entry is the number of nodes with $k$ nodes at $l$ distance. The limits of $l$ and $k$ are $0 \leq l \leq d$ and $0 \leq k \leq N - 1$, with $d$ being the diameter of the graph. The row entries of the network matrix $B_{lk}$ are probability distributions of a random node having $k$ nodes at a distance $l$:

$$P(k|l) = \frac{B_{lk}}{N} \tag{9}$$

For a second graph $h$, if the network matrix is $B'_{lk}$ with a corresponding probability distribution of $Q(k|l)$ and diameter $d'$, the Kullback Leibler (KL) divergence between $P(k|l)$ and $Q(k|l)$ is expressed as:

$$KL(P(k|l)||P(k|l)) = \sum_{l=0}^{max(d,d')} \sum_{k=0}^{N} P(k,l) log \frac{P(k,l)}{Q(k,l)} \tag{10}$$

The portrait divergence $D_{PD}(g,h)$ is computed by the Jensen Shannon divergence between $P(k|l)$ and $Q(k|l)$:

$$D_{PD}(g,h) = JSD(P(k|l), Q(k|l). \tag{11}$$

This can be expressed in terms of Kullback Leibler divergences and mixture distributions as:

$$D_{PD}(g,h) = \frac{1}{2}(KL(P||M) + KL(Q||M)) \tag{12}$$

where the mixture distribution $M$ of $P(k|l)$ and $Q(k|l)$ is given by:

$$M = \frac{1}{2}(P(k|l) + Q(k|l)) \tag{13}$$

The portrait divergence measure provides a single value $0 \leq D_{PD}(g,h) \leq 1$ for any pair of graphs. Bagrow and Bollt (2019) applied the portrait divergence measure to both synthetic and real world networks. The code implementation of portrait divergence attached with Bagrow and Bollt (2019) is used to construct the distance matrices for all sub-graphs within the three regions of interest. The network portrait or the $B_{lk}$ matrix for the example fracture graphs are depicted as heatmaps in Fig. 13.

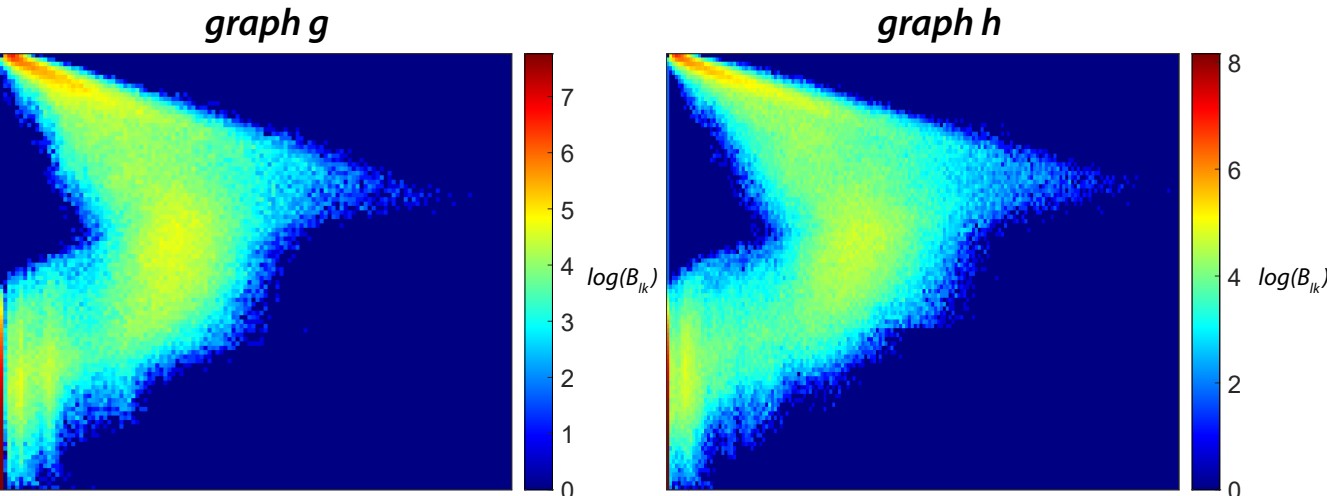

**Figure 13.** Heatmap representations of network portrait sparse matrices ($B_{lk}$) for the two example fracture graphs

### 4.2.4 Laplacian Spectral Descriptor

The NetLSD distance was introduced by Tsitsulin et al. (2018). It is based on a Frobenius norm computed between heat trace signatures of normalized Laplacian matrices of two graphs. For a graph $g$ with a normalized Laplacians $L$ and $n$ nodes, a heat kernel matrix is defined as:

$$H_t = e^{-tL} = \sum_{j=1}^{n} e^{-t\lambda_j} \phi_j \phi_j^T \tag{14}$$

Using the heat kernel matrix $H_t$, a heat trace $h_t$ is defined as:

$$h_t = \sum_{j=1}^{n} e^{-t\lambda_j} \tag{15}$$

For a second graph $g'$ with a heat trace signature of $h_t'$, the NetLSD distance $D_{LSD}$ is then the Frobenius norm of the two heat signatures as:

$$D_{LSD} = ||h_t, h_t'||_{Frobenius} \tag{16}$$

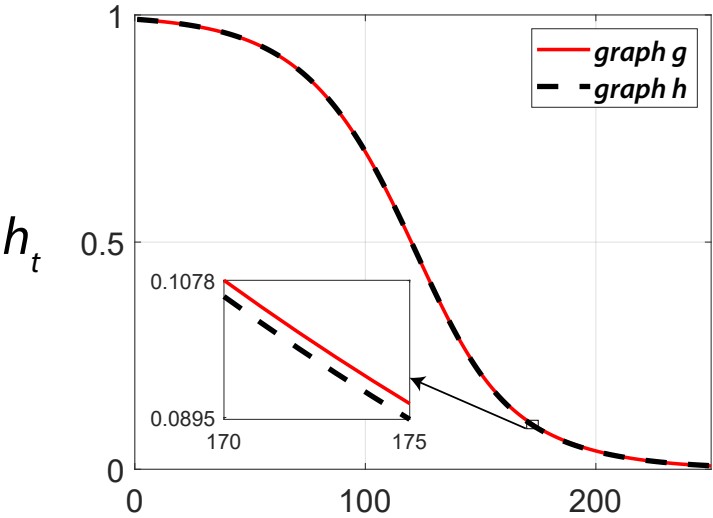

**Figure 14.** Comparing heat trace signature vectors for the two example fracture graphs computed using NetLSD

Figure. 14 depicts heat trace signatures computed using the NetLSD python package implemented by Tsitsulin et al. (2018) for the two example fracture graphs. We use this package to populate the distance matrices associated with sub-graphs from each region.

The values of graph similarity computed using the four metrics described by Equations. (4), (5), (12), and (16) for the two example fracture graphs depicted in Fig. 11(c) and Fig. 11(g) are summarized in Table. 3.

**Table 3.** Summary of graph similarities computed for example fracture networks

| Graph Similarity | Value |
| --- | --- |
| Fingerprint Distance [$D_{FP}$] | 0.1414 |
| D-measure [$D_{DM}$] | 0.1244 |
| Portrait Divergence [$D_{PD}$] | 0.2926 |
| NetLSD [$D_{LSD}$] | 0.0147 |

## 4.3   Hierarchical Clustering

After sub-sampling the fracture networks (see Section 4.1) and using the graph distance metrics described in Section 4.2 to construct distance matrices, we apply hierarchical clustering. HC can be done in an agglomerative versus divisive manner

(Hennig et al., 2016). We utilize the agglomerative approach, which generally follows the steps described in Algorithm 1.

---
**Algorithm 1** Agglomerative Clustering
---

    **Input:** Data $D = [X_1, X_2, .... X_n]$

    **Output:** Dendrogram $C = [C_1, C_2 ... C_m]$

    (i). *Initialization.* $m$ clusters of one element each with pair-wise distances computed and stored in symmetric square distance matrix $D_{dist}$

    (ii). form pair $C_i$ and $C_j$ that are closest within $C$

    (iii). form cluster $C_k = C_i \cup C_j$ and generate a new dendrogram node

    (iv). update $D_{dist}$ after computing distance between $C_k$ and $C - C_k$

    (v). delete rows and columns corresponding to $C_i$, $C_j$ from $D_{dist}$ and add rows and columns pertaining to $C_k$

    (vi). repeat (i) - (v) till only a single cluster remains
---

Based on how linking of clusters is done as per Algorithm 1(iii), HC can be classified into methods such as single linkage, complete linkage, unweighted pair-group average, weighted pair-group average, unweighted pair-group centroid, weighted pair group centroid, and Ward's method (Wierzchoń and Kłopotek, 2018). Ward's method performs the linkage by minimizing the sum-of-squares of distances between objects and cluster centres. We use Ward's method implemented within the R statistical programming environment to apply the HC to all the sub-graph distance data.

## 5 Results

We first show region-wise results of graph property computations. Intra-region spatial clustering resulting from the combined application of graph similarity measures with HC is then discussed. We use the following abbreviations for brevity throughout the section: FP - fingerprint distance, DM - D-measure, LSD - NetLSD, PD - portrait divergence.

### 5.1 Region-wise graph characteristics

Fingerprints pertaining to the regions is depicted in Fig. 15(a). The peak of the fingerprint plot is highest at a shape factor of 0.4 for Region 1 and increases to above 0.5 for Regions 2 and 3. Histograms in Fig. 15(a) depict the number of polygons within each area bin pertaining to fracture networks in each region. The network portraits or $B_{lk}$ matrices of each sub-graph within the three regions are combined to create ensemble region-wise network portraits depicted as heatmaps in Fig. 15(b). The non-zero entries in the $B_{lk}$ matrices, indicated by warmer colours in the heatmaps, have visibly different patterns. Heat traces for the sub-graphs in each region are shown in Fig. 15(c). Figure. 16 depicts the variation of the network properties that are components of the D-measure distance i.e., $\alpha$- centrality, NND, and $\mu$ for sub-graphs for the three regions.

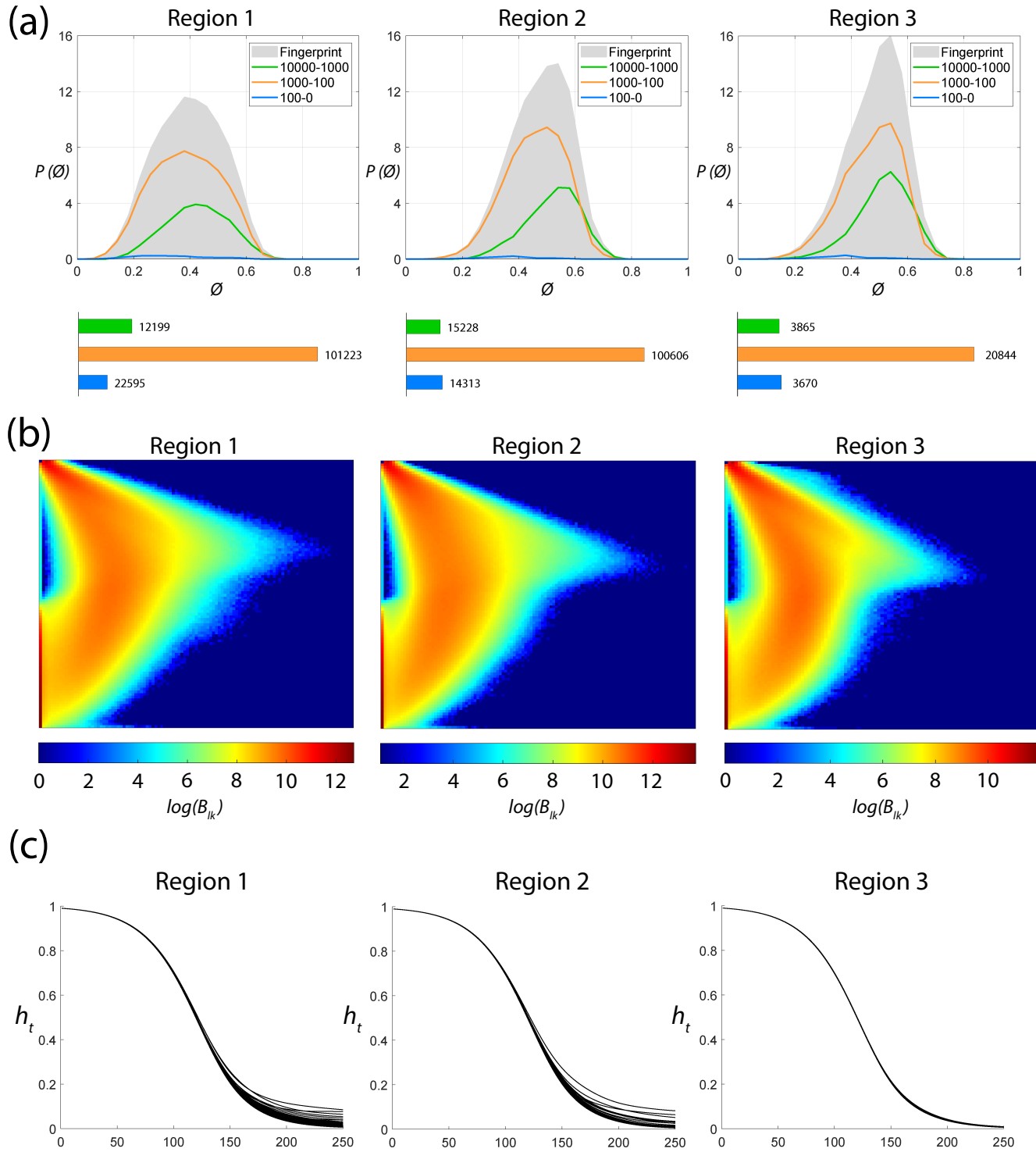

**Figure 15.** Region-wise graph properties (a) Fingerprints (b) Network Portrait Ensembles (c) Heat trace vectors

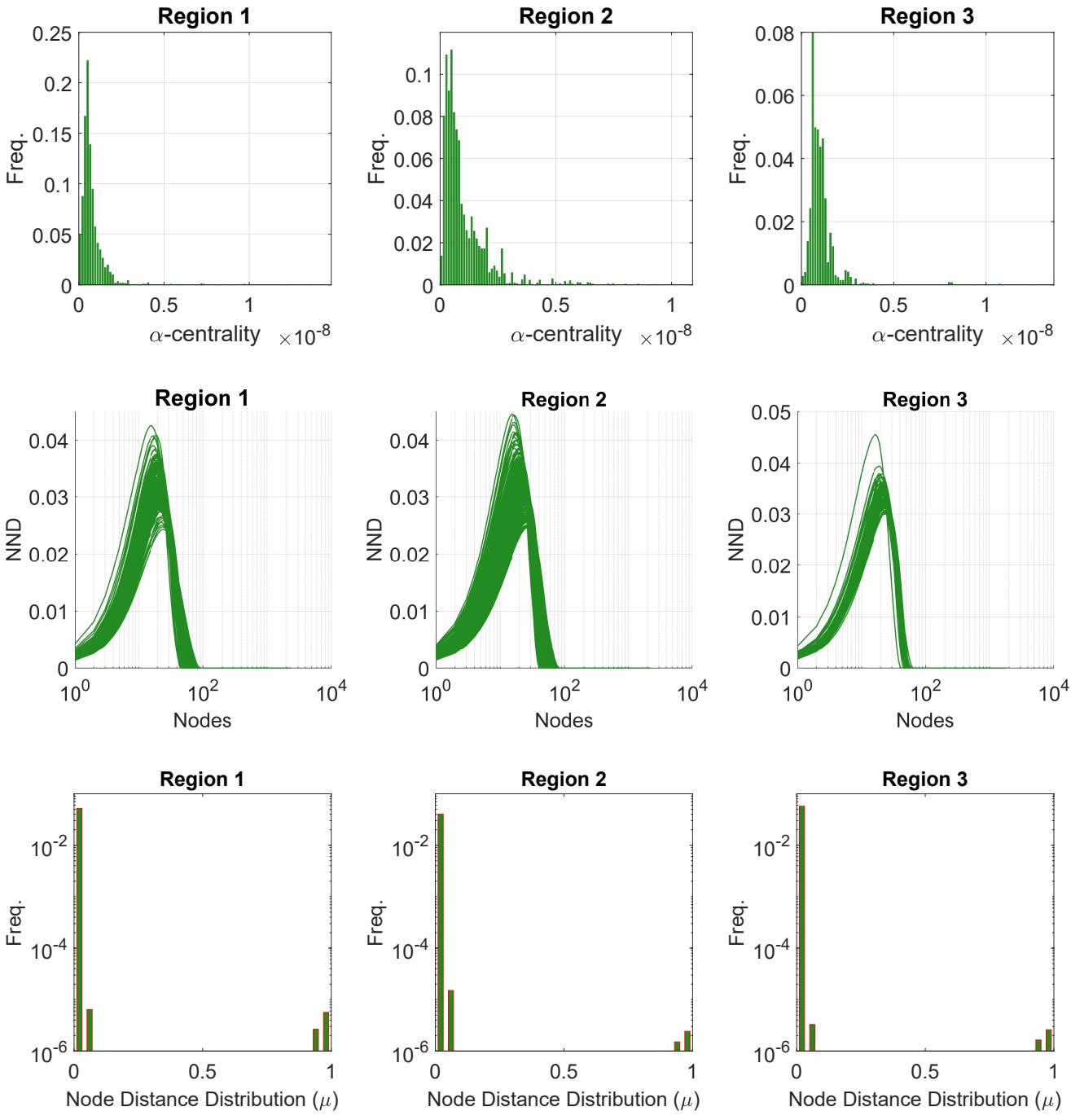

**Figure 16.** Region-wise properties used to compute the D-measure represented as ensemble plots of $\alpha$-centrality, network node dispersion (NND), and node distance ($\mu$) distributions for subgraphs

## 5.2 Intra-region spatial variation

Intra-region spatial variation results can be presented as distance matrix heatmaps corresponding to each graph similarity metric. Dendrograms depict the hierarchical organization of the sub-graphs corresponding to similarity entries within the distance matrix entries. The intra-regional variation is more intuitively illustrated spatially by showing sub-graphs using an appropriate colour scheme that groups similar clusters under colours picked within a linear spectrum. This section presents the clustering results for all three regions using a combination of dendrograms, spatial cluster maps, and heatmaps.

### 5.2.1 Analysis of spatial variation in Region 1

The spatial distribution of clusters pertaining to the four distance metrics overlain over the network is shown in Fig. 17(a)-(d) along with the associated dendrograms for the top 10 clusters. The sub-graphs are represented by coloured discs that follow a diverging colour scheme. The number of sub-graphs within each of the top 10 clusters is also listed under the dendrogram branches. It may be noted that the top 10 clusters are shown to depict, analyse, and compare the spatial variation across distance measures. A complete, uncut dendrogram and associated heatmaps of the similarity measures are depicted in Appendix Fig. A1. We can cut the dendrogram at different heights guided by slope changes in the weighted sum of squares plots shown in Fig. A1. The boundaries of spatial clusters vary with the dendrogram cut height, with sub-regions emerging by traversing deeper into the dendrogram. This variation is depicted in Appendix Figs. B1 - B4 for a range of clusters varying from 4-10. The number of sub-samples for the four similarity measures pertaining to a dendrogram cut of $k = 10$ is tabulated in Table. 4.

**Table 4.** Summary of sub-graphs within each cluster of Region 1 for k = 10

| Metric ↓ | cluster 1 | cluster 2 | cluster 3 | cluster 4 | cluster 5 | cluster 6 | cluster 7 | cluster 8 | cluster 9 | cluster 10 |
|---|---|---|---|---|---|---|---|---|---|---|
| FP | 3 | 36 | 24 | 47 | 2 | 24 | 41 | 16 | 20 | 6 |
| DM | 24 | 25 | 5 | 15 | 17 | 19 | 40 | 39 | 10 | 25 |
| LSD | 12 | 17 | 21 | 23 | 16 | 28 | 11 | 30 | 13 | 48 |
| PD | 38 | 17 | 3 | 5 | 24 | 25 | 79 | 13 | 6 | 9 |
| Total | | | | | 219 | | | | | |

We can observe that spatial autocorrelation exists for the FP (Fig. 17.a), DM (Fig. 17.b), and PD (Fig. 17.d) similarity measures. The LSD yields a speckled pattern with no obvious spatial autocorrelation (Fig. 17.c). In order to compare clustering results derived from the graph similarity measures, the spatial fracture persistence $P_{20}$ and $P_{21}$ computed using box-counting (box size of $0.5 \times 0.5$ m) is depicted in Fig. 17(e) and Fig. 17(f), respectively. Comparing clusters derived from graph similarity measures to the fracture persistence plots reveals boundaries within the network that are not easily discernable from the latter. Since LSD does not show spatial autocorrelation, we do not analyse it further.

Fig. 18(a)-(c) depicts topology histograms and rose plots of the clusters pertaining to the remaining three similarity measures. The orientation rose plots and topological summaries are generated by combining all circular samples identified under a cluster into ten clusters sub-graphs from the larger region fracture graph. It can be observed from the rose plots that the clusters have

varying fracture orientations that transitions across the hierarchy identified by the dendrograms. The topological summaries of the clusters do not vary significantly. Appendix Figs. C1-C3 depicts zoomed-in sub-graphs corresponding to each of the top 10

clusters that visually confirm the intra-regional variation.

We briefly describe the characteristics of the clustering results prefixing '$n$' to the number of subsamples within a cluster to refer to a particular cluster at a k=10 dendrogram cut. From Fig. 17(a) and the zoomed-in archetypal examples in Fig. C1, the clustering derived from FP seems to have a N-S variation trend. The trend is corroborated by observing the dendrogram, which splits into a northern branch comprising of clusters $n36$, $n24^b$, $n47$ and a southern branch with clusters $n6$, $n20$, $n41$, $n16$, $n2$,

$n24^a$. An outlier branch $n3$ exists at the boundary between northern and southern branches.

A similar variation is observable from the result of DM (see Fig. 17.b). However, the cluster demarcations are less stark than with FP with a notable stippled pattern. A major dendrogram division is a branch consisting of a thin sliver in the N-E (clusters $n24$, $n25^b$, $n5$, $n15$) which also include some boundary periphery samplings in the wast and south of Region 1. The southwestern sliver is mainly contained in a branch containing cluster $n17$. The central parts of Region 1 fall under the dendrogram

branch containing clusters $n39$, $n10$, $n25^a$. The remainder of the Region 1 is covered by branch containing clusters $n19$ & $n40$. Fig. C2 depicts archetypal examples of sub-graphs relevant to each cluster for DM.

The results of PD also depict N-S variation (see Fig. 17.d) in the clustering. Similar to DM, PD is also sensitive to the sub-graph completeness with peripheral clusters represented under $n13$. The branches comprising $n13$, $n6$, $n9$, and $n38$, $n17$ closely correspond to the trend of high fracture persistence (compare with Fig. 17.e and Fig. 17.f). Similar to results from FP

and DM, the thin sliver in the N-E of Region 1 is captured under the branch with clusters $n3$, $n5$, $n24$. The remainder of Region 1 falls under clusters $n25$, $n79$. Fig. C3 depicts archetypal examples of sub-graphs corresponding to each cluster for PD.

# Region 1

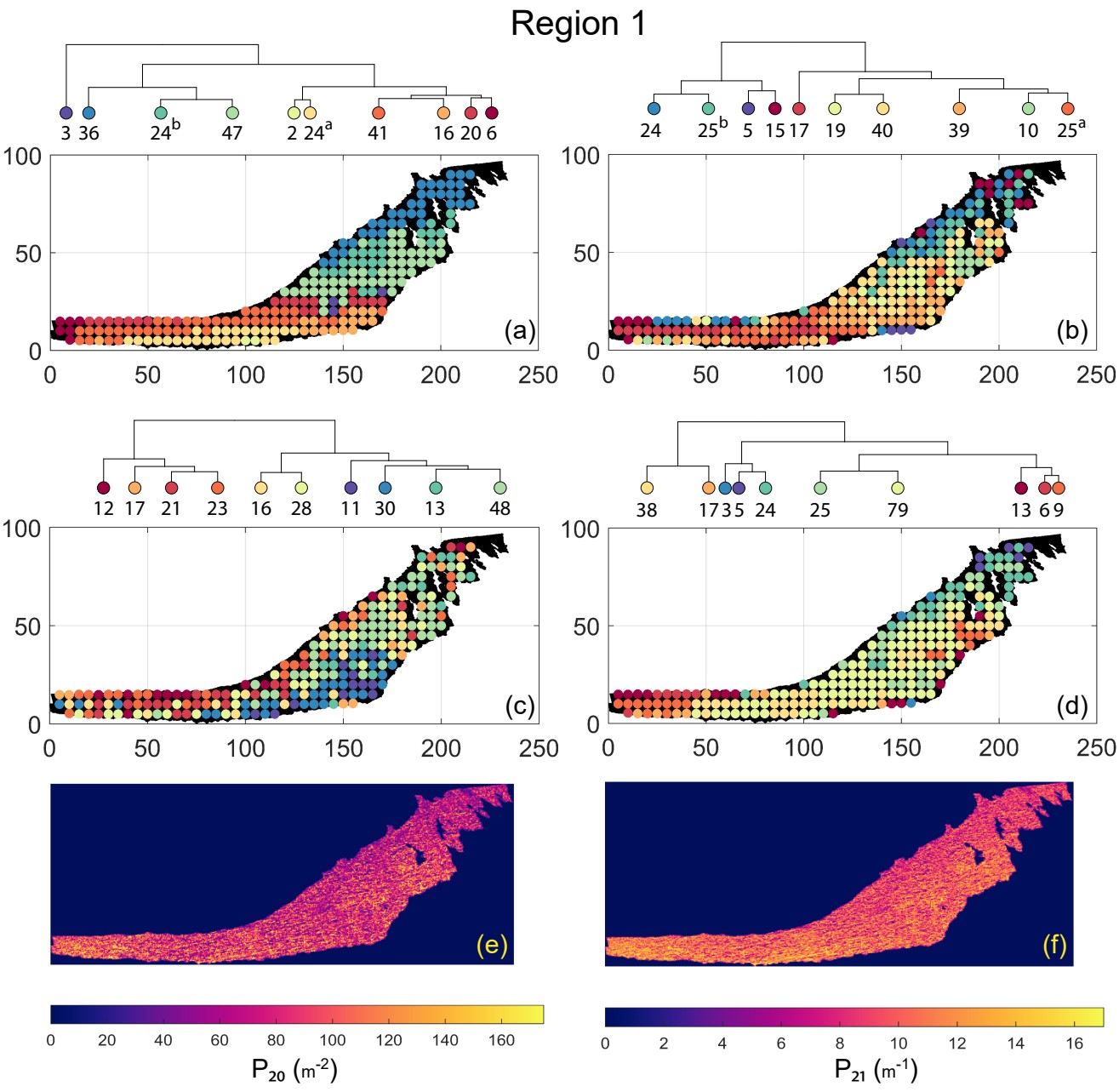

**Figure 17.** Hierarchical clustering results for Region 1 depicting the top 10 clusters using (a) Fingerprint distance (b) D-measure distance (c) NetLSD distance (d) Portrait Divergence distance (e) spatial $P_{20}$ (f) spatial $P_{21}$

**Figure 18.** Variation in fracture orientations and topological summary for Region 1 corresponding to (a) Fingerprint (b) D-Measure (c) Portrait divergence

### 5.2.2 Analysis of spatial variation in Region 2

Spatial distribution along with dendrograms of top ten clusters pertaining to the four graph similarity measures for Region 2 is depicted in Fig. 19. The full dendrograms and heatmaps are placed in Appendix Fig. A2. The variation of spatial clusters with different choices of dendrogram cut-heights is shown in Appendix Figs. B5 - B8. The number of sub-samples for the four similarity measures pertaining to a dendrogram cut of $k = 10$ is tabulated in Table. 5. Similar to Region 1, there is marked spatial autocorrelation with FP (Fig. 19.a), DM (Fig. 19.b), and PD (Fig. 19.d), whereas the LSD (Fig. 19.c) shows a speckled pattern. The spatial clustering results can be compared with the fracture persistence plots in Fig. 19(e) and Fig. 19(f).

**Table 5.** Summary of sub-graphs within each cluster of Region 2 for k = 10

| Metric ↓ | cluster 1 | cluster 2 | cluster 3 | cluster 4 | cluster 5 | cluster 6 | cluster 7 | cluster 8 | cluster 9 | cluster 10 |
|---|---|---|---|---|---|---|---|---|---|---|
| FP | 20 | 22 | 41 | 52 | 9 | 17 | 36 | 8 | 3 | 4 |
| DM | 19 | 20 | 14 | 25 | 10 | 16 | 23 | 38 | 23 | 24 |
| LSD | 2 | 6 | 31 | 28 | 53 | 9 | 15 | 38 | 5 | 25 |
| PD | 17 | 30 | 17 | 8 | 24 | 15 | 28 | 24 | 15 | 34 |
| Total | | | | | 212 | | | | | |

Node degree histograms and rose plots depict the differences in network topology and fracture orientations between the identified clusters pertaining to FP (Fig. 20.a), DM (Fig. 20.b), and PD (Fig. 20.c). For all three measures, the shape of rose plots indicates a transition of principal orientations smoothly across clusters. For example in Fig. 20(a) for FP, the more complex fracturing in the west of Region 2 is depicted by cluster $n20$ with a very diffuse rose plot, changing orientations to a predominantly orthogonal pattern in cluster $n03$. The DM (clusters $n16$ and $n10$ in Fig. 20.b) and PD (clusters $n08$ and $n17^b$ in Fig. 20.c) also identify this region of orthogonal fracturing. The corresponding topological summaries also depict an increased proportion of degree-4 nodes as compared to the histograms of other clusters.

From FP clustering results (see Fig. 19.a), the dendrogram identifies a western branch with clusters $n20$ and $n22$. The branch comprising of clusters $n8$, $n3$, $n4$ correspond to the radial fracturing region identified by Gillespie et al. (1993) that originates from the fault in the SE of Region 2. Clusters $n9$, $n17$, $n36$ all under a branch covering parts of Region 2 further away from the radial fracturing region. Clusters $n41$, $n52$ originate under a branch forming the northern and eastern boundaries of Region 2. Fig. C4 depicts archetypal sub-graphs under each cluster in detail for FP. The clustering results of DM (Fig. 19.b) and PD (Fig. 19.d) appear to be similar and with dendrograms roughly splitting into three main branches that correspond to specific portions of Region 1. First is the radial fracturing area represented by branch forming clusters $n10$, $n16$, $n23^b$ for D-measure (Fig. 19.b) and branch forming clusters $n17^b$, $n8$ for the portrait divergence (Fig. 19.d). The area to the N-W periphery of Region 2, farthest away from the fault, is represented by branch forming clusters $n19$, $n20$, $n14$, $n25$ for DM and by branch forming clusters $n24^a$, $n15^a$, $n34$ for PD. The transition region branch is represented within the DM dendrogram by clusters $n38$, $n23^a$, $n24$ and within the PD dendrogram by clusters $n24^b$, $n15^b$, $n28$. Fig. C5 and Fig. C6 depict detailed sub-graph examples for DM and PD, respectively.

**Figure 19.** Hierarchical clustering results for Region 2 depicting the top 10 clusters using (a) Fingerprint distance (b) D-measure distance (c) NetLSD distance (d) Portrait Divergence distance. (e) spatial $P_{20}$ (f) spatial $P_{21}$

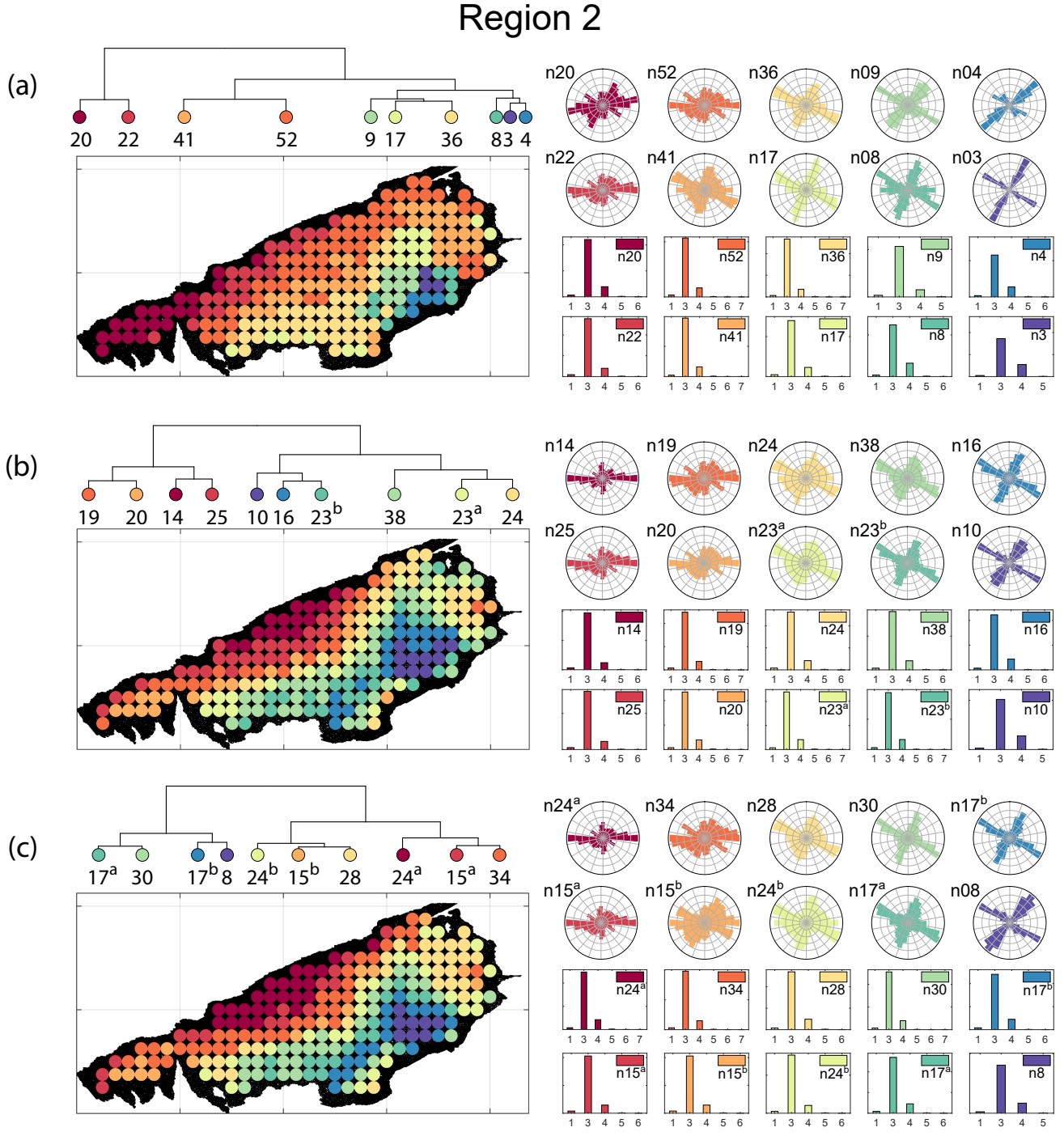

**Figure 20.** Variation in fracture orientations and topological summary for Region 2 corresponding to (a) Fingerprint (b) D-Measure (c) Portrait divergence

### 5.2.3 Analysis of spatial variation in Region 3

The spatial distribution along with dendrograms of the top 10 clusters pertaining to the four graph similarity measures for Region 3 is depicted in Fig. 21. The full dendrograms and heatmaps are placed in Appendix Fig. A3. The variation of spatial clusters with different choices of dendrogram cut-heights (and number of clusters) is shown in Appendix Figs. B9 - B12. Similar to the Region 1 and 2 results, there is marked spatial autocorrelation with FP (Fig. 21.a), DM (Fig. 21.b), and PD (Fig. 21.d), whereas the LSD (Fig. 21.c) shows a stippled pattern. The spatial clustering results can be compared with the fracture persistence plots in Fig. 21(e) and Fig. 21(f). The number of sub-samples for the four similarity measures associated with a dendrogram cut of $k = 10$ is tabulated in Table. 6.

**Table 6.** Summary of sub-graphs within each cluster of Region 3 for k = 10

| Metric ↓ | cluster 1 | cluster 2 | cluster 3 | cluster 4 | cluster 5 | cluster 6 | cluster 7 | cluster 8 | cluster 9 | cluster 10 |
|---|---|---|---|---|---|---|---|---|---|---|
| FP | 2 | 5 | 28 | 11 | 27 | 6 | 9 | 4 | 11 | 14 |
| DM | 24 | 15 | 6 | 16 | 1 | 5 | 5 | 25 | 3 | 17 |
| LSD | 9 | 25 | 4 | 9 | 6 | 5 | 5 | 10 | 16 | 28 |
| PD | 7 | 23 | 25 | 1 | 5 | 21 | 4 | 16 | 3 | 12 |
| Total | | | | | 117 | | | | | |

Node degree histograms and rose plots depict the differences in network topology and fracture orientations between the identified clusters relating to FP (Fig. 22.a), DM (Fig. 22.b), and PD (Fig. 22.c). For all three measures, the shape of rose plots indicates a transition of principal orientations smoothly across clusters. For example, in Fig. 22(a) for the fingerprint measure, the cluster $n06$ in the west of Region 3 has three main sets that become orthogonal in cluster $n09$, the nearest cluster eastwards. Cluster $n05$ at the eastern extremity of Region 3 has an orthogonal pattern that has rotated almost 80 degrees clockwise compared to the western boundary. Orientations of fractures clusters between the eastern-most and western-most clusters show transitions between the extremal archetypes.

From the FP clustering results (Fig. 21.a), the spatial variation appears to have an E-W trend. From the dendrogram, an eastern branch comprising clusters $n6$, $n9$, 4, $11^a$, 14 and a western branch consisting of clusters $n5$, $n28$, $n11^b$, $n27$ can be identified. An outlier branch with Cluster $n2$ appears at the interface between the eastern and western branches. Detailed visualization of archetypal sub-graphs relating to each of the FP clusters is presented in Fig. C7. The dendrogram structure and spatial clustering for the DM (Fig. 21.b) depicts a central region represented by a branch containing clusters $n24$, $n15$, $n6$, $n16$. The eastern and western peripheries organize as clusters $n1$, $n5^a$, $n5^b$, $n25$, $n3$, $n17$ under a second branch. Underneath this branch, clusters $n1$, $n5^a$, $n5^b$ correspond to extremities of the Region 3, which are not fully sampled. The dendrogram structure for the PD (Fig. 21.d) is similar with clusters $n7$, $n23$, $n25$ organizing under the branch representing the central region and clusters $n1$, $n5$, $n21$, $n4$, $n16$, $n3$, $n12$ forming the eastern and western peripheral regions. Clusters $n1$ and $n5$ pertain to extremities of Region 3, which are not fully sampled. Figure. C8 and Fig. C9 depict zoomed-in sections of the sub-graphs relating to each of the top clusters that confirm the detected intra-regional variation for DM and PD, respectively.

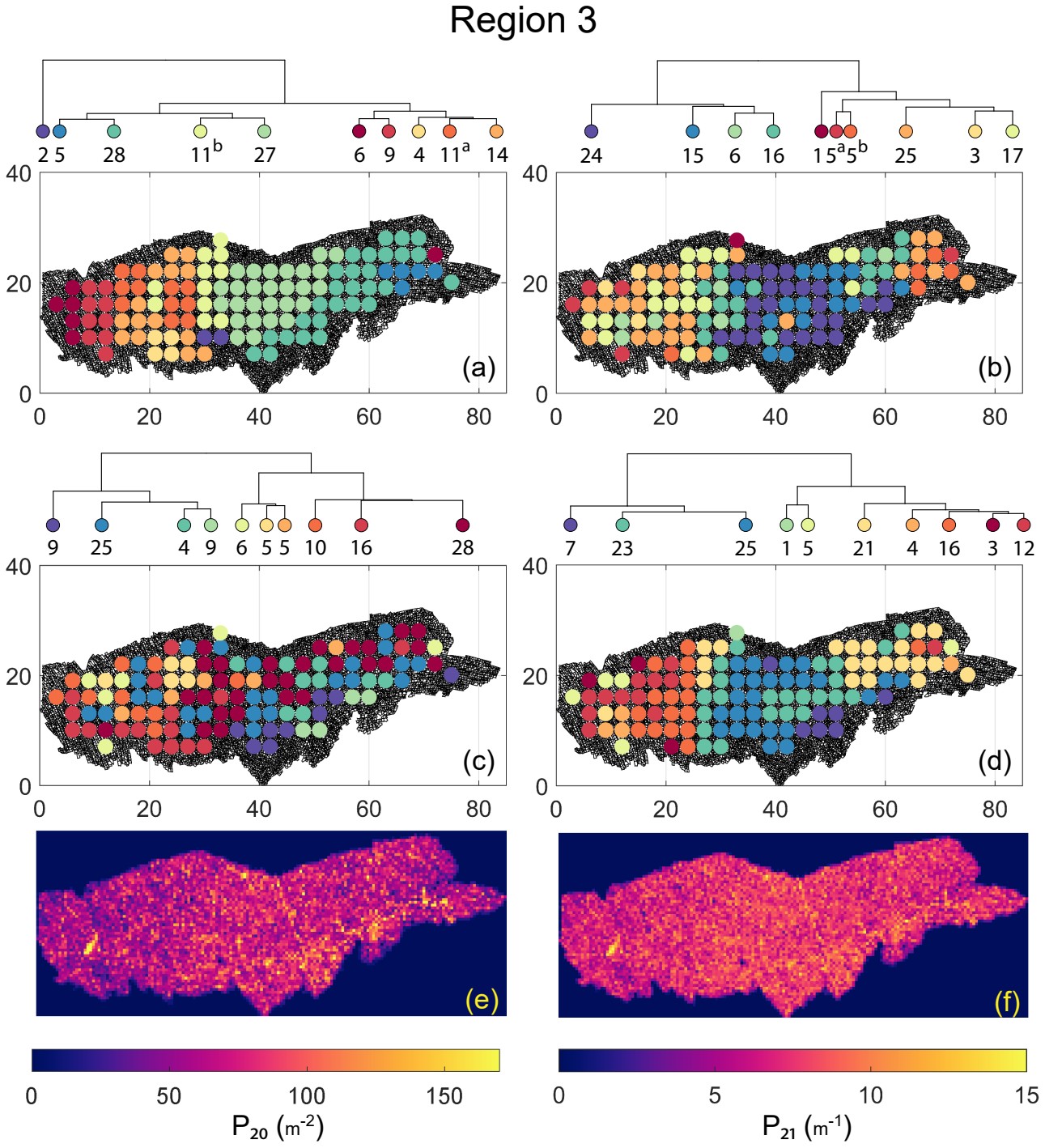

**Figure 21.** Hierarchical clustering results for Region 3 depicting the top 10 clusters using (a) Fingerprint distance (b) D-measure distance (c) NetLSD distance (d) Portrait Divergence distance (e) spatial $P_{20}$ (f) spatial $P_{21}$

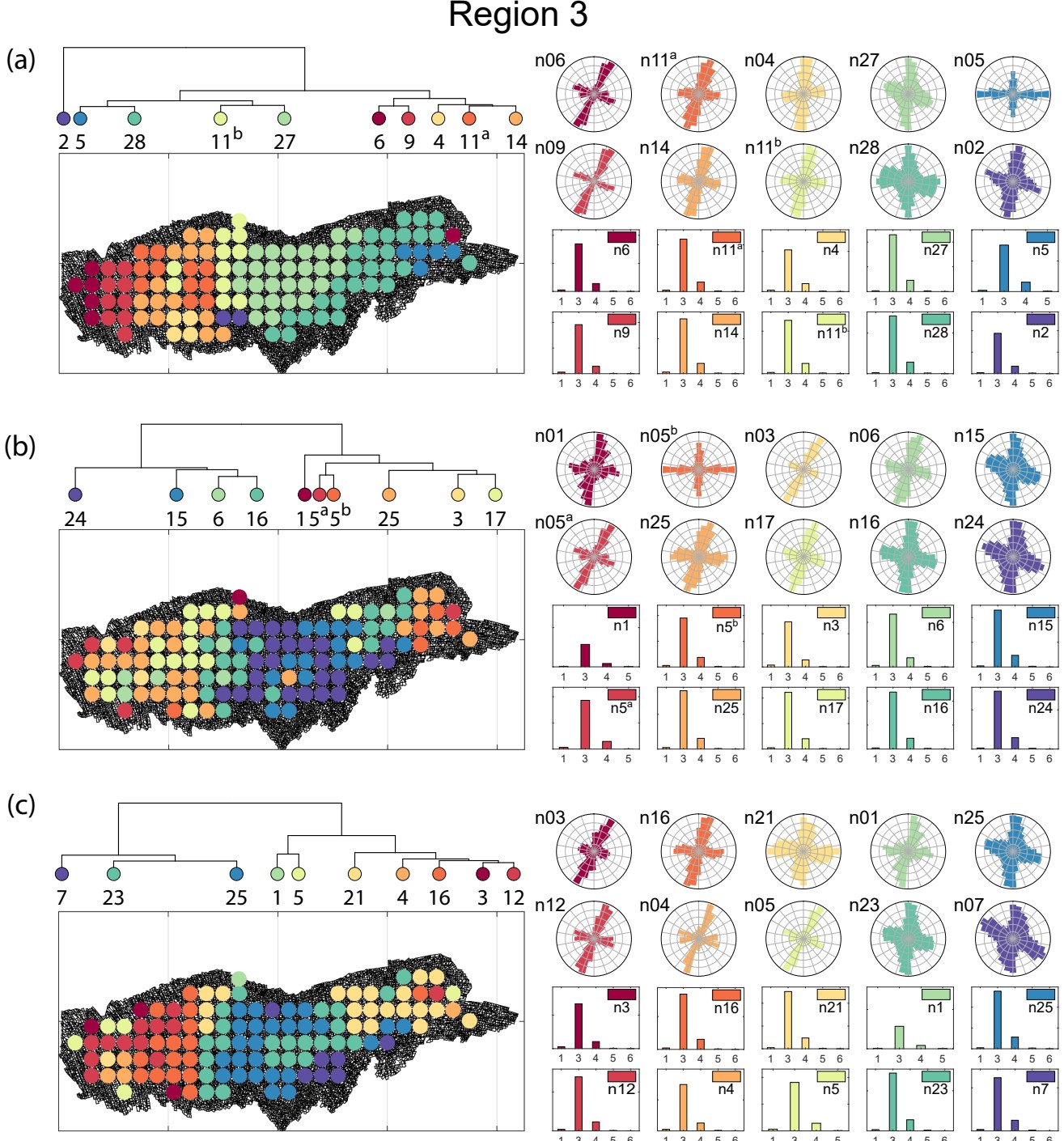

**Figure 22.** Variation in fracture orientations and topological summary for Region 3 corresponding to (a) Fingerprint (b) D-Measure (c) Portrait divergence

## 6 Discussion

Within the structural geology literature, the quantitative fracture persistence measures of Dershowitz and Herda (1992), the topological approach of Sanderson and Nixon (2015), and qualitative descriptions are most commonly resorted to for comparing 2D fracture networks. The lack of quantitative measures for spatial network data is partially due to the lack of extensive 2D fracture trace data. Using the fully mapped, UAV-derived dataset of an extensive fracture network, it is possible to systematically investigate 2D fracture network organization variations.

In this contribution, we treat 2D fracture networks as planar graph structures and apply graph similarity measures to quantitatively compare sub-samplings within large fracture networks and discover clusters of similarity. The statistical technique of HC was used along with graph distance metrics to extract spatial clusters. Sub-graphs within a spatial cluster are more similar to each other than other clusters. A hierarchy of patterns is derived based on similarity scores, which can be examined at deeper levels.

One can argue that variation exists at multiple length scales, and more granular inquiry would lead to different clusters. While our choices of grid-spacing and sub-sampling of graphs were to keep computational requirements in mind, it is possible to do more dense sub-sampling than what we have already achieved to further highlight spatial variations within a given network. The clusters that we have depicted are particular to the spacing and sampling diameters that we have chosen. In this section, we discuss some additional perspectives and issues related to our methodology and results.

– **Linking spatial variation patterns to fracturing drivers** The results indicate that spatial variation in fracture networks is not always evident from the ubiquitously used fracture persistence measures, such as $P_{20}$ and $P_{21}$. The proposed method highlights variations in network structure which can then help draw inferences into possible drivers for the spatial differences. In the case of Regions 2 and 3, the proximity to the fault influences network development. Such a model has been proposed by Peacock and Sanderson (1995), Gillespie et al. (2011), and Wyller (2019), where the

oldest fractures are long and radial, emanating from local asperities within the fault. These older fractures then influence the development of younger fractures. This is observed in Region 2, where clusters form roughly parallel to the E-N-E striking fault with the direction of variation to the N-W. Region 3 is positioned between two such asperity epicentres. There are long, radial fractures on the eastern and western extremities with a transition region in-between. The direction of cluster variation trends E-W. Fracture pattern variation in Region 1 is not affected by faulting. Since Regions 1 and

2, pertain to a single layer, the N-E regions of Region 1 show visual similarities between the westernmost extremities of Region 2. The intraregional variations in Region 1 could be due to layer thickness variation, although we do not have sufficient thickness data to confirm this.

The analysis of spatial variation can assist in deciphering fracture timing. Given the temporal nature of network formation, it is desirable to delineate network evolution into relative episodes of fracturing. In previous analyses specific to the

430 Lilstock dataset used in this contribution, Passchier et al. (2021) identified jointing sets with timing history based on fracture length, strike, and topological relationships. Although the temporal history is identified from joints that were picked manually but not wholly by Passchier et al. (2021), there is still a discernable spatial variation where some jointing sets

are localized while others occur throughout the outcrop. Identifying spatial clustering in complete networks provides a basis by which joint sets can then be arranged in a hierarchy of temporal development.

– **On the choice of a graph distance metric** We have restricted our investigation scope to four state-of-the-art graph similarity distances from the recent graph theory literature. Many more graph distances applicable to spatial graphs exist (Hartle et al., 2020; Tantardini et al., 2019); furthermore, the best means remain an open problem in network science research. Some novel distance measures are not graph-based but derive from persistent homology (such as Feng, 2020). In this approach that considers the *shape of data*, persistence diagrams are generated from spatial graphs, 440     and bottleneck distances are combined with hierarchical clustering to discover clusters. The results from Feng (2020) compared favourably to that of Louf and Barthelemy (2014) when applied to patterns of cities.

As may be observed from our results, the metrics highlight certain aspects of the fracture network while not considering others. For instance, the fingerprint distance only considers block area and shape factor distributions of the blocks and neglects orientations. The other three distances use graph properties directly, and hence orientation information (or the 445     lack of it) is a consequence of how the spatial graph is defined. We used weighted graphs that incorporate euclidean distance between nodes as edge weights for the similarity computations. However, each edge also has a striking attribute to completely describe its position in 2D space (in the case of 3D, it needs a dip). Ideally, the edge weight should then be a vector, $w = [l, \theta]$ incorporating both lengths, 'l' and orientation, '$\theta$', but the distance metrics we use do not allow the use of non-scalar weights.

– **Do REV's exist for fracture networks** In the context of fractured reservoir modelling, identification of a representative elemental volume (REV) aids continuum-based simulation approaches. However, the complexities of fluid flow and transport through fractured porous media require an explicit representation of fractures. Given the difficulties associated with obtaining realistic network geometries, stochastic-process-based methods derived from sparse fracture data are commonplace. However, these methods are often unable to represent inherent non-stationarity in spatial variation 455     (Thovert et al., 2017), and work by Andresen et al. (2013) find that DFNs from nature exhibit disassortativity, which is not a property of generated networks. Other techniques based on multipoint statistics (Bruna et al., 2019b) attempt image-based approaches to modelling non-stationary networks. Estrada and Sheerin (2017) present a different approach in which DFNs are directly generated as spatial graphs (referred to as *random rectangular graphs*). Such a method can incorporate insights from outcrop-derived NFRs.

Regardless of the method used to extrapolate, stationarity decisions have to be made based on hard data, and this is where our approach is helpful. We can use outcrop-derived networks to define and delineate stationarity's spatial boundaries and assign a particular type of network with due cognition of the inherent graph structure. Much literature exists on linking fracture patterns to high-deformation drivers such as folding, faulting, and diapirism, with the goal being to identify and correlate appropriate outcrop analogues to particular subsurface conditions. As our clustering results indicate, at the 465     dimensional scales of sampling we have used, Tobler's first law of geography applies to fracture networks. Therefore, a representative network based on network similarity can be derived. The method can be applied to analogues for which

data already exists. Further work is required to differentiate fluid-flow and transport responses of the identified cluster type.

- **Other clustering methods** We have used a combination of HC and graph distance metrics to delineate regions within a spatial graph and arrange them in a hierarchy of similarities. Within the graph theory literature, there are other non-HC methods based on graph properties such as modularity (Newman and Girvan, 2004; Blondel et al., 2008; Traag et al., 2019) or by graph spectral partitioning (Fiedler, 1973; Spielman and Teng, 2007). Recent developments using graph neural networks and graph machine learning include modifications on the concept of modularity (Tsitsulin et al., 2020) and spectral methods (Bianchi et al., 2020) towards the goal of partitioning graphs into clusters.

## 7   Conclusions

This contribution presents a method to automatically identify spatial clusters and quantify intra-network spatial variation within 2D fracture networks. We test the technique on 2D trace data from a prominent limestone outcrop within the Lilstock pavements, located off the southern coast of the Bristol Channel, UK. The fracture network data that spans three separate regions and covers over 14,200 sq.m is converted to the form of planar graph structures, spatially sampled into sub-graphs, and then compared using four different graph-distance measures. The pair-wise similarities in the form of distance matrices are used to arrange region-wise sub-graphs into a hierarchical relationship structure, also referred to as a dendrogram, using the statistical technique of hierarchical clustering. Positional order information from the dendrogram is used to render maps depicting the spatial variation within the fracture networks. The delineations of these intra-network sub-patterns provide a way to identify representative elemental volumes that preserve fracture networks' topological and geometric properties. The presence of these sub-regions can also serve as a guide to making decisions on stationarity with respect to geostatistical modelling. The main findings of the study are summarized as:

- representing fracture networks as graphs enable combining hierarchical clustering and graph-distance metrics to reveal interesting intra-network spatial similarity patterns not otherwise discernable from existing global or local fracture network descriptors.

- organization of fracture network sub-graphs based on pair-wise similarities into a hierarchical tree enables identification of spatial clustering at different dendrogram heights with newer and more granular cluster boundaries emerging at successively deeper levels of enquiry.

- spatial autocorrelation is more apparent with the fingerprint, D-measure, and the portrait divergence distances than the NetLSD, which yields speckled patterns with little or no spatial autocorrelation.

- spatial variation maps deriving from hierarchical clustering using the D-measure and portrait divergence identify similar spatial clusters and cluster boundaries. However, with the fingerprint distance, the cluster boundaries are different.

    – fracture segment orientations show gradual variation in segment strikes across the identified clusters despite orientation not being explicitly considered and only euclidean distance being used to weight spatial graph edges.

*Code and data availability.* A MATLAB implementation to compute graph fingerprints and fingerprint distance is available on the Github repository https://github.com/rahulprabhakaran/Fracture_Fingerprint/tree/v.1.0.0 [last access: 19 April 2021; see 10.5281/zenodo.4699961, Prabhakaran (2021)]. The implementation of the D-measure in the form of an R script is available as supplementary code with Schieber et al. (2017). The `NetLSD` python package used to compute the LSD distance as described in Tsitsulin et al. (2018) is available at https://pypi.org/project/NetLSD/. The code implementation for portrait divergence developed by Bagrow and Bollt (2019) can be obtained from https://github.com/bagrow/network-portrait-divergence/.

The circularly sampled fracture subgraphs are derived from the open fracture network dataset published by Prabhakaran (2021). The circularly sampled subgraphs are available for download as a data supplement to this manuscript (Prabhakaran et al., 2021a).

## Appendix A: Heatmaps and dendrograms

# Region 1

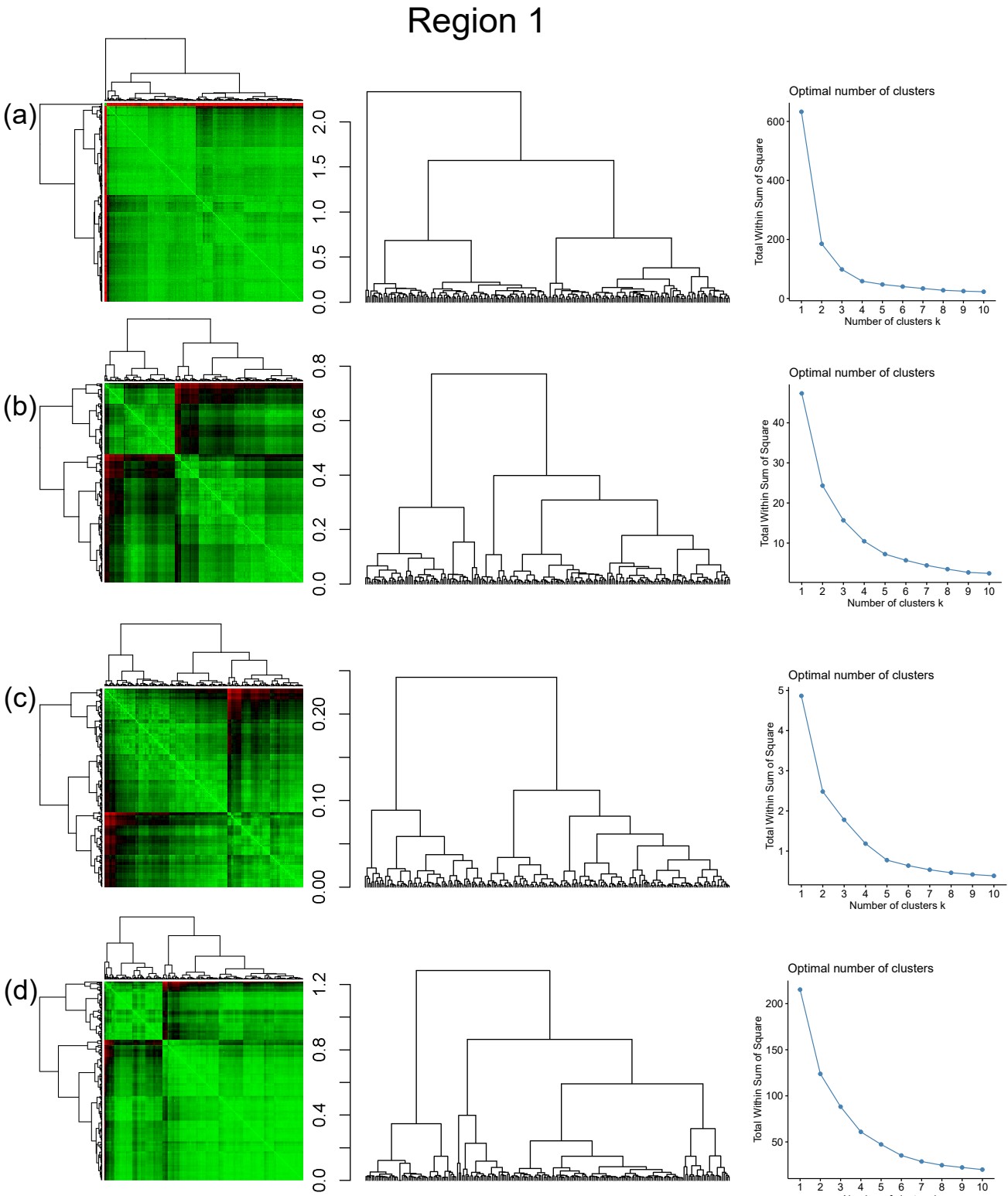

**Figure A1.** Combined symmetric heatmap of distance matrix and dendrograms, dendrograms, and sum-of-squares elbow plots for Region 1

(a) Fingerprint (b) D-measure (c) NetLSD (d) Portrait Divergence

# Region 2

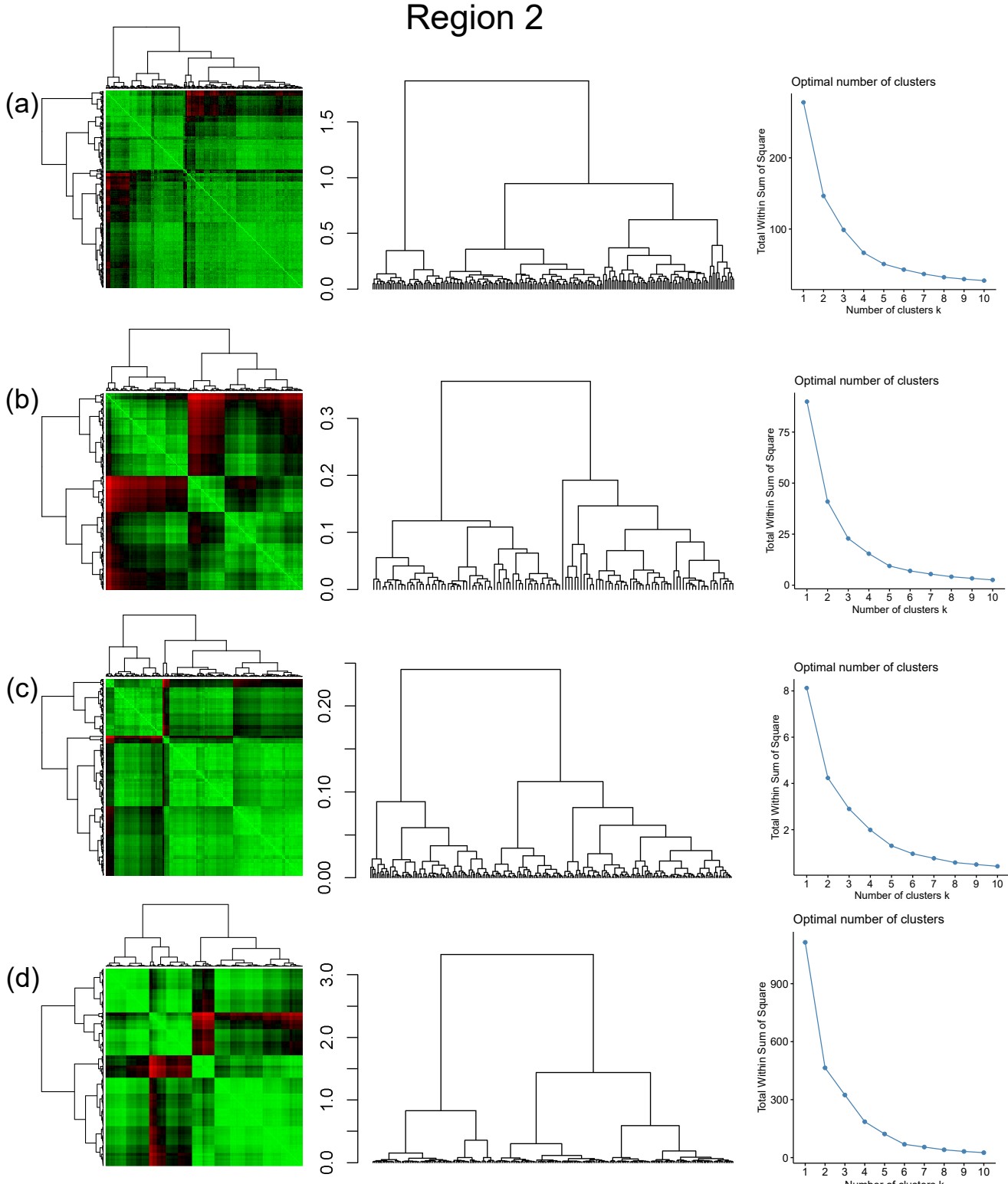

**Figure A2.** Combined symmetric heatmap of distance matrix and dendrograms, dendrograms, and sum-of-squares elbow plots for Region 2

(a) Fingerprint (b) D-measure (c) NetLSD (d) Portrait Divergence

# Region 3

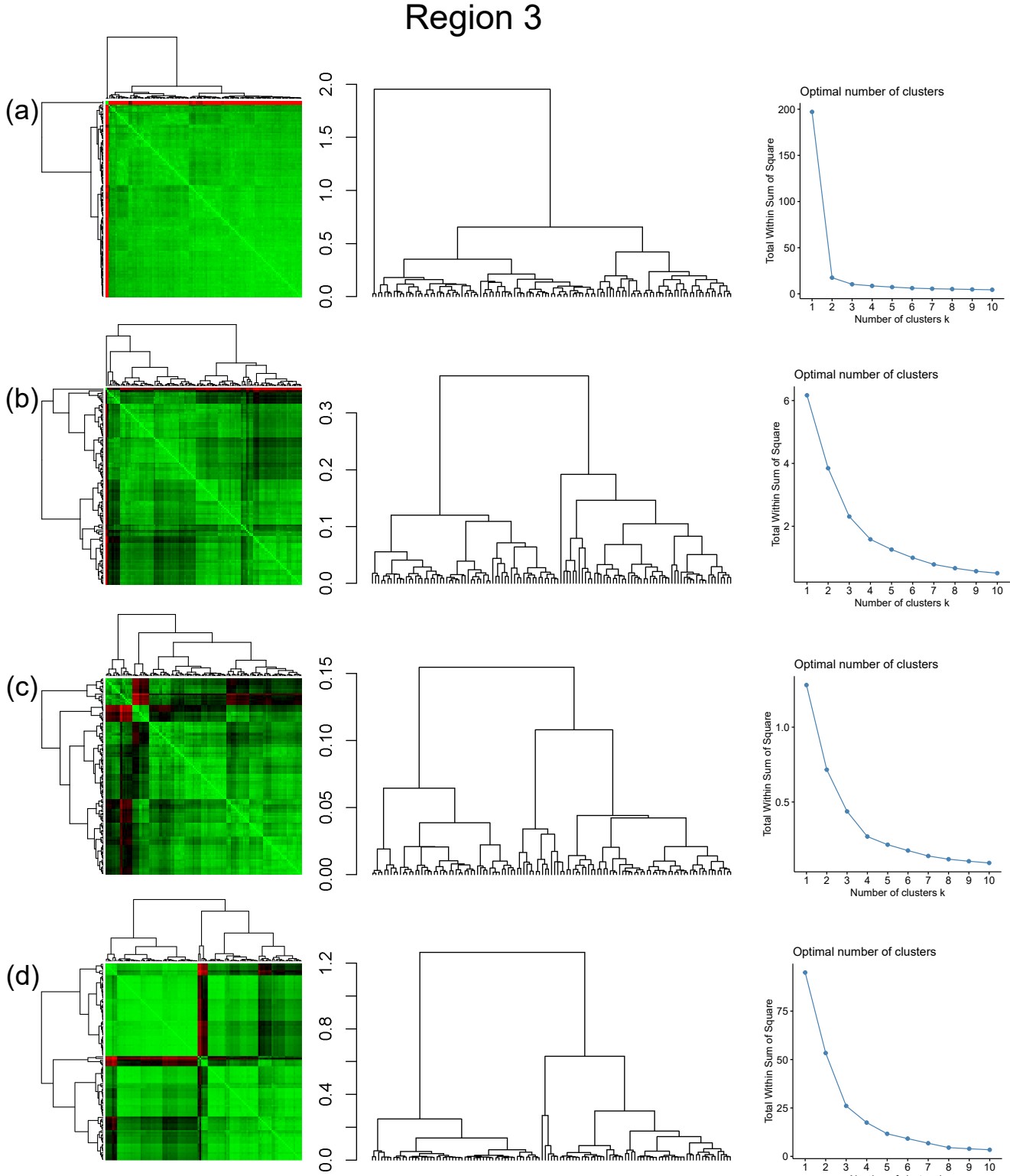

**Figure A3.** Combined symmetric heatmap of distance matrix and dendrograms, dendrograms, and sum-of-squares elbow plots for Region 3
(a) Fingerprint (b) D-measure (c) NetLSD (d) Portrait Divergence

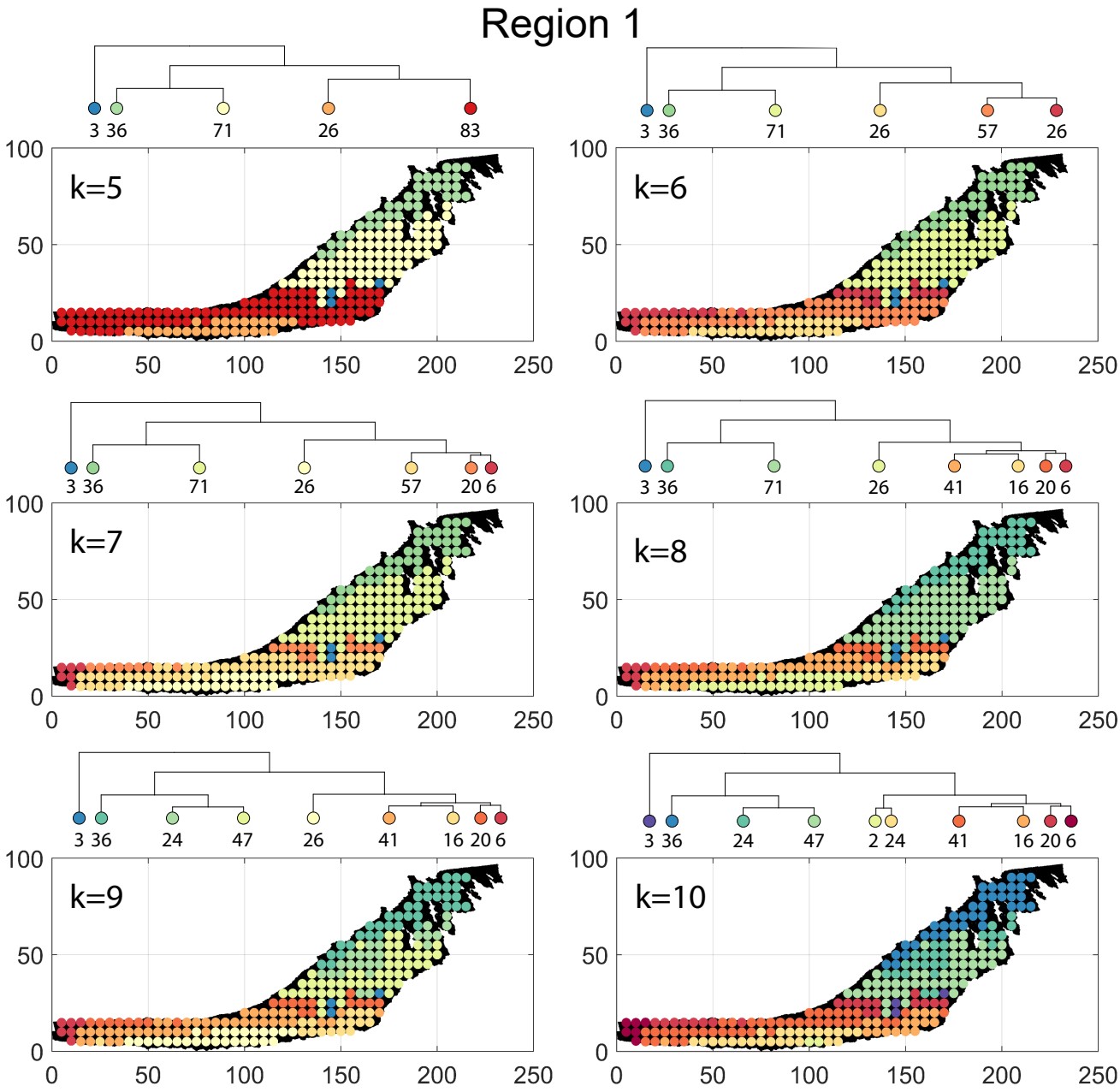

**Figure B1.** Variation in cluster boundaries for 'k' clusters in Region 1 using Fingerprint Distance

# Region 1

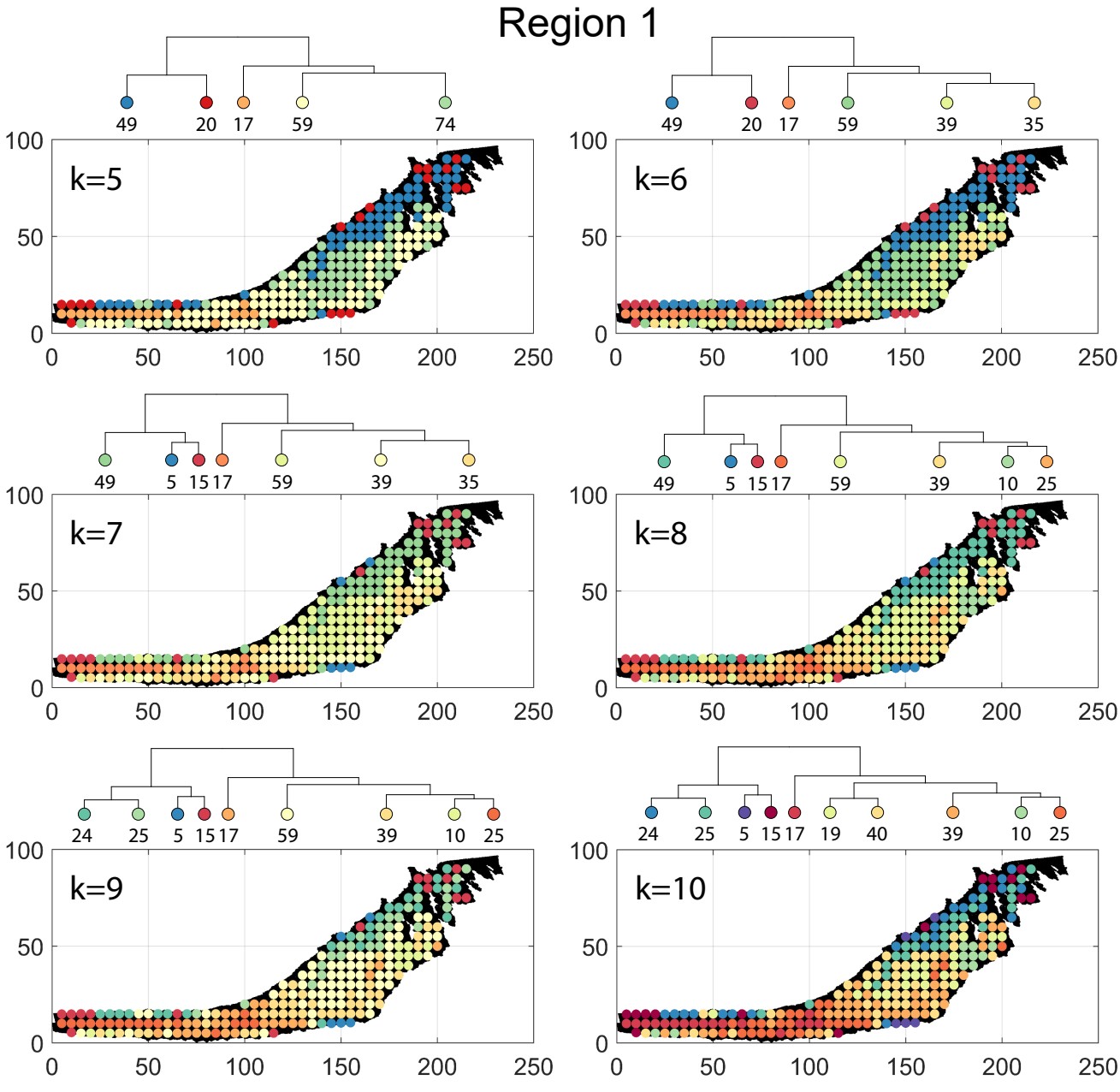

**Figure B2.** Variation in cluster boundaries for 'k' clusters in Region 1 using D-Measure

# Region 1

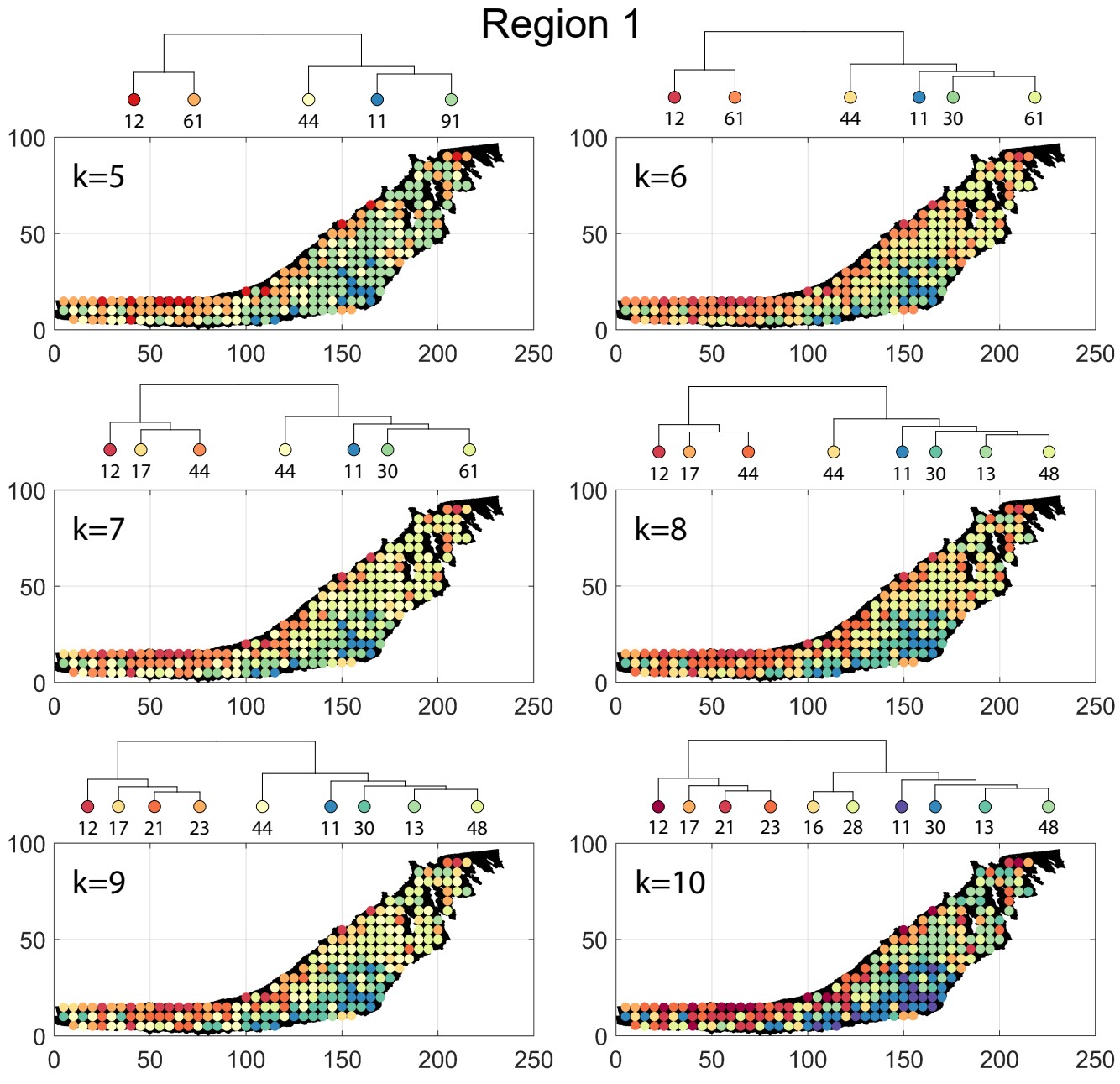

**Figure B3.** Variation in cluster boundaries for 'k' clusters in Region 1 using NetLSD

# Region 1

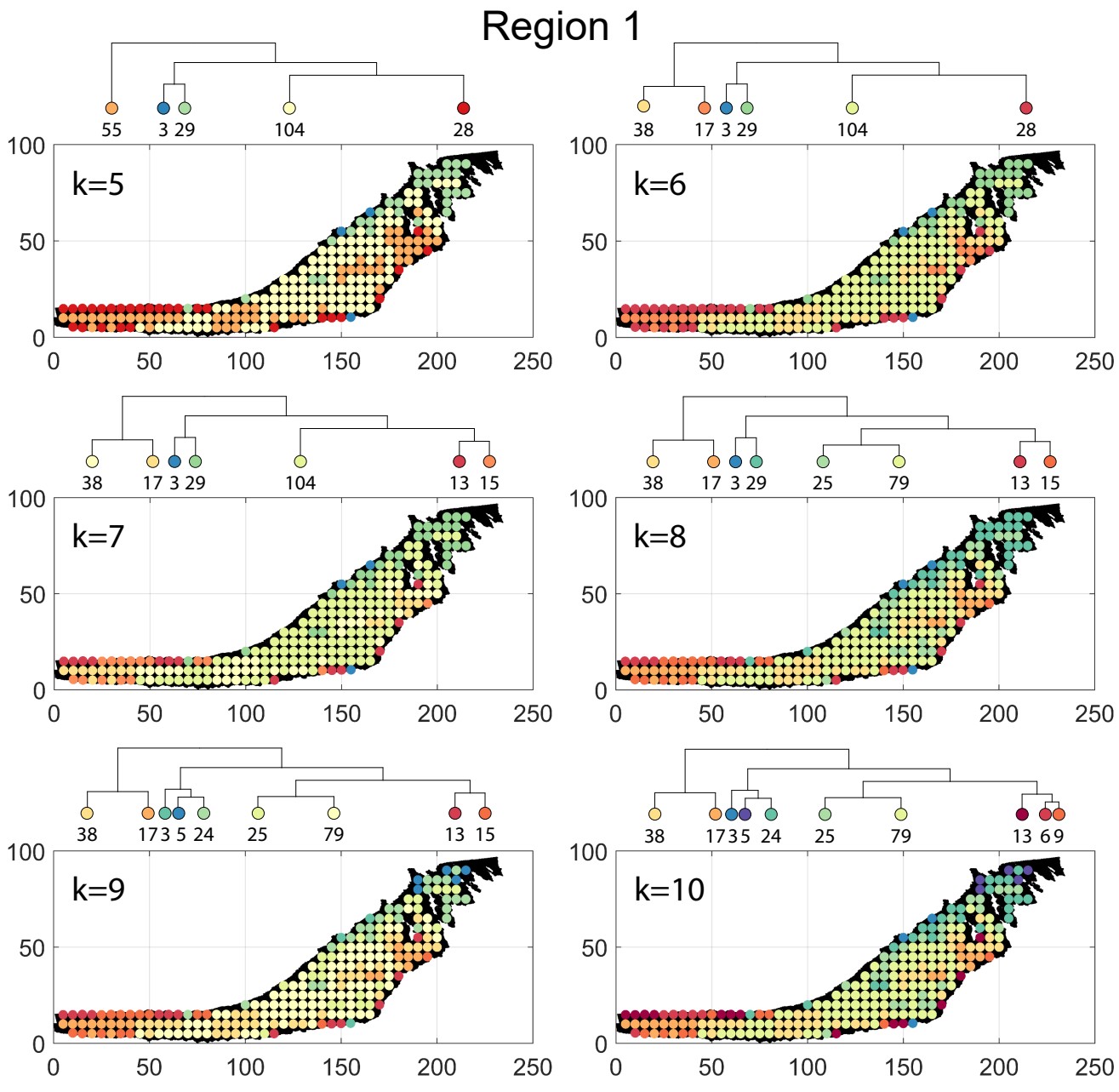

**Figure B4.** Variation in cluster boundaries for 'k' clusters in Region 1 using Portrait Divergence

# Region 2

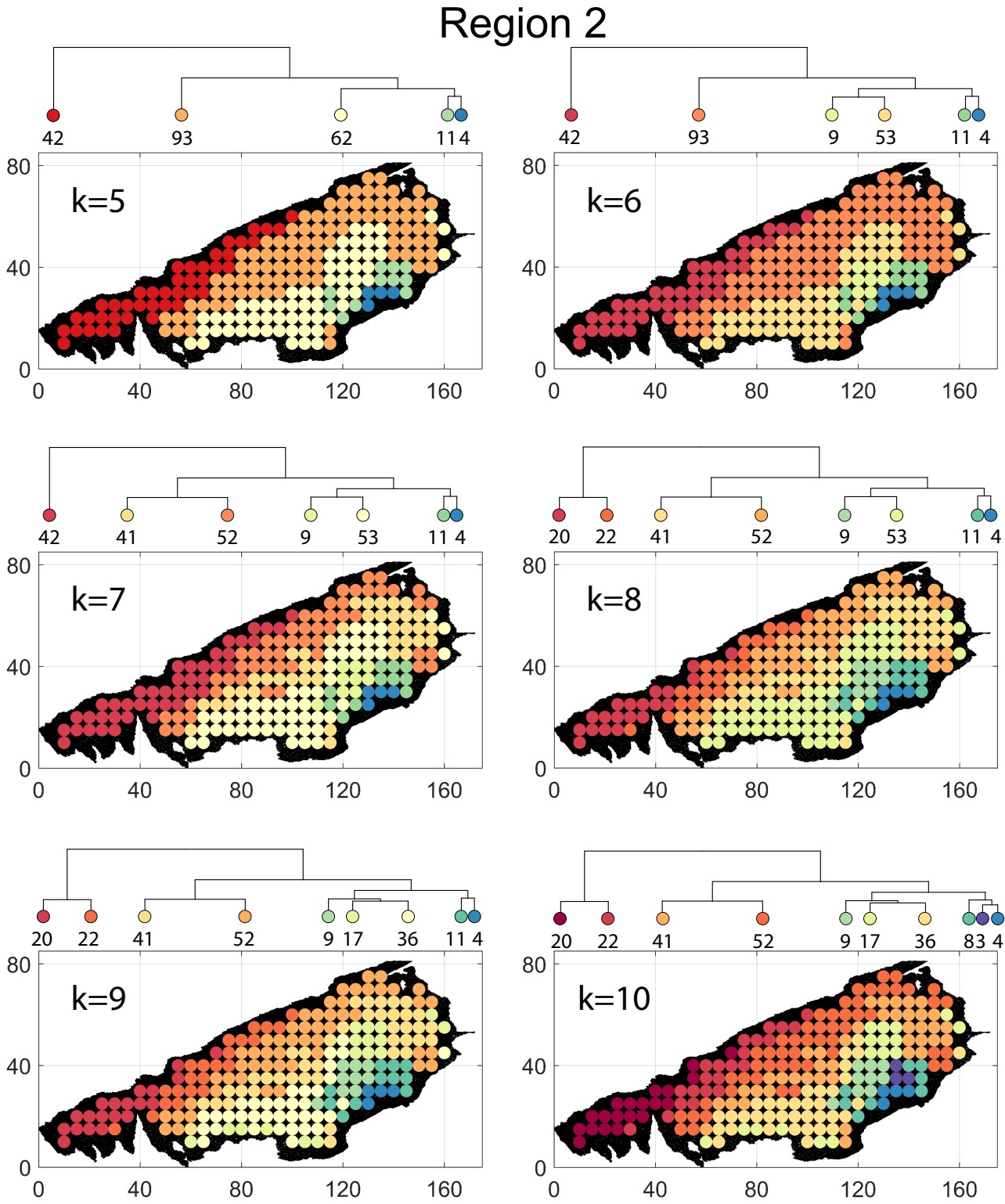

**Figure B5.** Variation in cluster boundaries for 'k' clusters in Region 2 using Fingerprint Distance

# Region 2

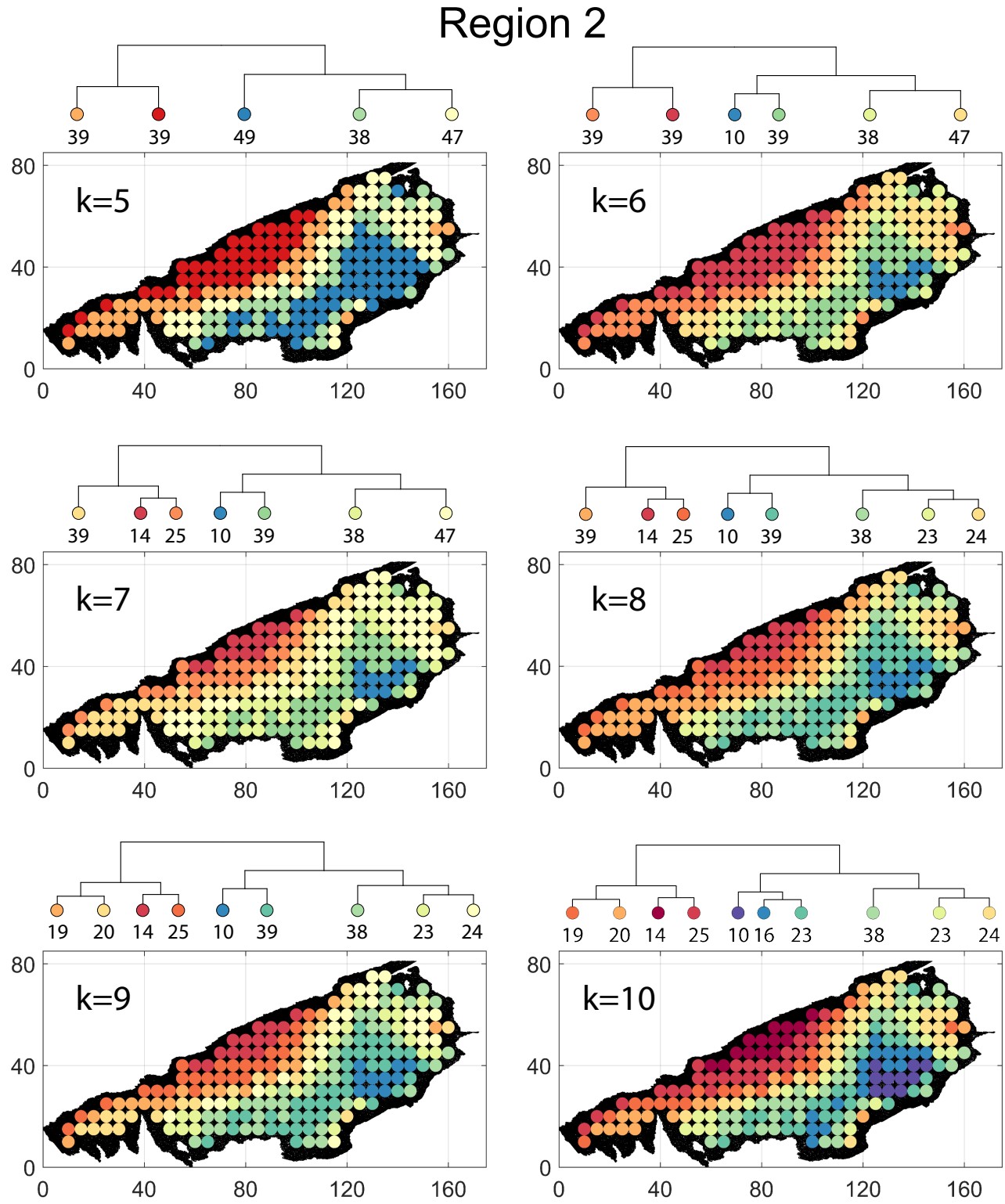

**Figure B6.** Variation in cluster boundaries for 'k' clusters in Region 2 using D-Measure

# Region 2

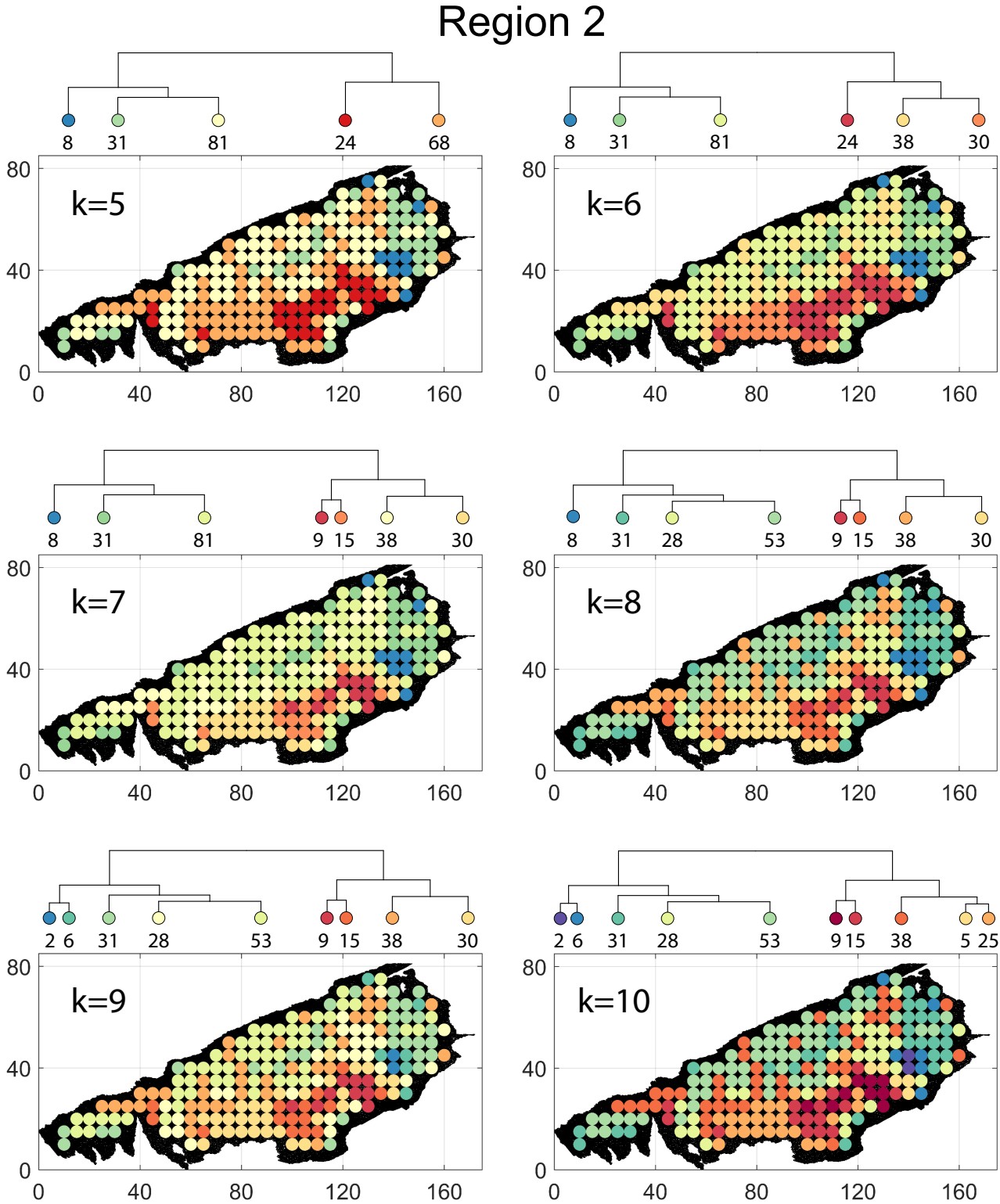

**Figure B7.** Variation in cluster boundaries for 'k' clusters in Region 2 using NetLSD

# Region 2

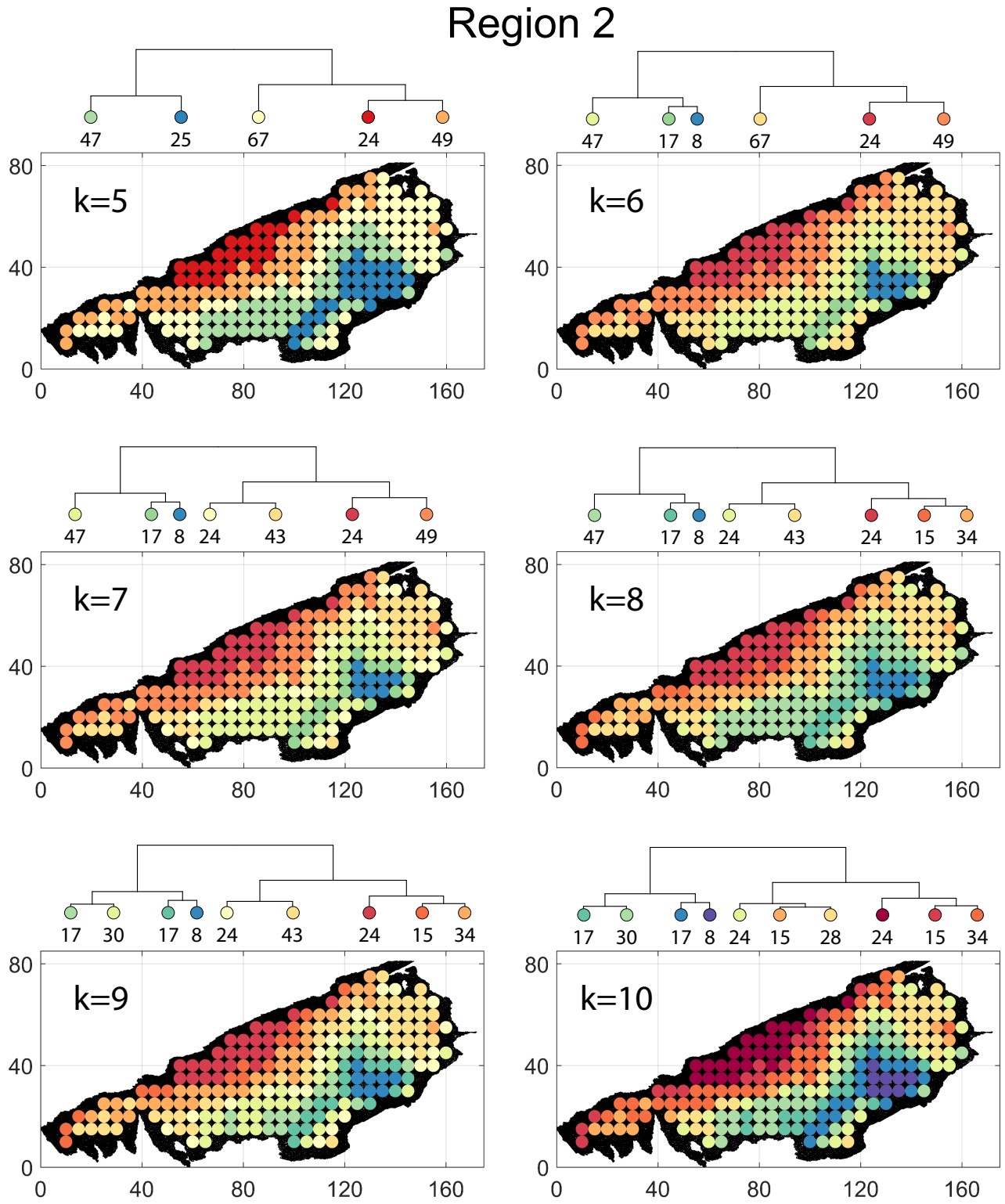

**Figure B8.** Variation in cluster boundaries for 'k' clusters in Region 2 using Portrait Divergence

# Region 3

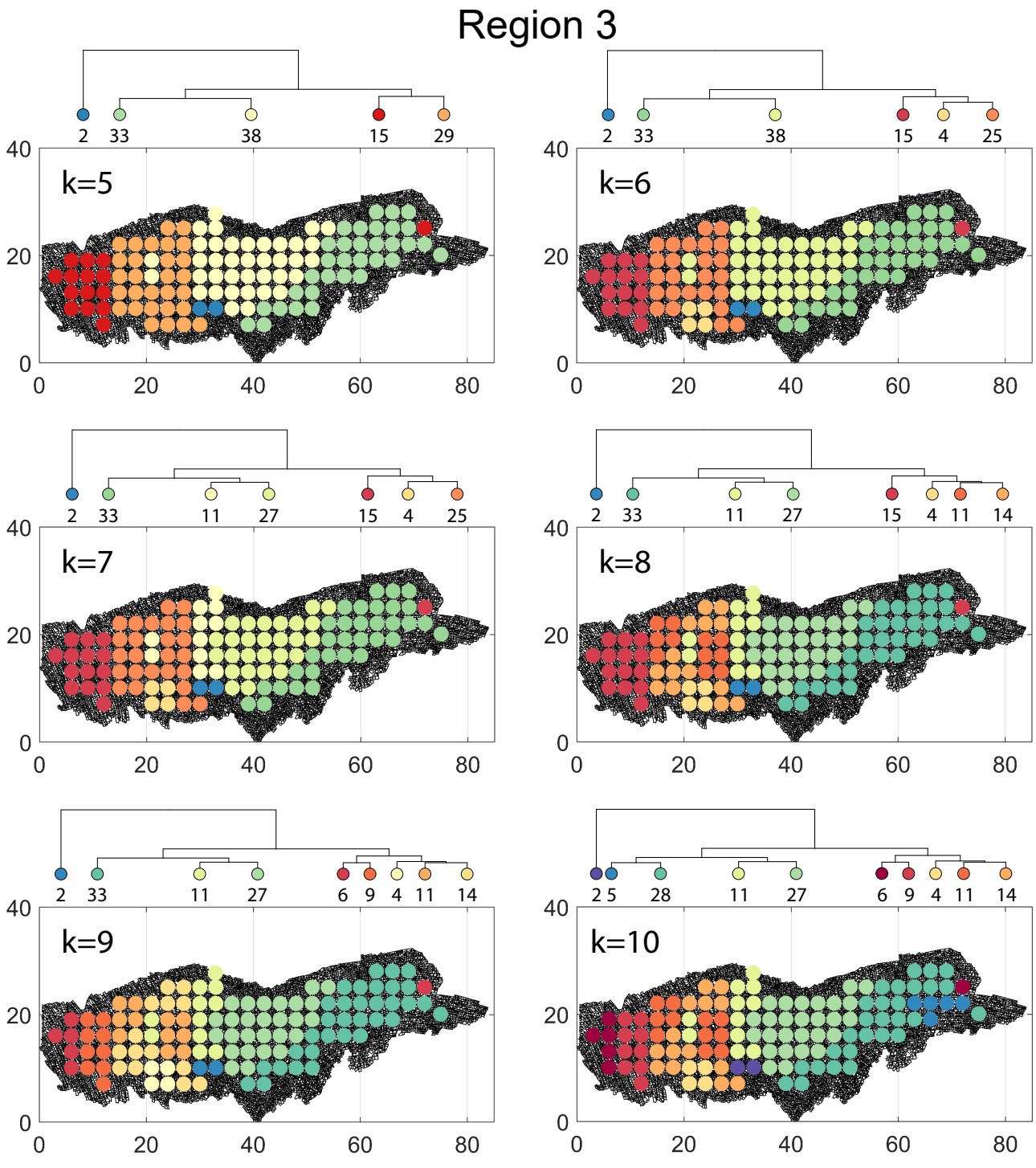

**Figure B9.** Variation in cluster boundaries for 'k' clusters in Region 3 using Fingerprint Distance

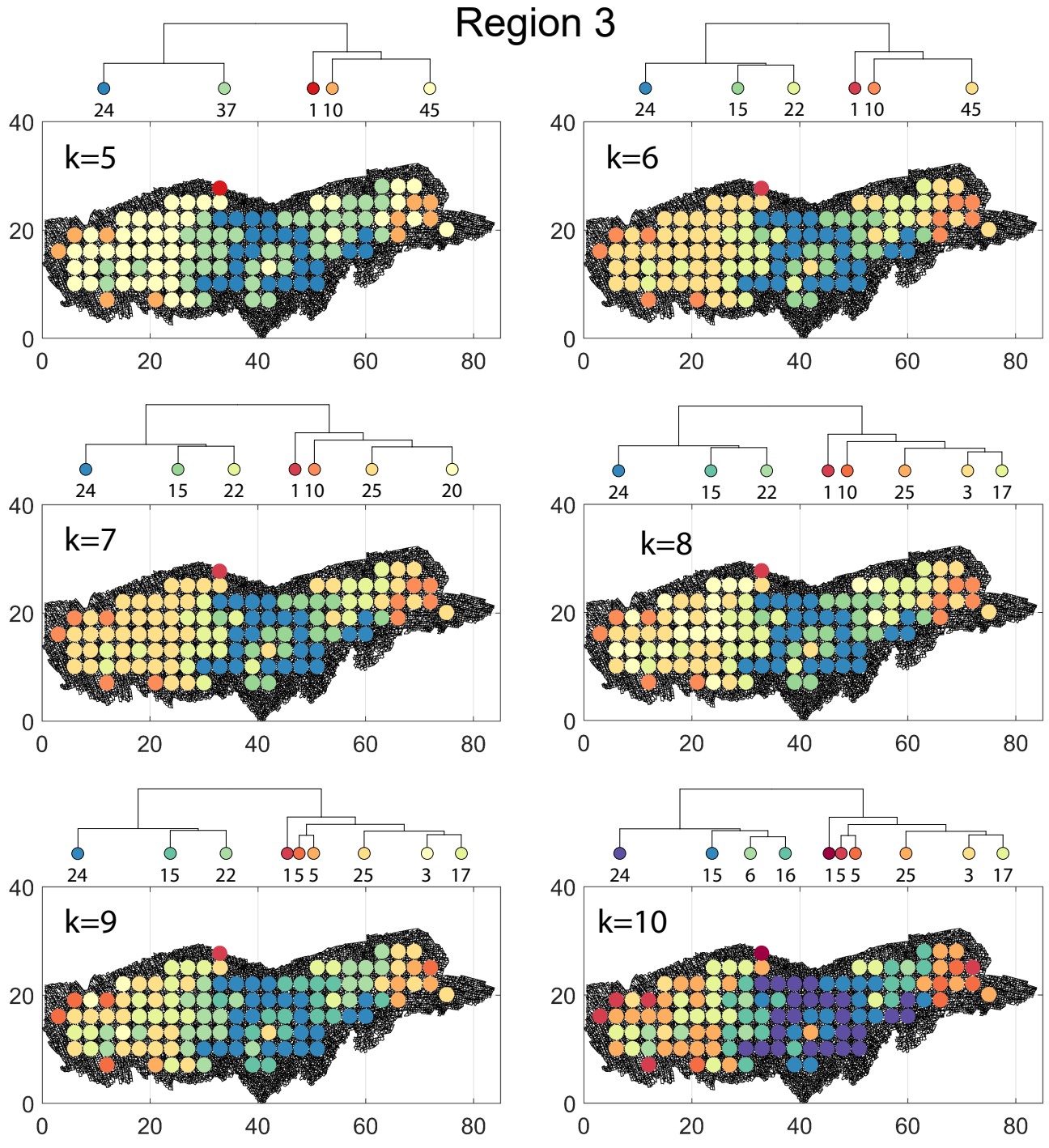

**Figure B10.** Variation in cluster boundaries for 'k' clusters in Region 3 using D-Measure

# Region 3

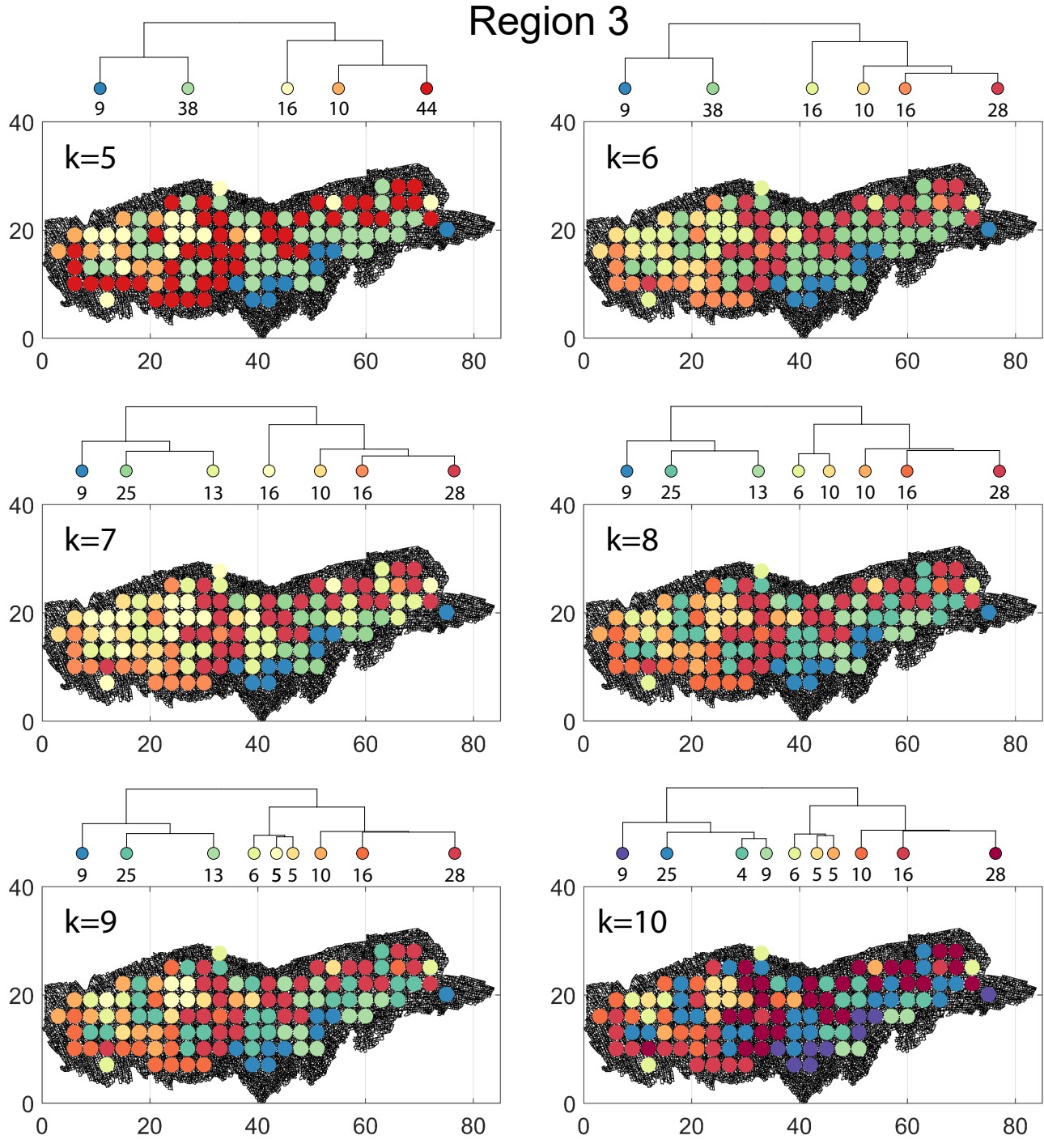

**Figure B11.** Variation in cluster boundaries for 'k' clusters in Region 3 using NetLSD

# Region 3

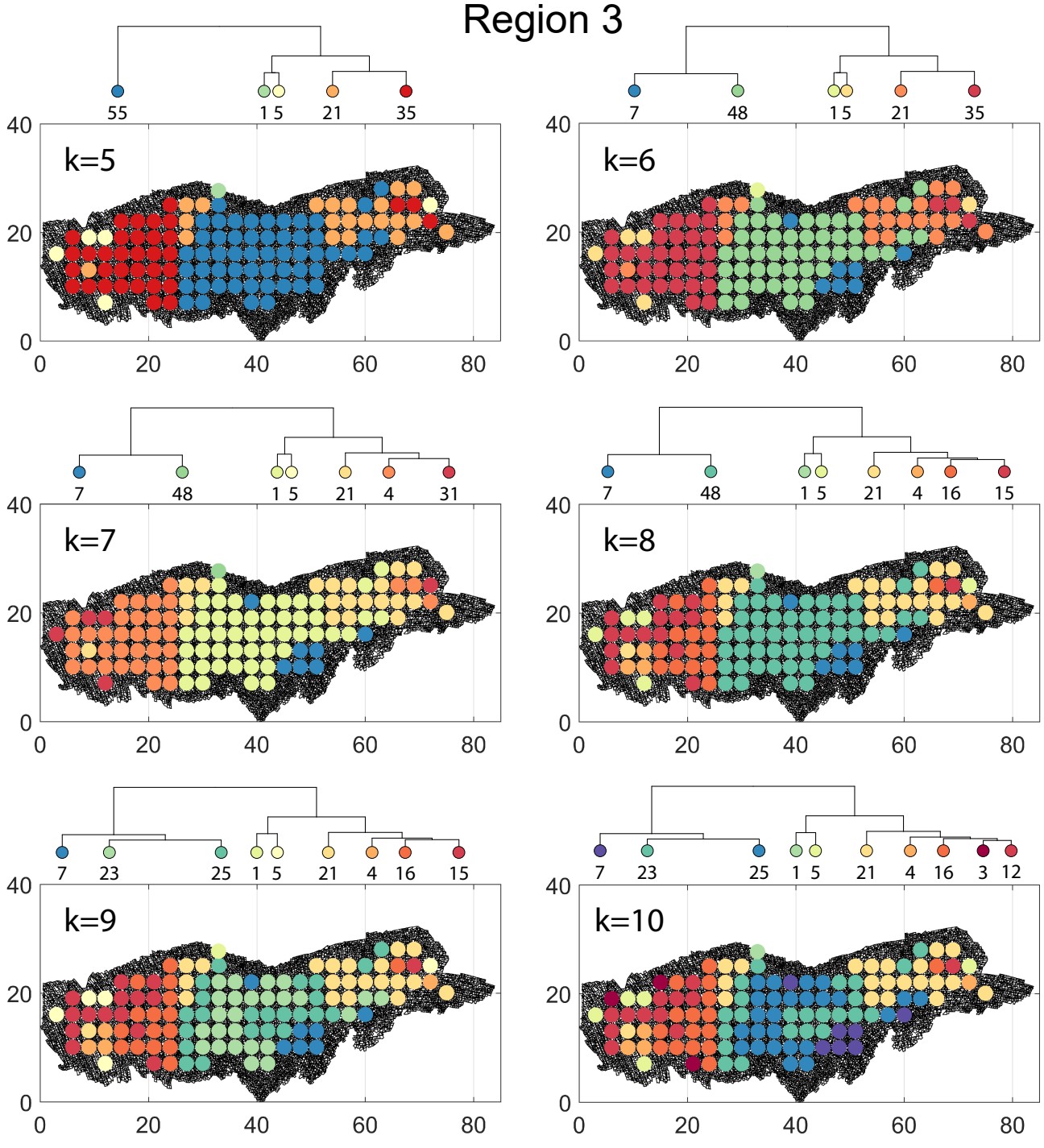

**Figure B12.** Variation in cluster boundaries for 'k' clusters in Region 3 using Portrait Divergence

**Appendix C:  Detailed spatial heterogeneity maps**

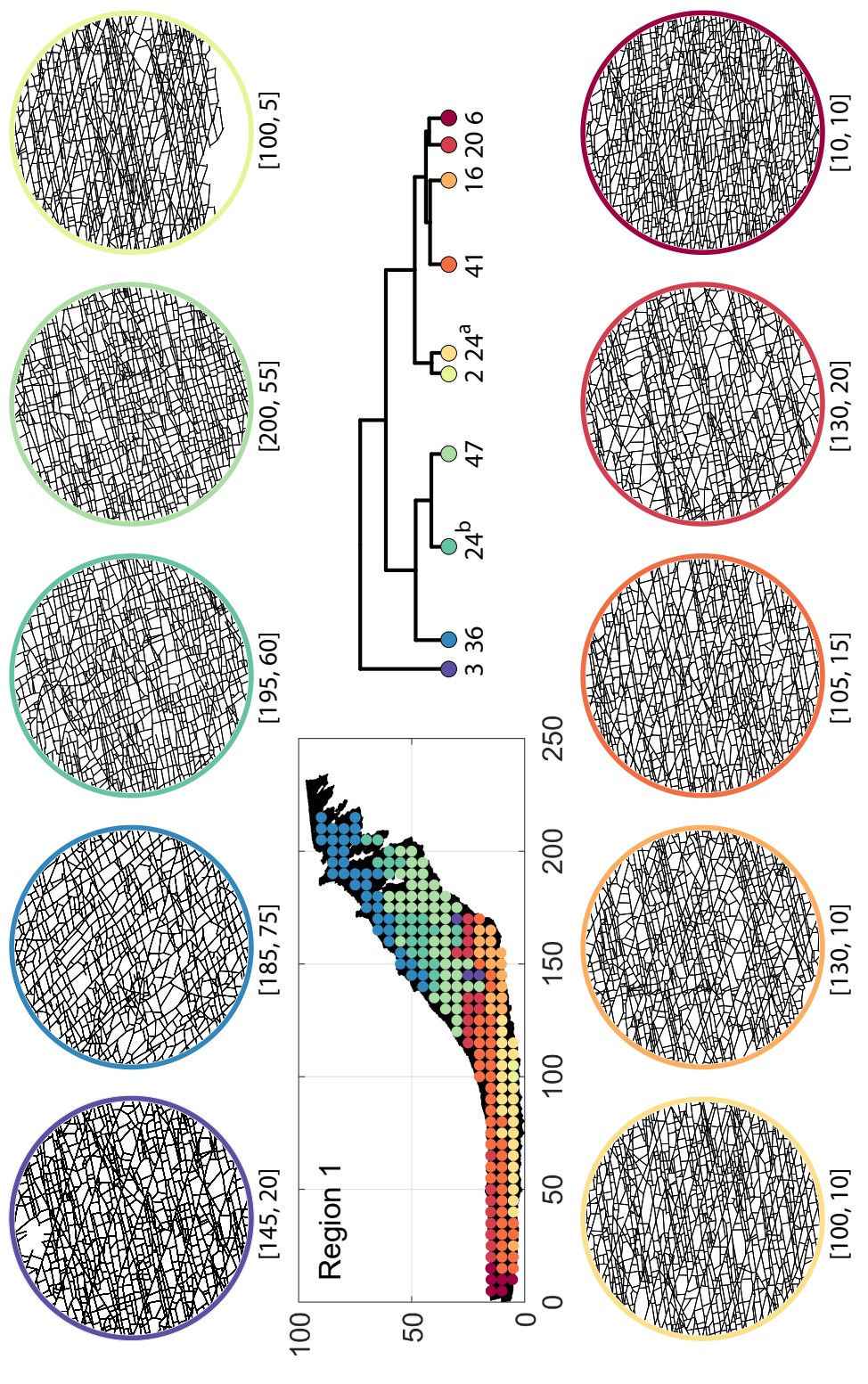

**Figure C1.** Circular subgraph samples depicting variation in fracturing style as identified in the largest ten clusters by fingerprint distance in Region 1. Coordinates of circular sample centres below each subgraph example.

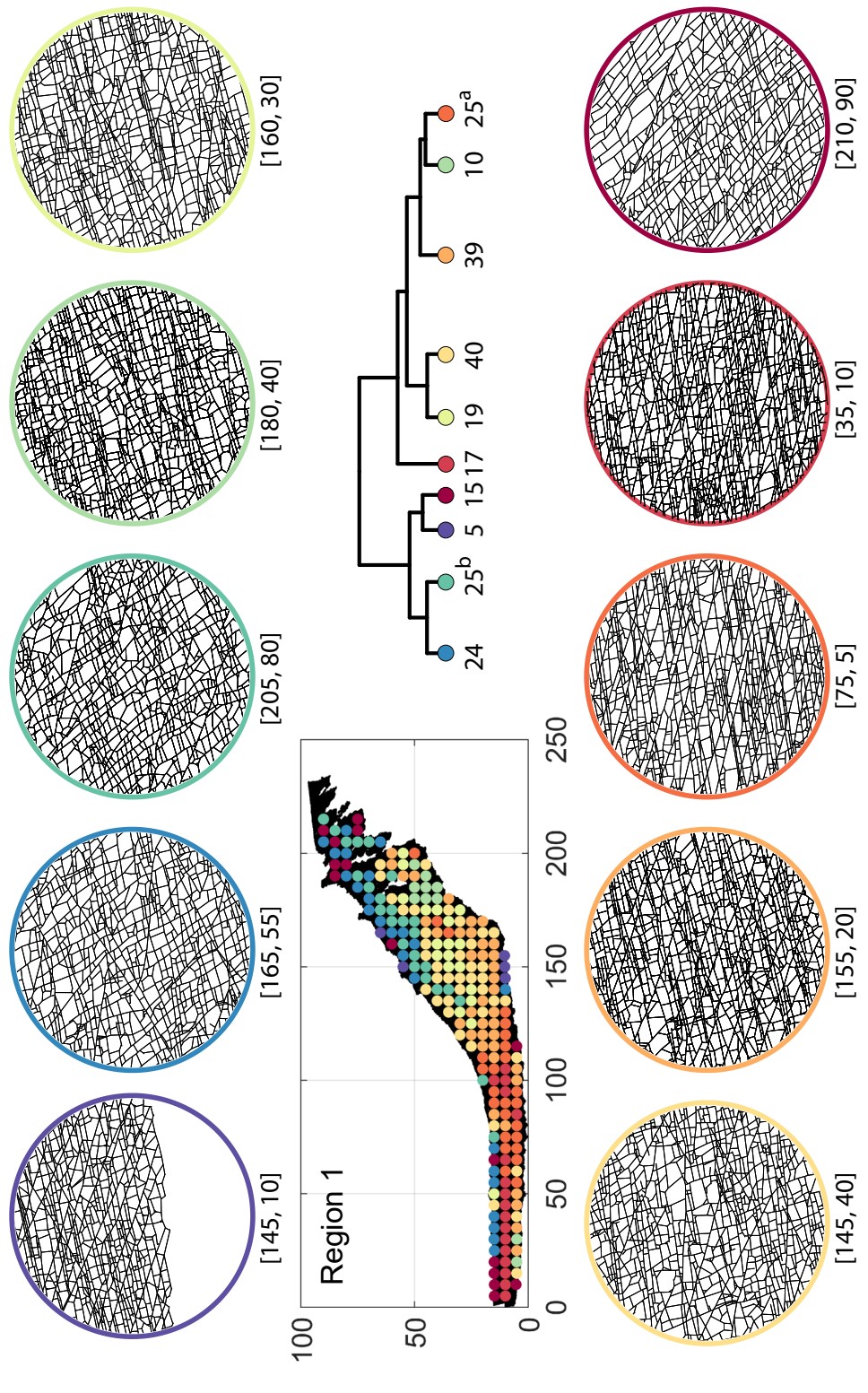

**Figure C2.** Circular subgraph samples depicting variation in fracturing style as identified in the largest ten clusters by D-measure in Region 1. Coordinates of circular sample centres below each subgraph example.

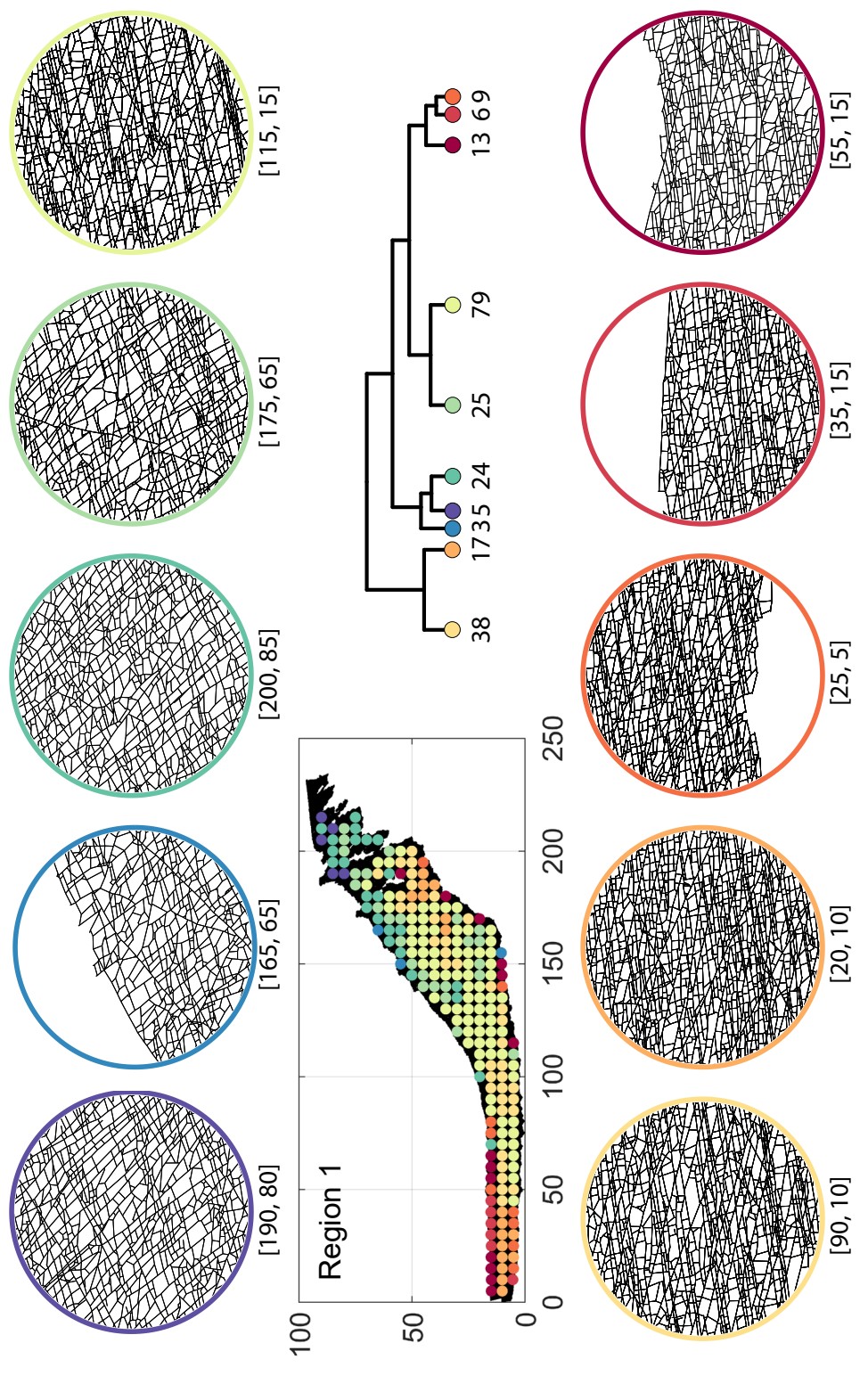

**Figure C3.** Circular subgraph samples depicting variation in fracturing style as identified in the largest ten clusters by portrait divergence in Region 1. Coordinates of circular sample centres below each subgraph example.

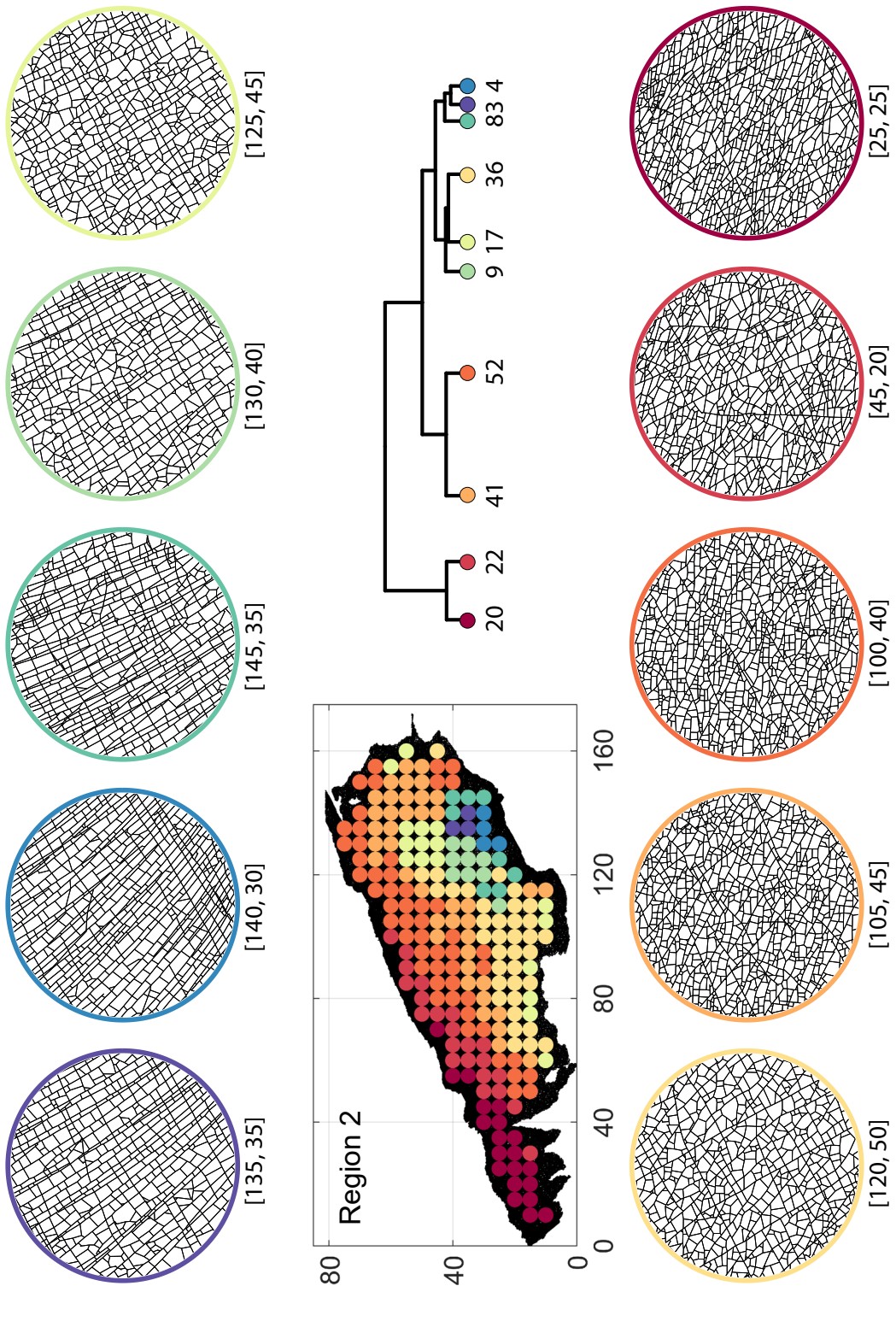

**Figure C4.** Circular subgraph samples depicting variation in fracturing style as identified in the largest ten clusters by fingerprint distance in Region 2. Coordinates of circular sample centres below each subgraph example.

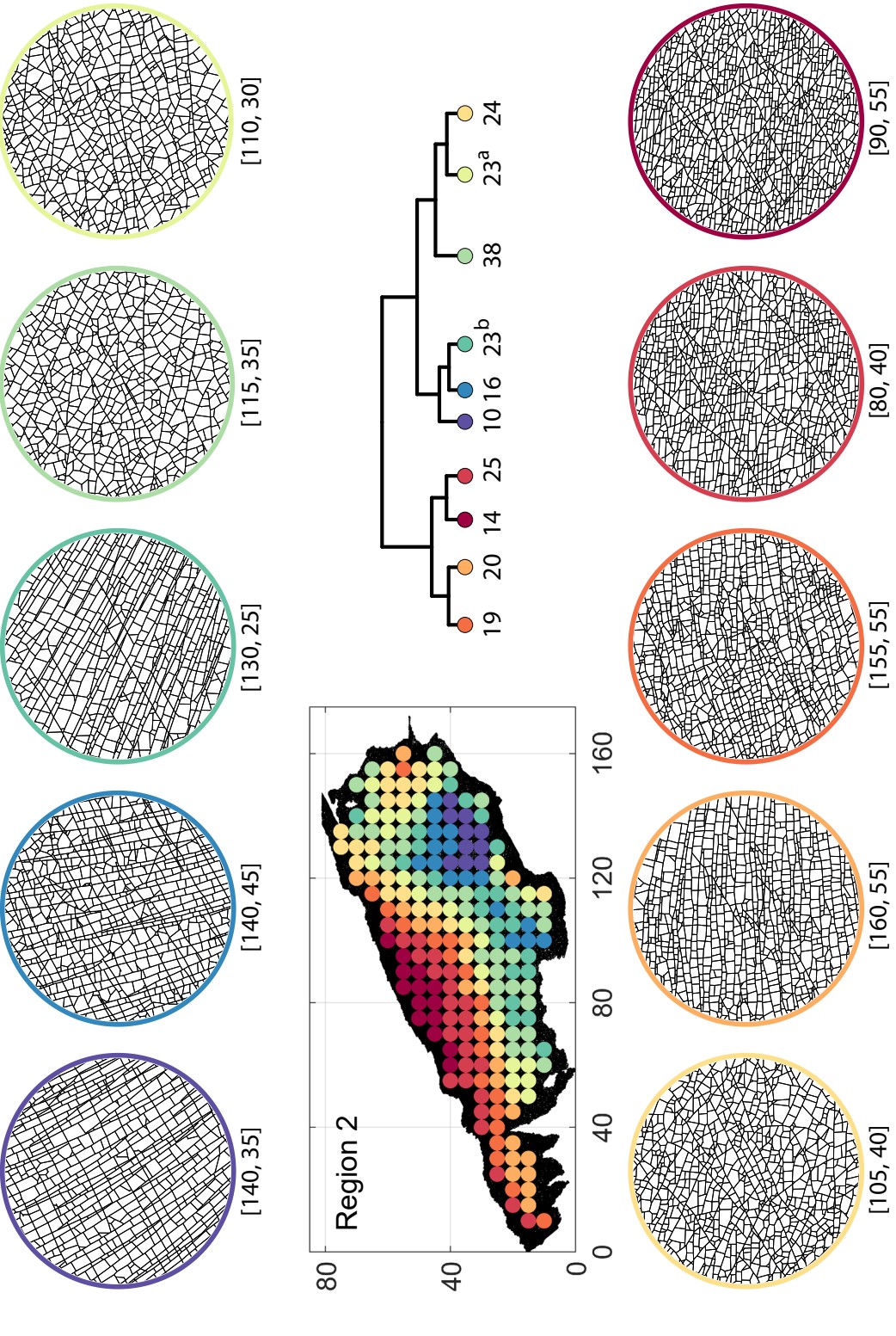

**Figure C5.** Circular subgraph samples depicting variation in fracturing style as identified in the largest ten clusters by D-measure in Region 2. Coordinates of circular sample centres below each subgraph example.

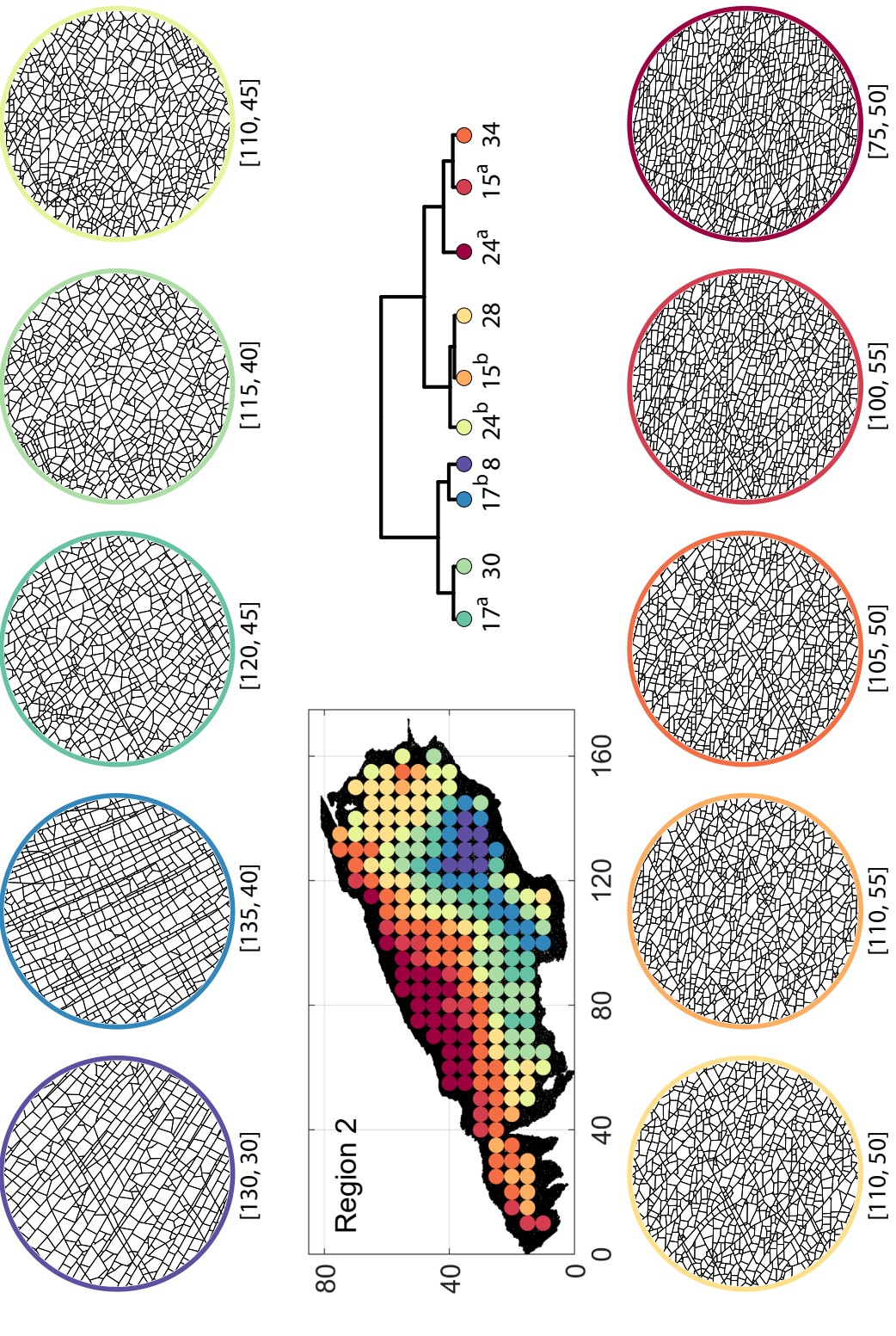

**Figure C6.** Circular subgraph samples depicting variation in fracturing style as identified in the largest ten clusters by portrait divergence in Region 2. Coordinates of circular sample centres below each subgraph example.

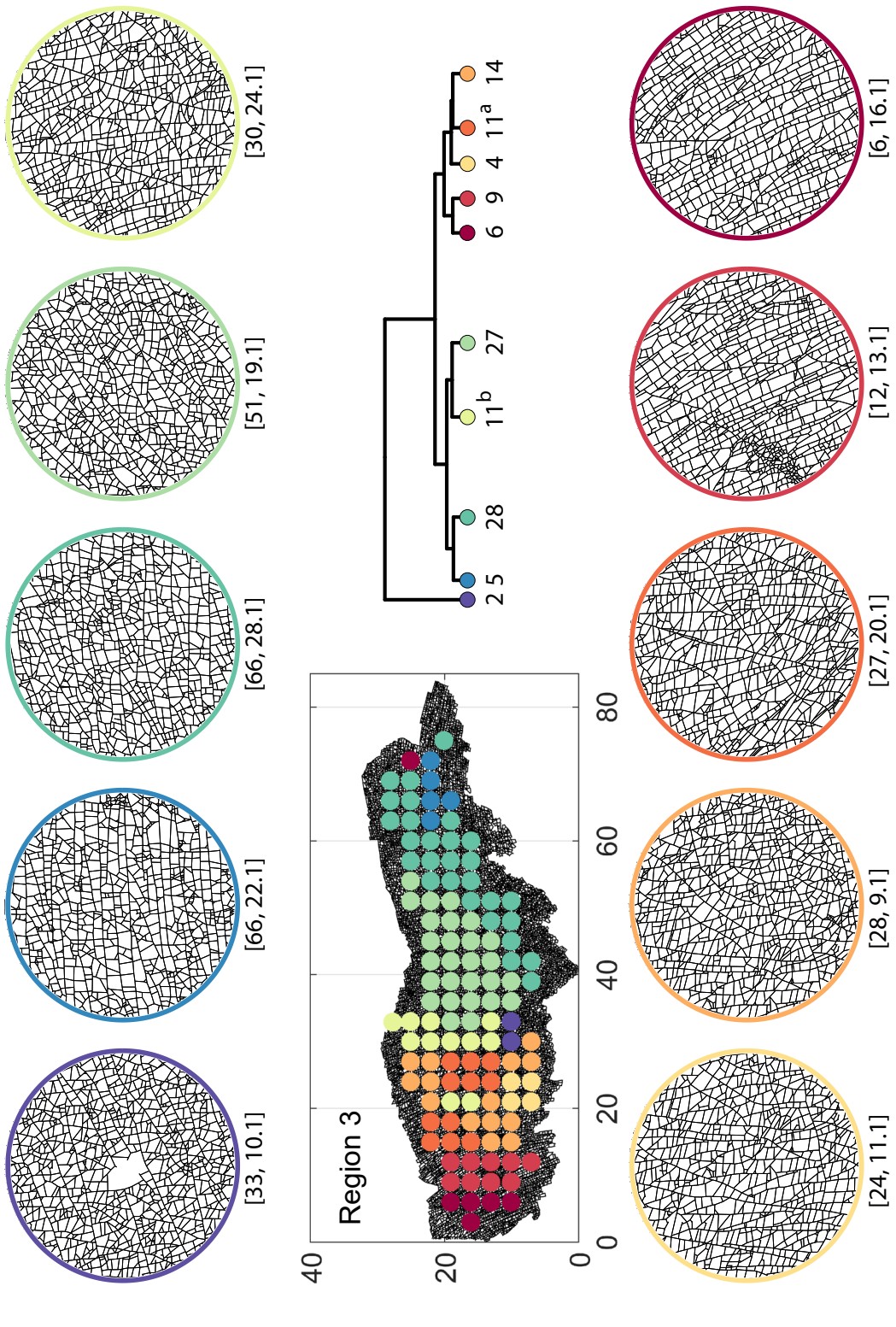

**Figure C7.** Circular subgraph samples depicting variation in fracturing style as identified in the largest ten clusters by fingerprint distance in Region 3. Coordinates of circular sample centres below each subgraph example.

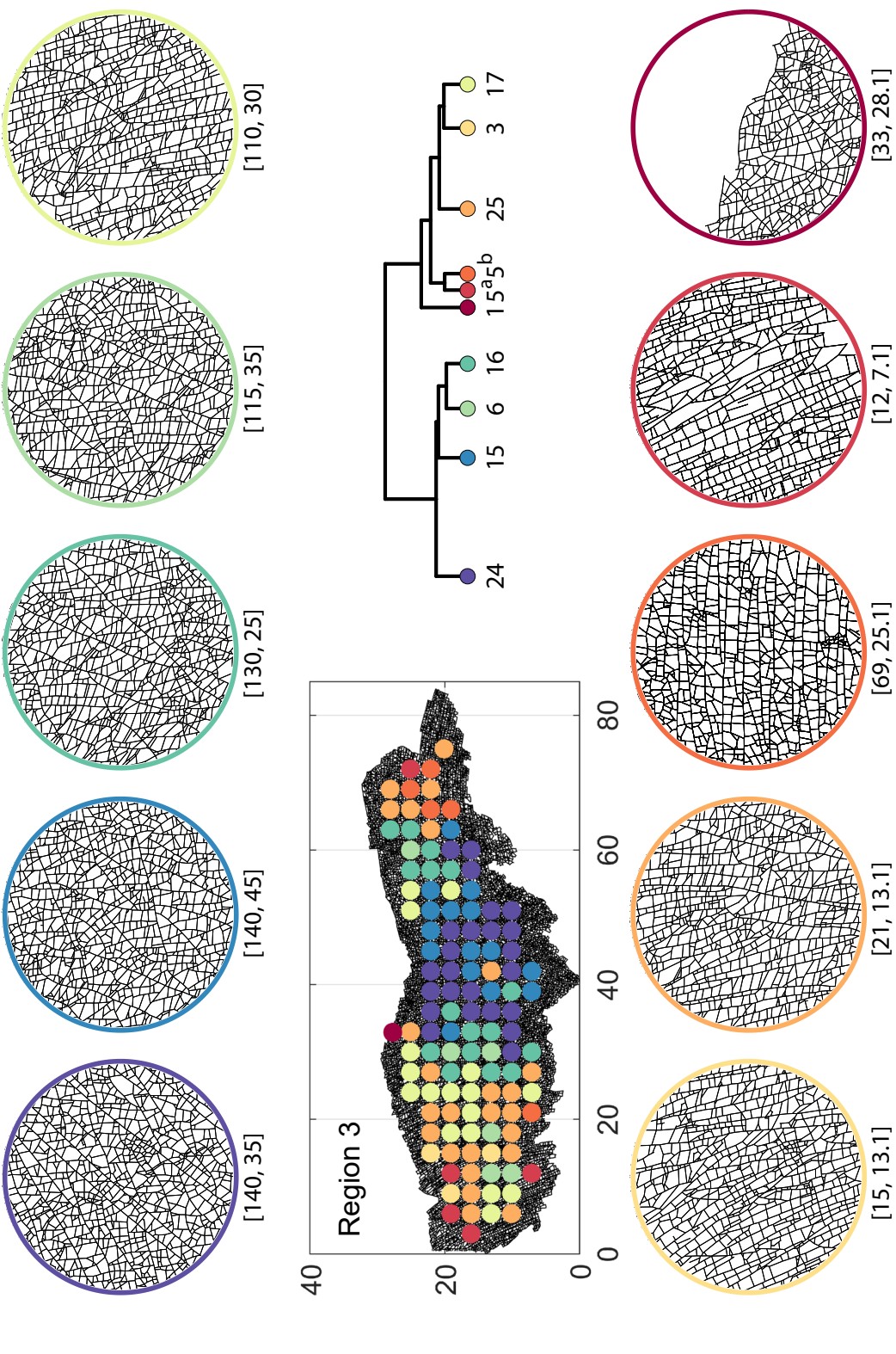

**Figure C8.** Circular subgraph samples depicting variation in fracturing style as identified in the largest ten clusters by D-measure in Region 3. Coordinates of circular sample centres below each subgraph example.

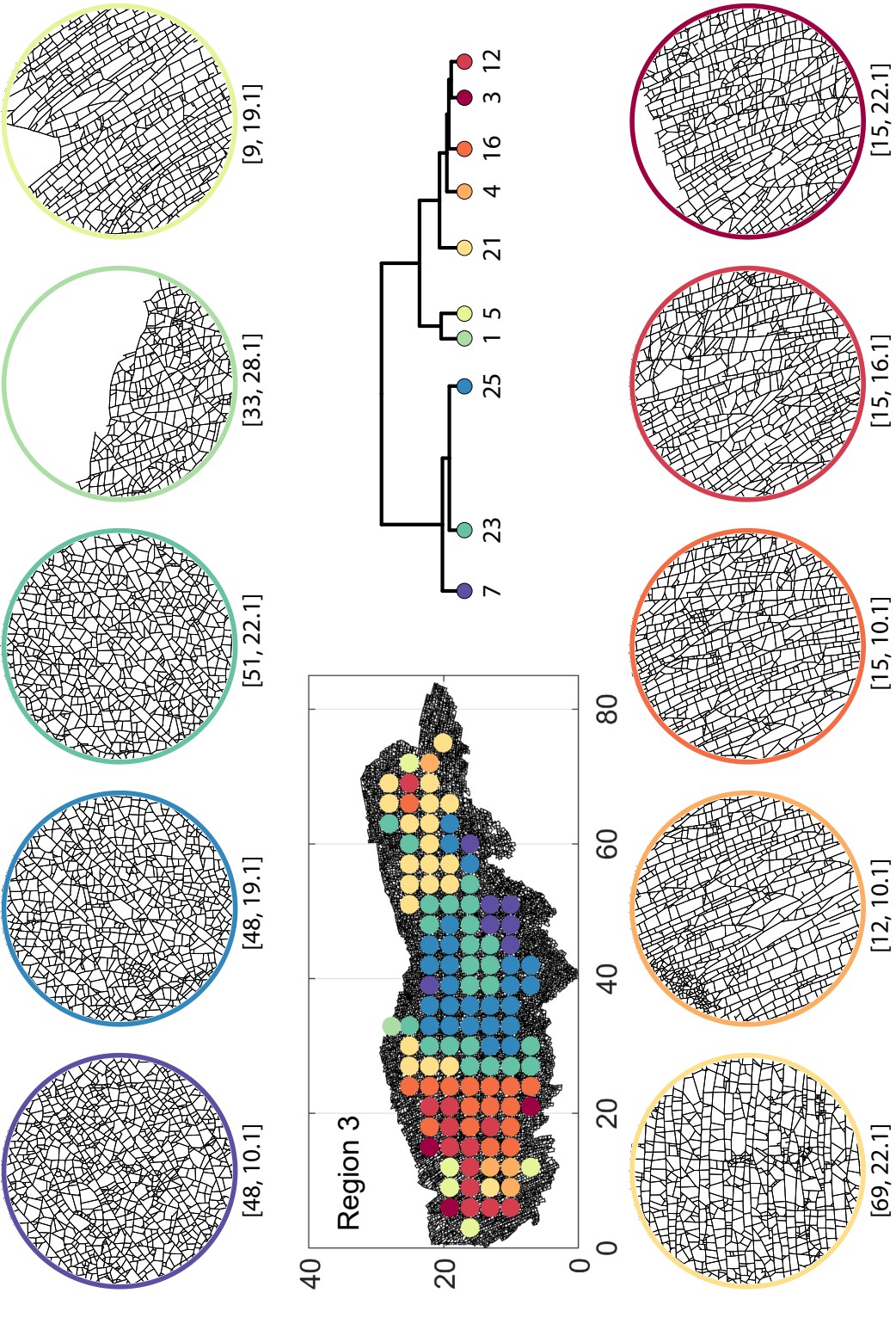

**Figure C9.** Circular subgraph samples depicting variation in fracturing style as identified in the largest ten clusters by portrait divergence in Region 3. Coordinates of circular sample centres below each subgraph example.

*Author contributions.* RP wrote the code to convert shapefiles to graphs, sample sub-graphs, compute fingerprints and fingerprint distances, did the HC analysis and wrote the manuscript with inputs from all authors. GB and JU contributed to development of methodology, structure of the manuscript and discussion of results. DS provided funding and contributed to discussions on the results and methods that are utilized but are not limited to this manuscript.

*Competing interests.* The authors declare that they have no known competing financial interests or personal relationships that could have appeared to influence the work reported in this paper.

*Acknowledgements.* The authors would like to thank Dr. Pierre-Olivier Bruna at TU Delft for useful discussions on spatial variation in fracturing. We would also like to thank Dr. David Sanderson and one other anonymous reviewer for comments that improved the quality of this contribution.

is the line number for Author contributions, 515 for Competing interests.

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
