# Peer review of "Investigating spatial heterogeneity within fracture networks using hierarchical clustering and graph distance metrics"

_Solid Earth, 2021_

## Referee Comment (RC2)

[referee-annotated manuscript omitted]

---

## Author Comment (AC1)

**se-2021-45 Reply to Reviewers: RC1**

**Rahul Prabhakaran, Giovanni Bertotti, Janos Urai, David Smeulders**

**28 - 7 - 2021**

**Dear Reviewer,**

We would like to take this opportunity to thank you for such a detailed review. In the following paragraphs, we reply point-wise to the issues that were raised. We hope to have addressed the concerns sufficiently to warrant reconsideration for publication.

**Kind Regards**

Rahul Prabhakaran (On behalf of all Authors)

**Anon 01**

**Reviewer Comment** Dr. Prabhakaran and co-authors here present a novel approach to spatial network analysis of fractures. This approach involves quantifying hierarchical clustering based on similarity of four statistics: fingerprint distance, D-measure, NetLSD, and portrait divergence. Hierarchical clusters are identified based on the similarity of areal sub-samples of a large fracture map (the example here being the Lilstock anticline, UK). The authors show maps of their results, noting apparent spatial autocorrelation in all analyses except for NetLSD.

The manuscript is mostly well written and appears to me to be mathematically sound. However, despite using several highly sophisticated statistics, the method relies on at least two rather qualitative and/or arbitrary choices: (i) the sampling window size and overlap amount, which would seem to exert an effect on the degree of the aforementioned spatial autocorrelation, and (ii) the selection of the number of clusters present, which should sensitively affect the resulting map patterns.

Moreover, the utility of the approach remains elusive, despite some time spent on it in the introduction and discussion. These four statistics are novel, but their meaning is fairly obscure. The main output of the technique is a map that highlights spatial variation in these statistics, which indeed could be useful. But the same types of patterns show up on maps of more inuitive statistics, like intensity (Figure 9, 14, 19). What does the technique tell you about the spatial arrangement of fractures, fracture connectivity, fracture drivers, etc., that exceeds the explanatory power of more conventional approaches?

Use of the term "clusters" (Line 96-99) is non-standard. Marrett et al. (2018), which the present paper cites, defines a cluster as a place where fracture spacing is smaller than it is elsewhere; as such, a uniformly spaced fracture pattern would lack clusters entirely. This definition is consistent with the framework described earlier in the present study (Line 26-7). Why the change in usage? I suggest using a different term for the output of the present approach that is not already widely used in pattern-quantification literature.

**Authors' Reply**

**Intensity maps versus graph similarity-based clustering maps**

• Fracture intensity maps, despite being a reasonable indicator of spatial heterogeneity, misses out information on fracture network structure and its spatial variation. There is a need to incorporate network attributes in spatial heterogeneity maps and therefore, the attempt to use graph-based measures.

• We note that there are differences between spatial heterogeneity maps derived from HC and graph similarities and those produced by the conventional  $P_{20}$  and  $P_{21}$  computations. Since spatial  $P_{20}$  and  $P_{21}$  are often used to depict spatial variation in fracture network, our objective was to make a qualitative comparison. The main advantage of the proposed methodology as opposed to the ubiquitous spatial  $P_{20}$  and  $P_{21}$  is that a hierarchical structure of variation is derived that encapsulates network properties. Given that network spatial organization influences flow, transport, and geomechanics, the explanatory power of  $P_{20}$  and  $P_{21}$  has limitations.

**Qualitative and arbitrary choices**

The choice of the number of clusters was based on weight of sum of squares (wss) plots depicted in Figures 8, 13, and 18. This plot also referred to as "elbow plots" is often used to identify optimal number of clusters by considering the variation in slope. The choice of five clusters was based after considering the 12 was plots corresponding to the four similarity metrics from each region. A second reason for the choice of 5 clusters is that we wanted to depicts results of spatial clustering across the considered similarity metrics. However, instead of making a single choice, we can easily depict the spatial variation for a range of cut-heights (clusters). When a diverging color scheme is chosen to depict sub-graphs under the various dendrogram branches, it can be observed that there is development of newer clusters and changing of cluster boundaries as one delves deeper into the dendrogram hierarchy. This is depicted in the figures below for the four similarity measures and corresponding to Region 2. In the revision, we extend this to all the Regions.

---

## Author Comment (AC2)

**se-2021-45 Reply to Reviewers: RC2**

Rahul Prabhakaran, Giovanni Bertotti, Janos Urai, David Smeulders

**28 - 7 - 2021**

Dear Prof.Dr.Sanderson,

We thank you for the detailed review of our submission. Please find below responses to the specific comments. We hope that we have sufficiently addressed all points that were raised.

Kind Regards

Rahul Prabhakaran (On behalf of all authors)

**DJS 01**

**Reviewer Comment** This paper is an interesting contribution to an important topic – the analysis spatial variation in fracture networks. The paper introduces some methods, established in other areas but new to this area of earth science. The work is based on some interesting field data, and is both well written and illustrated. I recommend publication, but offer the following comments (see also comments on early sections in annotated pdf).

Authors' Reply The authors would like to thank the reviewer for the detailed review of the manuscript.

**DJS 02**

**Reviewer Comment**

Section 1,2

The authors provide a concise, clear background to the treatment of fracture networks as graphs (Section 2). The level of explanation is appropriate for the rest of the paper, but I highlight, in an annotated version of the manuscript, a few areas that the authors might want to clarify or omit.

**Authors' Reply** We modify the Introduction section to address the issues raised by the reviewer in the annotated manuscript. The modifications that are made in the revised version are described in further detail below.

**DJS 03**

**Manuscript Text** Line 30: Regardless of how fractures' spatial arrangement is defined, quantitative analysis of spatial arrangements invariably leads to quantification of spatial variation.

Reviewer Comment This is a somewhat convoluted sentence and add little - omit.

Authors' Reply This sentence is omitted in the revision.

**DJS 04**

**Manuscript Text** Line 66-67: In fracture networks, the vertices are intersections between fractures and the edges represented by fracture segments connecting the vertices.

**Reviewer Comment** This is only true for the node/branch models (e.g. Sanderson and Nixon 2015) but not for other representations (e.g. Andresen et al 2013).

Authors' Reply We make clear in the revision that fracture intersections as graph nodes is the representation chosen by Sanderson and Nixon (2015) and the alternate representation of tip-to-tip fractures as graph nodes is preferred by others such as Andresen et al. (2013).

**DJS 05**

**Manuscript Text** Line 66-68: By assigning positional information to the vertices (also nodes), fractures in the form of graphs encapsulate both topological and spatial information.

Reviewer Comment (e.g. Sanderson et al. 2019)

Authors' Reply We add this reference to the sentence in the revised manuscript.

**DJS 06**

**Manuscript Text** Line 70-71: The degree of a graph node is simply the number of edges that intersect the particular node.

Reviewer Comment incident at a

Authors' Reply We modify this sentence using the suggested changes.

**DJS 07**

**Manuscript Text** Line 73-74: As can be seen in the case of the primal graph in Fig.1.c, the maximum node degree is 6, with the most common degree value being 3.

**Reviewer Comment** In fracture networks the degree is almost ubiquitously 1, 3, or 4, with higher degrees almost always arising due to problems in resolving closely spaces nodes.

Authors' Reply Yes, we agree. The uneven erosion in some locations in the Lilstock pavement make resolution of closely spacely nodes difficult. We insert the following two sentences in the revision, "It may be noted that node degrees in spatial graph representations of fracture network is most likely to be 1,3 or 4. For fracture networks interpreted from outcrop images, eroded fractures and enlarged apertures may lead to higher degrees due to issues in resolving closely spaced nodes."

**DJS 08**

Manuscript Text Lines 73-75: In the case of the dual graph, as depicted in Fig.1.d, the maximum degree can be much higher, and the longest fractures that have the highest number of intersections also have the highest degree. And resen et al. (2013) and Vevatne et al. (2014) suggested that fracture networks are therefore disassortative in that shorter fractures preferentially attach on to the longer fractures.

**Reviewer Comment** The use of the term "dual" is NOT that use in mathematics of Graph theory, where the dual involves interchange of nodes and regions, rather that nodes and branches/edges.

Authors' Reply Yes, we agree. The use of the term "dual" was popularized by Marc Barthelemy's group in the context of spatial networks/graphs. Since the usage has a different meaning in the mathematics of graph theory we make this clear in the revision by modifying the sentence as "In the case of the alternate representation, referred to as dual graphs by Barthelemy (2018), and depicted in Fig.1.d, the maximum degree can be much higher, and the longest fractures that have the highest number of intersections also have the highest degree."

**DJS 09**

**Manuscript Text** Line 88-91: Graph similarity may be differentiated from graph isomorphism in that the latter comparison can only return a binary outcome. An isomorphism test on two graphs G1 and G2 can only yield two results, either isomorphic or not. Graph similarity on G1 and G2, on the other hand, should return a real-valued quantity that converges to zero when the two graphs approach isomorphism (or complete similarity).

**Reviewer Comment** The comparison of "similarity" (as used in this paper) with "isomorphism" is unclear from this sentence, especially as the authors do not define "isomorphism".

Authors' Reply In the revision, we add a simple definition of isomorphism and also a reference to the book of Van Steen (2010) for the interested audience.

**DJS 10**

Manuscript Text Line 108: HC is an unsupervised statistical clustering method....

**Reviewer Comment** *Hierarchical clustering (HC)*

Authors' **Reply** The acronym is expanded in the revision.

**DJS 11**

Manuscript Text Lines 124 - 130:

- proximity and influence of faults explained by fluid-driven radial-jointing emanating from asperities within fault (Rawnsley et al., 1998; Gillespie et al., 1993 etc)

- spatial variation of thicknesses of intercalated limestone and shale layers (Belayneh, 2004)

- proximity to high-deformation features such as folding (Belayneh and Cosgrove, 2004)

- interplay between regional and local stresses resulting in complex stress fields (Whitaker and Engelder, 2005)

– inheritance from spatial distribution of pre-existing vein / stylolite networks that influenced later joint network development (Wyller, 2019; Dart et al., 1995)

**Reviewer Comment** This list is a bit of a "straw man" and is very incomplete. There are many papers on these joints, and a discussion of how they relate to folds, faults, veins burial, uplift, etc. Do you really want to include this? How does your study contribute to the discussion.

Authors' Reply The paragraph was an attempt to explain some hypotheses postulated by previous authors who have worked in the specific location. Since an in-depth explanation of the reasons behind the spatial heterogeneity deviates from the scope of this manuscript which is more methodological in its aims. Hence in the revised manuscript, we will convert this list into a simplified sentence so as to simply direct the reader to an unexhaustive set of works.

**DJS 12**

**Manuscript Text** Line 141-142: The distribution of joints within a particular length bin is also highly variable.

Reviewer Comment This is the first mention that the fractures are "joints".

Authors' Reply We make it clear in the start of the Section 3 Fracture Datasets, that the Lilstock data is a joint network system.

**DJS 13**

**Manuscript Text** Line 152-153: ... we remove all edges from the sub-graphs emanating from degree-1 nodes that contact the periphery of the circular sample.

**Reviewer Comment** Is this really what you did? The removal of these peripheral edges would convert some 3-nodes (Y) to 2-nodes, unless you also remove the nodes (i.e. produce an edge-induced sub-graph). This is unclear from Fig. 6.

Authors' Reply Yes, the removal of edges create degree-2 nodes which was retained. The objective here was to prevent node degree-distributions to be dominated by I-nodes within the sub-graphs. We make this clear in the revision.

**DJS 14**

Manuscript Text

**Reviewer Comment Section 3**

The introduction to the field area is similarly clear and concise. There is no mention of a recent paper by Procter and Sanderson (2017) that discusses the spatial variability of fractures in the same geological units, just a few kilometres to the west. This paper not only uses graphs to represent the network, but also provides a statistical evaluation of the between layer and within layer variability of fracture intensity.

Authors' Reply We include this reference in the revision. The Kilve outcrops are similar to Lilstock in tectonic origin and the methodology is also of interest to the reader in the context of spatial heterogeneity in same geological units. We add the following sentence in the revision, "Recent work on fractures at the Kilve outcrop (Procter and Sanderson, 2018), exposing the same geological units as that of the regions of interest depicted in Fig.3, conclude that anomalous fracture intensity exists in fracturing at various locations and suggest that variability in fracture intensity cannot be fully explained by variations in thickness, compositional, or textural variations."

**DJS 15**

**Reviewer Comment** Section 4: Methods**

Section 4.2 discusses a range of graph measures used in the subsequent hierarchical clustering. There is a lot of technical detail in the definition of these measures, which is difficult to follow but the sources are all clearly stated. What would be most helpful to the reader would be an evaluation of what each measure is contributing in terms of the geometry and topology of the network. For example:

- The "fingerprint measure" clearly defines a block "shape", both in terms of the number of sides and aspect ratio of the overall shape. Given that most block have 4-6 sides, an average aspect ratio would a little less than 2 and I would expect that this parameter would mainly be reflecting variation in aspect ratio.

- The D-measure is mainly based on the clustering of the node distribution. Given that the divergence and alpha centrality seem to vary little, I would think this measure mainly reflects variation in the node intensity, which seems to be supported by Fig. 14.

It would be good to have a similar evaluation of the other measures. In particular, it is not clear to me how the variation in fracture orientation is captured by these measures. Since the distribution of sets with differing orientation is a major feature of at least two of these regions (Passchier et al 2021), it is surprising that this aspect is omitted from description of the measures and appears to play little role in the clustering.

**Authors' Reply**

• We insert a new figure explaining the aspect ratio variation for a few regular shapes and for some fracture block areas in Section 4.2.1 of the revision.

Figure 1: Illustration of shapefactors

• Within Section 4.2.2, we illustrate the distribution of NND, alpha centrality, and the network node dispersion for the same subgraph as depicted in Fig.7(c) using a new figure.

---

## Author Response (AR1)

**se-2021-45 Authors Response**

Rahul Prabhakaran, Giovanni Bertotti, Janos Urai, David Smeulders

18-8-2021

Prof. Dr. David Healy

Handling Editor - Manuscript SE-2021-45

Solid Earth Journal

Copernicus Publications

Dear Dr.Healy,

Please find attached our revised manuscript which has been improved based on the comments of Dr. David Sanderson and an anonymous reviewer. Also attached is a marked down version created using "latexdiff" that highlights the changes in the revised manuscript compared to the original preprint. Text that has been removed is marked in red, while added or changed text is marked in blue. We had previously replied point-wise to the comments made by anonymous reviewer (https://doi.org/10.5194/se-2021-45-AC1) and Dr. David Sanderson (https://doi.org/10.5194/se-2021-45-AC2). The nomenclature that we used to refer to specific comments is used to refer to the highlighted changes in the revised manuscript and marked down version. We hope that our responses and the revised version of the manuscript is satisfactory to the editor and reviewers and we thank them for their time and effort.

On behalf of co-authors

Rahul Prabhakaran

r.prabhakaran@tudelft.nl
* * *
**Response to Comments by Anonymous Reviewer**

**Anon 01**

**Reviewer Comment:** *Dr. Prabhakaran and co-authors here present a novel approach to spatial network analysis of fractures. This approach involves quantifying hierarchical clustering based on similarity of four statistics: fingerprint distance, D-measure, NetLSD, and portrait divergence. Hierarchical clusters are identified based on the similarity of areal sub-samples of a large fracture map (the example here being the Lilstock anticline, UK). The authors show maps of their results, noting apparent spatial autocorrelation in all analyses except for NetLSD.*

*The manuscript is mostly well written and appears to me to be mathematically sound. However, despite using several highly sophisticated statistics, the method relies on at least two rather qualitative and/or arbitrary choices: (i) the sampling window size and overlap amount, which would seem to exert an effect on the degree of the aforementioned spatial autocorrelation, and (ii) the selection of the number of clusters present, which should sensitively affect the resulting map patterns.*

*Moreover, the utility of the approach remains elusive, despite some time spent on it in the introduction and discussion. These four statistics are novel, but their meaning is fairly obscure. The main output of the technique is a map that highlights spatial variation in these statistics, which indeed could be useful. But the*

*same types of patterns show up on maps of more inuitive statistics, like intensity (Figure 9, 14, 19). What does the technique tell you about the spatial arrangement of fractures, fracture connectivity, fracture drivers, etc., that exceeds the explanatory power of more conventional approaches?*

*Use of the term "clusters" (Line 96-99) is non-standard. Marrett et al. (2018), which the present paper cites, defines a cluster as a place where fracture spacing is smaller than it is elsewhere; as such, a uniformly spaced fracture pattern would lack clusters entirely. This definition is consistent with the framework described earlier in the present study (Line 26-7). Why the change in usage? I suggest using a different term for the output of the present approach that is not already widely used in pattern-quantification literature.*

**Authors' Response**

**Qualitative and arbitrary choices**

(i) From theories of spatial sampling there are many ways to sample spatial data, such as using boxes, hexagons etc, with advantages and disadvantages to each choice. In the case of fracture networks, circular sampling is standard with multiple statistical procedures specific to circular scanlines. Maintaining overlaps between sampling points, or a moving window approach, was necessary since we were looking to identify boundaries in the considered spatial networks where transitions between styles of fracturing occurs. The first consideration on the choice of sampling circle diameter and amount of overlap for the hierarchical clustering (HC) was that of the large computational overhead. The larger the sampling diameter, the more is the number of graph nodes and edges in the clipped circular sub-graph, increasing the computational time to compute the graph similarity. However, when the sampling diameter is too large, heterogeneity within each sample manifests and this variation is absorbed when converted to fingerprints, NND, portraits etc. In the case of the sample overlap, as the overlap decreases, number of samples within the network increases. This increases the computational time although we would have a very high-resolution map of variation. Our choices are justified since the primary focus was on a method to identify spatial variation.

(ii) The choice of the number of clusters was based on weight of sum of squares (wss) plots depicted in Figures 8, 13, and 18 of the original preprint. This plot, also referred to as "elbow plots", is often used to identify optimal number of clusters by considering the variation in slope. The choice of five clusters was primarily based after considering the 12 wss plots corresponding to the four similarity metrics from each region. Another reason for the choice of 5 clusters is that we wanted to compare results of spatial clustering across the considered similarity metrics. However, instead of making a single choice, we can easily depict the spatial variation for a range of cut-heights (clusters). In the revised manuscript, we have added a set of spatial plots (Figures B1-B12) that depict variation for a range of clusters, 4 - 10. With a diverging color scheme. From these plots, the development of newer clusters and changing of cluster boundaries as one delves deeper into the dendrogram hierarchy, can be observed. However, in order to depict the variation and to describe the clusters within the Results section, we use a dendrogram cut of 10 clusters (Figures 17-22 in the revised manuscript). Figures 10-12,15-17,20-22 in the preprint which depicted zoomed-in 10x10 m cutouts from the three regions corresponding to the top four clusters are now modified to depict zoomed-in subsamples for the more granular dendrograms that depict 10 clusters. These are depicted as archetypal circular subsamples in Figures C1-C9 pertaining to each of the ten clusters.

**Utility of the method in explaining connectivity, spatial arrangement, and drivers**

- The method is graph-based hence we can interrogate network properties of clusters in a more detailed manner without loss of any topological information. Since the dendrogram contains similarity information down to each sub-graph sampling, the output is analogous to a network variogram. Both of these are significant improvements over fracture persistence measures. In the revision, we have added rose diagrams and topological summaries of the derived clusters to highlight the differences in network properties (Figures 18,20, and 22 in the revised manuscript). The orientations in each cluster visibly vary across dendrogram branches. Such a variation is identified without explicitly taking into account fracture edge orientations while weighting the primal graphs. Again this information cannot be derived from fracture persistence alone.

- Our aim in this contribution was not to use the proposed technique to delve into details of the structural evolution of the region, but rather focus on a methodology for quantifiying spatial variation, generally applicable to any fracture network. We chose the Lilstock dataset owing to the unprecedented spatial resolution, network size, and spatial extent. The reasons for the complexity and variation in fracturing styles in Lilstock, is still not fully explained despite several decades of work in the area. 2D fracture trace maps from which spatial variation is quantifiable, serves as one tool in a multidisciplinary set that probably requires more evidence, possibly incorporating data fusion from multiple remote sensing sources etc.

**Usage of the term "cluster"**

We agree that the context in which the term "cluster" was used by Marrett et al. (2018), is different from our usage which stems from a statistical sense. The clusters of Marrett et al. (2018) refer to regions of higher density of scanline intersections. In the revision, we rephrase the Lines 26-27 in the preprint so that there is no ambiguity between statistical clustering and the terminology of Marrett et al. (2018) (Line 31-33 in revised manuscript, Line 39-41 in the marked down manuscript).

**Anon 02**

**Reviewer Comment:** [Line 8, 10, 11, 289, 344 Preprint] *S/V agreement.*

**Authors Response** Line 8,10,11 of the Preprint is removed in the revision since the Abstract is rewritten as per suggestion of the second reviewer [Line 14-15 and 19-20 in marked down manuscript].

Line 289 of the Preprint is removed in the revision since the Results section was rewritten for a more granular dendrogram description of 10 clusters compared to the previous 5 [Line 418 in marked down manuscript].

Line 344 of the Preprint is rewritten for grammatical correctness [Line 408-409 in revised mansucript, Line 548-549 in marked down manuscript].

**Anon 03**

**Reviewer Comment:** [Line 21 Preprint] *ambiguous pronoun "they"*

**Authors Response** In the revision, we replace "they" with "fracture patterns" [Line 26 in revised manuscript, Line 34 in marked down manuscript].

**Anon 04**

**Reviewer Comment:** [Line 24 Preprint] *I suggest abandoning this acronym. It is only used once more in the paper (Line 396) and furthermore it does not stand for its definition.*

**Authors Response** We remove the acronym in the revision [Line 29 in revised manuscript, Line 37 in marked down manuscript].

**Anon 05**

**Reviewer Comment:** [Line 27 Preprint] *See definition comment above; also, in this framework I believe that "irregularly spaced" is a synonym for "clustered" and "regularly spaced" is a synonym for "periodically spaced."*

**Authors Response** Yes, Laubach et al. (2018) was referring to scanlines with statistically insignificant clustering in irregularly-spaced fractures, statistically significant clustering in irregularly-spaced fractures, and regularly-spaced fractures. We, therefore, replace the sentence to "Within such a framework, fracture objects are either regularly spaced, irregularly-spaced with statistically significant regions of close spacing, and irregularly-spaced with statistically insignificant regions of close spacing (Laubach et al., 2018)" [Line 31-33 in revised manuscript, Line 39-41 in marked down manuscript].

**Anon 06**

**Reviewer Comment:** [Line 42 Preprint] *None of these drawbacks is unique to 1D analysis.*

**Authors Response** We agree that these issues persist even in 2D trace maps. Hence, we remove this sentence and replace with the following "Scanlines do not provide information on properties such as fracture length, spatial arrangements, and relationships with other fractures." [Line 46-47 in revised manuscript, Line 56-58 in marked down manuscript]

**Anon 07**

**Reviewer Comment:** [Line 57 Preprint] *I am not an expert on stationarity, but I don't think this is true. Can't an irregular pattern nevertheless be stationary, meaning have attributes that do not change with position, above a certain size-scale?*

**Authors Response** The point we were making here is that conventional geostatistics, where transforming a fracture network into a pixel/voxel property such as persistence measures, and then using variograms/semi-variograms to quantify spatial variation, is inadequate for spatial networks. A regular network pattern, that can be decomposed into a unit motif and then replicated over a region of interest, yielding the original pattern, can be considered to be stationary at the spatial scale of the motif. Since this sentence is quite ambiguous in its implication, we rephrase the sentence in the revision [Line 61-62 in revised manuscript, Line 72-74 in marked down manuscript].

**Anon 08**

**Reviewer Comment:** [Line 66 Preprint] *Also "called" nodes?*

**Authors Response** Suggested change is implemented in revision [Line 71 in revised manuscript, Line 83-84 in marked down manuscript].

**Anon 09**

**Reviewer Comment:** [Line 92 Preprint] *What do undirected and weighted mean?*

**Authors Response** We added a new figure to the revision (Figure 2) that depicts examples of an unweighted, weighted, and directed graph along with the corresponding adjacency matrices. A sentence is inserted within the text to describe this figure and make clear the terminology [Line 94-97 in revised manuscript, Line 108-111 in marked down manuscript].

**Anon 10**

**Reviewer Comment:** [Figure 2 Preprint] *The construction of this figure is very difficult to understand until we read about Algorithm 1 below.*

**Authors Response** We modify the caption of the figure (Figure 3 in revision) to refer to the Algorithm.

**Anon 11**

**Reviewer Comment:** [Line 106 Preprint] *: HC undefined acronym until Line 250, and even then it should be made explicit.*

**Authors Response** The acronym is expanded in the revision [Line 123 in revised manuscript, Line 139 in marked down manuscript].

**Anon 12**

**Reviewer Comment:** [Line 124-130 Preprint] *I would add synkinematic cementation (Hooker and Katz, 2015, Am. J. Sci.) to this list.*

**Authors Response** The suggested reference is added in the revision [Line 147-148 in revised manuscript, Line 166-167 in marked down manuscript].

**Anon 13**

**Reviewer Comment:** [Line 124-125 Preprint] *"etc" inappropriate—if more studies have made the point and need to be listed, list them; else put "e.g." and then your chosen, representative references.*

**Authors Response** We rewrite the sentences removing "etc" in the revision as per suggestion [Line 142-148 in the revised manuscript, Line 160-167 in marked down manuscript].

**Anon 14**

**Reviewer Comment:** [Table 1 Preprint] *Are these Edges and Nodes in the primal or dual graph sense?*

**Authors Response** The edges and nodes tabulated refer to the primal graph. The number of fractures in the table are equal to the nodes in the dual graph. We make this clear in the revised manuscript [Line 154-155 in the revised manuscript, Line 173-174 in marked down manuscript].

**Anon 15**

**Reviewer Comment:** [Line 141 Preprint] *I don't understand: how is curvature illustrated in these plots?*

**Authors Response** We were implying that Regions 2 and 3 have quite sinuous fracturing which is qualitatively observed from the networks and the photogrammetric images. To make a more quantitative comparison between the three regions, we introduce a metric that adds up the difference in orientation between each adjacent fracture edge in a tip-to-tip fracture. The slope of the summation of the strikes of constituent fracture edges when plotted as a scatter against total fracture length is then an indicator of curvature and can be used to compare across regions. This plot is inserted as Figure 6 in the revision. From this plot it can be observed that there is positive correlation for all regions. The slope in Region 1 is the least while that in Region 3 is highest. Region 2 containing observations with the most scatter indicating the presence of both curved and straight fractures. A larger, fullpage cutout from Region 2, inserted as Figure 7 in the revision, highlights the radial fracturing regime with very sinuous fractures. These figures are described in the revision [Line 164-167 in revised manuscript, Line 189-192 in marked down manuscript].

**Anon 16**

**Reviewer Comment:** [Line 176 Preprint] *Greater regularity: can you justify this? A long, thin rectangular block would be highly "regular" and yet have a shape factor near zero.*

**Authors Response** The regularity that was implied here is the degree by which a block shape approaches a polygon with equal edge lengths. Such shapes would have the largest value of $\phi$. We inserted a figure to indicate the same (Fig 10 in the revision) and text is added to describe the figure [Line 203-205 in revised manuscript, Line 229-231 in marked down manuscript].

**Anon 17**

**Reviewer Comment:** [Line 179 Preprint] *"distribution of block-face regions"—you mean areas?*

**Authors Response** Word "regions" is replaced with "areas" [Line 208 in revised manuscript, Line 234 in marked down manuscript].

**Anon 18**

**Reviewer Comment:** [Line 203 Preprint] *Line 203: "w.r.t" seems needlessly curt; why not spell it out? Also 424.*

**Authors Response** The acronym is expanded in the revision [Line 233-234,485 in revised manuscript, Line 261,631 in marked down manuscript]

**Anon 19**

**Reviewer Comment:** [Line 258 Preprint] *"HC clusters" redundant*

**Authors Response** The acronym HC is removed in the rewritten results section [Line 331 in marked down manuscript]

**Anon 20**

**Reviewer Comment:** [Line 259-260 Preprint] *Line 260: "decisions on the height"—see major comment above about the arbitrariness of this decision. You put a tremendous amount of effort into quantifying aspects of the spatial arrangement, and then seem to make an arbitrary decision as to how to bin your clusters.*

**Authors Response** In the revision, we depict a range of dendrogram cuts that highlights how dendrogram branches bifurcate with newer emergent clusters and changing cluster boundaries (Figures B1-B12). The "elbow plot" or "weighted-sum-of-squares plot" is generally used as an indicator for the cut-height based on the decreasing statistical significance as quantified by the similarity matrix. In the rewritten Results section, we now depict 10 clusters rather than previous five [Figures 17-18, 19-20, 21-22 in the revised manuscript].

**Anon 21**

**Reviewer Comment:** [Line 297 Preprint] *"correctly" a loaded term without more explanation.*

**Authors Response** We removed this sentence in the revision [Line 434 in marked down manuscript].

**Anon 22**

**Reviewer Comment:** [Line 298,315 Preprint] *Line 298: I would omit "clear" and let the reader judge. Also "clearly" in 315.*

**Authors Response** These usages are removed in the rewritten Results section. [Line 435 and 483 in marked down manuscript].

**Anon 23**

**Reviewer Comment:** [Figure 18 Preprint] *Figure 18 caption: acronyms not used in figure?*

**Authors Response** The caption is modified in the revised figure [Fig. A3 in revision].

**Anon 24**

**Reviewer Comment:** [Line 342-343 Preprint] *"unsupervised"—see major comment above about arbitrary choices. The user has to make the call about what qualifies as a cluster here; I don't think this approach matches the spirit of unsupervised learning.*

**Authors Response** In the statistical literature, hierarchical clustering is "more unsupervised" than techniques such as k-means, since the latter requires an apriori number of clusters to be specified. In HC, the output is a hierarchy which the user can then make a call on the definition of a cluster. Some authors also refer to it as semi-supervised clustering since there is a known distance metric that is applied on the data. Since the usage may be confusing to the audience, we modified the phrasing of the sentence [Line 406-407 in revised manuscript, Line 546-547 in marked down manuscript].

**Anon 25**

**Reviewer Comment:** [Line 357 Preprint] *"striking" for "trending"*

**Authors Response** "Trending" is replaced with "striking" in the revision [Line 422 in revised manuscript, Line 562 in marked down manuscript].

**Anon 26**

**Reviewer Comment:** [Line 359-360 Preprint] *"background-variation"—such an interpretive term, especially considering the only external clustering control you have eliminated, to some degree, is faulting.*

**Authors Response** Yes, we agree with the reviewer that the use of the term background-variation can be ambiguous. We modify this sentence in the revision [Line 424 in revised manuscript, Line 564-565 in marked down manuscript].

**Anon 27**

**Reviewer Comment:** [Line 385-386 Preprint] *"we will tackle"—this is inappropriate. Anyone who wants to extend your work is entirely free to do so. I am sure you are not intentionally "marking your territory," but we need to be especially cognizant of this kind of thing, especially in these days of pre-prints, self-archiving, predatory journals, and similar avenues whereby unscrupulous workers can "jump the gun" on an idea before proper peer review.*

**Authors Response** This point was also raised by another reviewer and we were trying to flag the audience to the way graphs are represented in this contribution. The primal graph representation using weighted graphs that we use for the graph similarity computation does not explictly encode orientation information. Hence, if we consider a simple square lattice and rotate it by 45 degrees, the graph similarity measure would not detect such a difference. As we show in Figures 18,20, and 22 in the revision, for natural networks, the method is still able to identify variations in orientations even when orientation is not explictly included as primal graph edge weights. In the preprint, we proposed the idea of normalizing the edge weights using a combination of both length and orientation. Since we did not attempt this in the course of this work, we remove this sentence in the revision [Line 590-591 in marked down manuscript].

**Anon 28**

**Reviewer Comment:** [Line 391 Preprint] *"Line 391: inherent non-stationarity" see comment on Line 57*

**Authors Response** We feel that this context is different from our previous reference to non-stationarity in fracture networks in Line 57 of preprint. In the previous case we are referring to ground observations that indicate for fractures, simple motifs or patterns do not replicate in a regular fashion forming a network. It is more common to see network spatial distribution vary quite irregularly.

In Line 391 of preprint, we are making the point that in conventional geostatistics, spatial data are classified as continuously varying data, point data, and gridded data. Many spatial statistics procedures are well-developed for these three specific data types, but in the case of spatial network data it is not the case. The work of Thovert et al. (2017) use point process-based approaches to create DFNs while the work of Bruna et al. (2019) treat fractures as a facies types distributed as regularly-gridded pixels. Both the approaches try to recreate spatial variation without explicitly using a spatial network representation. Our contribution is directed to addressing the issue of spatial variation through the spatial graph representation and the reference to stationarity is for network data that does not fall into the three general classes of spatial data.

**Anon 29**

**Reviewer Comment:** [Line 392 Preprint] *Line 392: define "disassortativity"*

**Authors Response** Since this is an important property of naturally occuring networks we briefly define dissortativity with a reference to Newman (2002) in Section 2.1 of the revision. Disassortative networks have a negative assortativiy coefficient. In the case of the three spatial networks we considered, the assortativity coefficients for the graphs in the dual form are negative. This is also reported by Andresen et al. (2013) and Vevatne et al. (2014) and this information is added to Section 2.1 [Line 86-89 in revised manuscript, Line 98-101 in marked down manuscript].

**Anon 30**

**Reviewer Comment:** [Line 397 Preprint] *"stationarity" for "stationariness"?*

**Authors Response** The term is replaced [Line 460 in revised manuscript, Line 602 in marked down manuscript].

**Anon 31**

**Reviewer Comment:** [Line 397 Preprint] *"have to be made based on hard data"—see major comment on arbitrarily choosing the number of clusters, above. Have you made your decisions based on hard data?*

**Authors Response** In this work, we do not address the network extrapolation problem which in conventional geostatistics requires an assumption of stationarity. The hard data that we are referring to in this context are large-scale networks that are extensive and spatially continuous enough to be able to identify the spatial variation. We present our work as a graph-based method to identify variation in natural networks which can be then be used to guide decisions on stationarity to address the network extrapolation problem.

**Anon 32**

**Reviewer Comment:** [Line 408 Preprint] *define "modularity"*

**Authors Response** Modularity is a quantitative measure introduced by Newman and Girvan (2004) to identify possible clusterings within a graph. Modularity represents a fraction of graph edges connecting nodes within a cluster subtracted by an expected fraction if the edge distributions between nodes are random. Maximum possible modularity of 1, indicates networks with strong community structure. Blondel et al. (2008) and Traag et al. (2019) present fast computational procedures to compute graph clustering based on the modularity measure. Since the work of Blondel et al. (2008) and Traag et al. (2019) is based on Newman and Girvan (2004), we inserted this reference in the revision so that the reader can easily refer to the original sources [Line 471 in revised manuscript, Line 613 in marked down manuscript].

**Anon 33**

**Reviewer Comment:** [Line 411 Preprint] *What is graph partitioning, and is that a worthy goal?*

**Authors Response** Graph partitioning is an interchangeable term for graph clustering. The sentence is modified to make clear that partitioning creates clusters [Line 474 in revised manuscript, Line 616 in marked down manuscript].

**Anon 34**

**Reviewer Comment:** [Line 418 Preprint] *"hierarchical" for "hierarchically"?*

**Authors Response** The spelling is corrected in the revision [Line 482 in revised manuscript, Line 627 in marked down manuscript].

**Response to Dr. David Sanderson**

**DJS 03**

**Reviewer Comment:** [Line 30 Preprint] *This is a somewhat convoluted sentence and add little - omit.*

**Authors Response** This sentence is removed in the revision [Lines 43-45 in marked down manuscript].

**DJS 04**

**Reviewer Comment:** [Line 66-67 Preprint] *This is only true for the node/branch models (e.g. Sanderson and Nixon 2015) but not for other representations (e.g. Andresen et al 2013).*

**Authors Response** We inserted the reference to Sanderson and Nixon (2015) at the end of the sentence in the Revised manuscript [Line 71 in revised manuscript, Line 83 in marked down manuscript].

**DJS 05**

**Reviewer Comment:** [Line 66-68 Preprint] *(e.g. Sanderson et al. 2019)*

**Authors Response** The suggested reference, Sanderson et al. (2019), is added to the sentence [Line 72 in revised manuscript, Line 84 in marked down manuscript].

**DJS 06**

**Reviewer Comment:** [Line 70-71 Preprint] *incident at a*

Sentence is modified as per suggestion [Line 77 in revised manuscript, Line 89 in marked down manuscript].

**DJS 07**

**Reviewer Comment:** [Line 73-74 Preprint] *In fracture networks the degree is almost ubiquitously 1, 3, or 4, with higher degrees almost always arising due to problems in resolving closely spaces nodes.*

**Authors Response** We add two sentences in the revision to highlight that erosion in outcrop fractures lead to issues in resolving closely spaced nodes [Line 79-82 in revised manuscript, Line 92-94 in marked down manuscript].

**DJS 08**

**Reviewer Comment:** [Line 73-75 Preprint] *The use of the term "dual" is NOT that use in mathematics of Graph theory, where the dual involves interchange of nodes and regions, rather that nodes and branches/edges.*

**Authors Response** We modify this sentence to make clear the context in which the term "dual" is used with the proper citation [Line 83-85 in revised manuscript, Line 95-97 in marked down manuscript].

**DJS 09**

**Reviewer Comment:** [Line 88-91 Preprint] *The comparison of "similarity" (as used in this paper) with "isomorphism" is unclear from this sentence, especially as the authors do not define "isomorphism".*

**Authors Response** A simple definition of graph isomorphism and a reference to the book of Van Steen (2010) is added [Line 102-104 in revised manuscript, Line 116-119 in marked down manuscript].

**DJS 10**

**Reviewer Comment:** [Line 108 Preprint] *Hierarchical clustering (HC)*

**Authors Response** The acronym is expanded in the revision [Line 123 in revised manuscript, Line 139 in marked down manuscript].

**DJS 11**

**Reviewer Comment:** [Lines 124-130 Preprint] *This list is a bit of a "straw man" and is very incomplete. There are many papers on these joints, and a discussion of how they relate to folds, faults, veins burial, uplift, etc. Do you really want to include this? How does your study contribute to the discussion.*

**Authors Response** Since our contribution is mainly methodological, we rephrase the list as a sentence that briefs the reader to an unexhaustive set of works that seek to explain the spatial variation in the Lilstock. The first reviewer also suggested an additional reference to be appended to the list [Line 142-148 in revised manuscript, Line 160-167 in marked down manuscript].

**DJS 12**

**Reviewer Comment:** [Line 141-142 Preprint] *This is the first mention that the fractures are "joints".*

**Authors Response** We make it clear in the start of the Section 3 Fracture Datasets, that the Lilstock data is a joint network system [Line 135-136 in revised manuscript, Line 152-153 in marked down manuscript].

**DJS 13**

**Reviewer Comment:** [Line 152-153 Preprint] *Is this really what you did? The removal of these peripheral edges would convert some 3-nodes (Y) to 2-nodes, unless you also remove the nodes (i.e. produce an edge-induced sub-graph). This is unclear from Fig. 6.*

**Authors Response** Yes, the removal of edges create degree-2 nodes which was retained. The objective here was to prevent node degree-distributions to be dominated by I-nodes within the sub-graphs.

**DJS 14**

**Reviewer Comment:** [Section 3 Fracture Datasets in Preprint] *The introduction to the field area is similarly clear and concise. There is no mention of a recent paper by Procter and Sanderson (2017) that discusses the spatial variability of fractures in the same geological units, just a few kilometres to the west. This paper not only uses graphs to represent the network, but also provides a statistical evaluation of the between layer and within layer variability of fracture intensity.*

**Authors Response** The suggested reference is included in the revision [Line 148-151 in revised manuscript, Line 167-170 in marked down manuscript].

**DJS 15**

**Reviewer Comment:** [Section 4 Methods in Preprint] *Section 4.2 discusses a range of graph measures used in the subsequent hierarchical clustering. There is a lot of technical detail in the definition of these measures, which is difficult to follow but the sources are all clearly stated. What would be most helpful to the reader would be an evaluation of what each measure is contributing in terms of the geometry and topology of the network. For example:*

*- The "fingerprint measure" clearly defines a block "shape", both in terms of the number of sides and aspect ratio of the overall shape. Given that most block have 4-6 sides, an average aspect ratio would a little less than 2 and I would expect that this parameter would mainly be reflecting variation in aspect ratio.*

*- The D-measure is mainly based on the clustering of the node distribution. Given that the divergence and alpha centrality seem to vary little, I would think this measure mainly reflects variation in the node intensity, which seems to be supported by Fig. 14.*

*It would be good to have a similar evaluation of the other measures. In particular, it is not clear to me how the variation in fracture orientation is captured by these measures. Since the distribution of sets with differing orientation is a major feature of at least two of these regions (Passchier et al 2021), it is surprising that this aspect is omitted from description of the measures and appears to play little role in the clustering.*

**Authors Response**

- A figure is added explaining the aspect ratio variation for a few regular shapes and for some fracture block areas in Fig. 10 of the revision under Section 4.2.1.

- In Figure 7 of the preprint, we depicted an example fracture graph from Region 1 and its fingerprint. In the revision, this figure is modified to include two example fracture graphs and their fingerprints to illustrate the visual differences in fracturing and the effect on the fingerprint. This corresponds to Fig. 11 in the revision. To illustrate the graph properties that are being compared for the other similarity measures for the two example graphs we insert three figures (Figs. 12-14) in the revision. The computed similarities for all the four measures is then summarized in a new table (Table. 3) in the revision.

- We do not incorporate fracture orientation directly in to the edge weights but only the segment lengths. The work of Passchier et al. (2021) did not consider fully traced networks but focussed on picking tip-to-tip sets manually. Despite not having considered orientation explicitly within the graph edge weights, the clusters are still able to identify differences in orientation. Examples of the variation of orientations pertaining to clusters for the similarity measures for the three considered regions are depicted in three new figures (Fig. 18,20,22) in the revision. (Note: We exclude NetLSD for the lack of spatial autocorrelation in the results.)

**DJS 16**

**Reviewer Comment:** [Section 5 Results in Preprint] *This provides a detailed analysis of three regions and presents the results of the mapping of spatial variability in terms of the HC of the measures used. The results are presented through sets of five similar diagrams for each of the three regions. Some of the material in these diagrams could be transferred to "supplementary material" (e.g. heat maps and dendrograms of the clustering).*

*It is probably worth comparing the characteristics of each region. From Table 1, the number of fractures/branches/nodes, with similar ranges of fracture intensity (P21) in Figs 9, 14 and 19. The average degrees are also very similar (~3.15) indicating a close approximation to a 3-regular graph (or mesh). Apart from the "fingerprint" of block shape (Fig. 7e), there is no information on the distributions of the other measures used in the clustering.*

*It seems to me, that Region 2 illustrates the strengths and weaknesses of the methodology, so starting with a complete analysis of this area would make sense. The other two regions could then be treated more concisely, with emphasis on what they contribute to the study.*

**Authors Response**

- The plots depicting the distance matrix heatmaps, dendrograms and w.s.s plots are moved to an Appendix A in the revision [Figs A1-A3 in the revised manuscript]. The zoomed-in section plots for the clusters, Figs. 10-12,15-17,20-22 in the preprint are now moved to Appendix C in the revision [Figs C1-C9 in the revised manuscript].

- We add two figures to illustrate the regionwise graph properties [Fig. 15 and Fig. 16 in the revised manuscript].

**DJS 17**

**Reviewer Comment:** [Line 378-386 Preprint] *In the section on choice of graph metrics, the paragraph from lines 378-386 is key to this paper. An unweighted graph cannot contain information on geometry, since a graph is invariant to change in shape and size. Thus, geometry must be represented through the embedding of the graph in a (geographical) space or by including geometrical measures in the weights. The former allows specification of orientation and length, and measure of the frequency and intensity of elements to unit area. The way this paragraph is written implies that length and orientation could be incorporated but were not in the present study – this is "shooting one's self in the foot".*

**Authors Response** In our work, the graph edges are weighted using geometric lengths of the fracture segments. We have not incorporated orientation as a weighting parameter for the edges. As pointed out earlier under DJS 15, despite not using orientation explicitly in the edge weights, the similarity measures are still able to identify variation in orientations. A simple dot-product as a scalar weight is not straight-forward as the weights would not have a clear minimum or maximum, as the weight would combine circularly distributed data (orientations) as well as data that follow a negative power-law (segment lengths). The last sentence was intended to convey this difficulty, but since the wording is confusing we remove it in the revision [Line 590-591 in marked down manuscript].

**Reviewer Comment:** [Abstract and Conclusion] *Both these sections are written in a vague way, expressing the aims and aspirations of the study, instead of focusing on the main findings.*

**Authors Response** The Abstract and Conclusions are rewritten in the revision to better convey the main findings of the Contribution [Line 2-18, Line 476-498 in revised manuscript; Line 3-26, Line 618-643 in marked

down manuscript].

**REFERENCES**

Andresen, C., Hansen, A., Le Goc, R., Davy, P., and Hope, S.: Topology of fracture networks, 1, 7, https://doi.org/10.3389/fphy.2013.00007, 2013.

Blondel, V. D., Guillaume, J.-L., Lambiotte, R., and Lefebvre, E.: Fast unfolding of communities in large networks, 2008, P10008, https://doi.org/10.1088/1742-5468/2008/10/p10008, 2008.

Bruna, P., Prabhakaran, R., Bertotti, G., Straubhaar, J., Plateaux, R., Maerten, L., Mariethoz, G., and Meda, M.: The mps-based fracture network simulation method: Application to subsurface domain, 2019, 1–5, https://doi.org/10.3997/2214-4609.201901679, 2019.

Laubach, S. E., Lamarche, J., Gauthier, B. D. M., Dunne, W. M., and Sanderson, D. J.: Spatial arrangement of faults and opening-mode fractures, 108, 2–15, https://doi.org/10.1016/j.jsg.2017.08.008, 2018.

Marrett, R., Gale, J. F. W., Gómez, L. A., and Laubach, S. E.: Correlation analysis of fracture arrangement in space, 108, 16–33, https://doi.org/10.1016/j.jsg.2017.06.012, 2018.

Newman, M. E. J.: Assortative mixing in networks, 89, 208701, https://doi.org/10.1103/PhysRevLett.89.208701, 2002.

Newman, M. E. J. and Girvan, M.: Finding and evaluating community structure in networks, 69, 026113, https://doi.org/10.1103/PhysRevE.69.026113, 2004.

Passchier, M., Passchier, C. W., Weismüller, C., and Urai, J. L.: The joint sets on the lilstock benches, uk. Observations based on mapping a full resolution uav-based image, 104332, https://doi.org/https://doi.org/10.1016/j.jsg.2021.104332, 2021.

Sanderson, D. J. and Nixon, C. W.: The use of topology in fracture network characterization, 72, 55–66, https://doi.org/10.1016/j.jsg.2015.01.005, 2015.

Sanderson, D. J., Peacock, D. C. P., Nixon, C. W., and Rotevatn, A.: Graph theory and the analysis of fracture networks, 125, 155–165, https://doi.org/10.1016/j.jsg.2018.04.011, 2019.

Thovert, J.-F., Mourzenko, V., and Adler, P.: Percolation in three-dimensional fracture networks for arbitrary size and shape distributions, 95 (4), 042112, https://doi.org/10.1103/PhysRevE.95.042112, 2017.

Traag, V. A., Waltman, L., and Eck, N. J. van: From louvain to leiden: Guaranteeing well-connected communities, 9, 5233, https://doi.org/10.1038/s41598-019-41695-z, 2019.

Van Steen, M.: Graph theory and complex networks: An introduction, 1st ed., 2010.

Vevatne, J. N., Rimstad, E., Hope, S. M., Korsnes, R., and Hansen, A.: Fracture networks in sea ice, 2, 21, https://doi.org/10.3389/fphy.2014.00021, 2014.